

# Optimizing Clifford gate generation for measurement-only topological quantum computation with Majorana zero modes

Alan Tran[1]⋆, Alex Bocharov[2], Bela Bauer[3] and Parsa Bonderson[3]

**1** Department of Physics, University of California, Santa Barbara, CA 93106, USA
**2** Microsoft Quantum, Redmond, Washington 98052, USA
**3** Microsoft Station Q, Santa Barbara, California 93106-6105, USA

⋆ adtran@physics.ucsb.edu

## Abstract

One of the main challenges for quantum computation is that while the number of gates required to perform a non-trivial quantum computation may be very large, decoherence and errors in realistic quantum architectures limit the number of physical gate operations that can be performed coherently. Therefore, an optimal mapping of the quantum algorithm into the physically available set of operations is of crucial importance. We examine this problem for a measurement-only topological quantum computer based on Majorana zero modes, where gates are performed through sequences of measurements. Such a scheme has been proposed as a practical, scalable approach to process quantum information in an array of topological qubits built using Majorana zero modes. Building on previous work that has shown that multi-qubit Clifford gates can be enacted in a topologically protected fashion in such qubit networks, we discuss methods to obtain the optimal measurement sequence for a given Clifford gate under the constraints imposed by the physical architecture, such as layout and the relative difficulty of implementing different types of measurements. Our methods also provide tools for comparative analysis of different architectures and strategies, given experimental characterizations of particular aspects of the systems under consideration. As a further non-trivial demonstration, we discuss an implementation of the surface code in Majorana-based topological qubits. We use the techniques developed here to obtain an optimized measurement sequence that implements the stabilizer measurements using only fermionic parity measurements on nearest-neighbor topological qubit islands.

# 1  Introduction

Recent experimental progress has established the existence of Majorana zero modes (MZMs) [1–3], in particular in their incarnation in hybrid semiconductor-superconductor heterostructures [4], as one of the most promising platforms for realizing topological quantum computation [5]. As the evidence for the successful experimental realization of such topological phases mounts [4], the question arises how to assemble a network of topological superconductors in a way that allows practical quantum information processing on many qubits. While several proposals have been put forward [6–9], we will focus here on the measurement-only approach of Ref. [9].

Measurement-only topological quantum computation, first proposed in Refs. [10, 11], appears particularly favorable in the context of MZMs since it avoids having to physically move the MZMs, which are bound to macroscopic defects (such as the ends of wires) and may be difficult to move without strongly disturbing the system. Instead, braiding transformations are effectively generated through a series of (potentially non-local) measurements on sets of MZMs involving the MZMs that encode the computational state that is to be manipulated, and another set of MZMs that serve as ancillary degrees of freedom. In the architectures proposed in Ref. [9], the required measurements are performed by coupling groups of MZMs to quantum dots, thus affecting the energy spectrum of the dot in a way that can be measured using established techniques. Importantly, the encoded quantum state as well as the operations being performed remain topologically protected, i.e. errors due to a large class of experimental imperfections are exponentially suppressed in system size and the topological gap.

The first challenge in compiling a given quantum circuit for a topological quantum computer based on MZMs is that their topologically protected operations are not by themselves computationally universal [5]: they can only produce multi-qubit Clifford gates, a subgroup of the unitary group that is efficiently simulatable on a classical computer [12]. To perform universal quantum computation, they need to be augmented by one additional non-Clifford gate. A typical choice is the so-called $T$-gate (or $\pi/8$-phase gate), which can be implemented by preparing and injecting a "magic state," which in turn can be prepared to high fidelity using distillation protocols [13]. However, this distillation is very resource-intensive and likely to be the bottleneck of quantum computation using MZMs.[1] It should be noted, however, that surface codes – one of the leading proposals for error correction based on conventional qubits – suffer from the same problem [15]. The set of available computational gates in our envisioned architecture will thus comprise some subset of the topologically-protected Clifford gates (such as all single-qubit operations together with two-qubit operations between all adjacent qubits), augmented by the $T$-gate, which will not be topologically protected. When compiling a given quantum algorithm from this gate set, the primary challenge is to reduce the number of $T$-gates. This problem has been the focus of much attention [16, 17].

In this paper, we focus on a second problem, which is particular to the measurement-only approach: synthesizing the topologically-protected Clifford gates from a sequence of measurements. The previously espoused strategy (see e.g. Ref. [9]) for generating Clifford gates in the measurement-only approach to topological quantum computing with MZMs was to first generate minimal-length measurement sequences for the basic (nearest-neighbor) braiding transformations for each qubit, and a measurement sequence for a two-qubit entangling gate between all pairs of qubits (or at least between all adjacent pairs of qubits), and then use the resulting gate set as the generating gate set used to synthesize any other Clifford gates. From the perspective of the fundamental operators, i.e. measure-

---

[1]For estimates for a relevant problem, see e.g. Ref. [14].

ments, this strategy may be inefficient, as there may exist shorter sequences of measurements that compile to the same gate. We will describe different strategies and protocols for optimizing the generation of computational gates via measurement sequences with the physical measurements themselves as the generating set of operations. We introduce a weighting system for different measurements in a given topological quantum computing architecture that provides a more meaningful metric than number of measurements with respect to which optimization is performed. We provide a demonstration of our methods using brute-force search to find optimal measurement sequence realizations for single-qubit Clifford gates and for two-qubit controlled-Pauli gates. Our methods may also be used to provide a comparative analysis of different strategies and architectures that are being considered for implementation.

Despite the topological protection, a scalable quantum computer built from topological qubits will still require error correction to achieve the desired logical error rates for nontrivial quantum computation. The surface code [18], which is a topological quantum error correction code in the broader class of stabilizer codes, represents one of the most promising proposals for scalable quantum error correction. Generally, stabilizer codes map very favorably onto Majorana-based quantum computers; however, the surface code requires measuring products of four Pauli operators, which can be challenging. Starting from standard techniques to do these measurements using ancillary qubits, we use the methods developed in this paper to propose an optimal measurement sequence that implements the desired operations in an array of Majorana-based qubits.

The outline of the paper is as follows. In Sec. 2, we review the physical architectures for measurement-only topological quantum computing using islands of six MZMs – the so-called "hexon" qubit architectures. We discuss the fact that physically performing different measurements will have different levels of difficulty, and describe a systematic approximation of such. We also discuss the possible advantages of different encodings of the computational and ancillary degrees of freedom in the physical MZMs. In Sec. 3, we describe the "forced-measurement" protocol, and several strategies to improve upon it. In Sec. 4, we describe the Majorana-Pauli tracking method that allows us to circumvent the use of forced-measurement protocols by tracking the measurement outcomes and their effect on the computation. Tracking methods are a more efficient alternative to forced-measurements, but they may only be employed when the measurement outcomes correspond to Abelian anyons. This is always the case for MZM-based architectures, which is the main focus of this paper. In Sec. 5, we discuss optimization and search strategies for measurement-only gates in the various architectures and methods that can be utilized. We provide a demonstration of our methods utilizing brute-force search to find optimizations of measurement sequences for all one-qubit and a subset of all two-qubit Clifford gates, with respect to difficulty weighted measurements. In Sec. 6, we introduce the surface code and explain our implementation using Majorana qubits. In Sec. 7, we discuss the application of our methods beyond the case of MZM-based platforms. Finally, in Appendix A, we provide an example where adaptive methods can improve a force-measurement sequence, in Appendix B, we discuss some strategies to improve measurement sequence efficiency when brute-force search becomes prohibitive, and in Appendix C, we provide explicit details of our demonstration of methods.

## 2 Majorana Hexon Architecture

The specific qubit platforms that we focus on in this paper are the MZM hexon architectures introduced in Ref. [9]. A single hexon is a superconducting island that contains six MZMs,

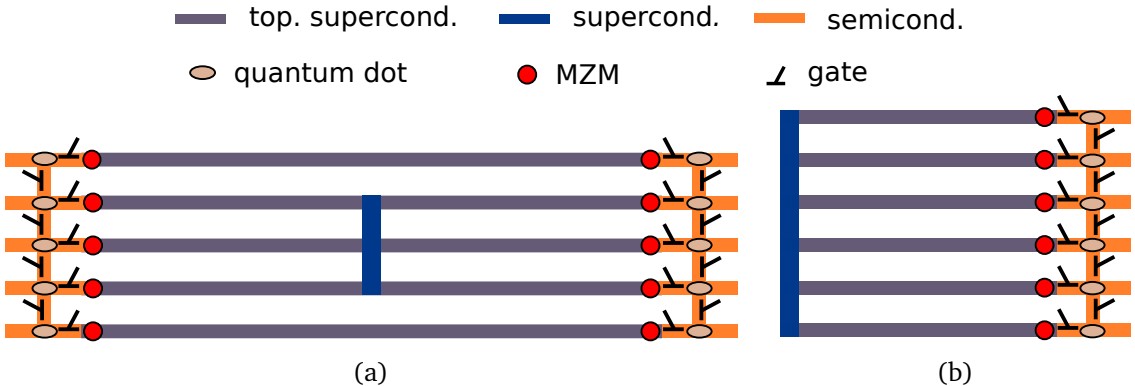

Figure 1: We consider two types of Majorana hexon architectures in detail: (a) the two-sided architecture and (b) the one-sided architecture. Shown here is a single qubit of each architecture with the required semiconducting quantum dots, cutter gates, and superconducting coherent links (top and bottom wire in the two-sided hexon) needed to perform all pairwise MZM measurements. The relative lengths of the vertical and horizontal dimensions are not to scale, and likely to be relatively much longer in the horizontal direction.

where some of these MZMs are used to encode the qubit state and some serve as ancillary degrees of freedom that facilitate measurement-based operations. While a qubit can also be formed from four MZMs (referred to as a tetron when on an isolated superconducting island), due to the absence of ancillary MZMs, such a qubit by itself does not permit any topologically-protected unitary gate operations. A hexon, on the other hand, allows for the full set of single-qubit Clifford gates to be implemented with topological protection. Therefore, we focus on the hexon architecture, though many of the techniques we develop here can be adapted to systems of several tetrons, where some tetrons serve as ancillary qubits.

For MZMs that emerge at the ends of nanowires, a hexon is formed by joining several Majorana nanowires via a spine made from trivial ($s$-wave) superconductor, as shown in Fig. 1. We consider both a two-sided hexon architecture and a one-sided hexon architecture, as shown in Figs. 1(a) and (b), respectively. In the two-sided architecture, three wires are joined by a spine in the middle and MZMs are present at both ends of the wire. In the one-sided architecture, six wires are joined at one of their ends and, thus, MZMs are present only at the other end. An important benefit of these architectures is that a single qubit island is galvanically isolated (except for weak coupling to dots, see below), and thus Coulomb interactions give rise to a finite charging energy $E_C$ for the island. This helps to prevent (extrinsic) quasiparticle poisoning, as the probability for an electron to tunnel onto or off of the island from outside is exponentially suppressed in the ratio of the charging energy $E_C$ to temperature, $\exp(-E_C/k_B T)$. Decoherence of topologically protected states due to thermally excited quasiparticles on the island is suppressed by $\exp(-\Delta/k_B T)$, where $\Delta$ is the topological gap. Degeneracy splitting due to virtual tunneling of fermions between MZMs is suppressed by $\exp(-L/\xi)$, where $L$ is the separation of MZMs and $\xi$ the superconducting coherence length.

Projective measurements of the joint fermionic parity of any two MZMs (2-MZM measurements) can be carried out by enabling weak coherent single-electron tunneling between the MZMs and adjacent quantum dots, forming an interference loop. Projective measurements of the collective fermionic parity of $2N$-MZMs may be performed similarly, though care must be taken to ensure that the interference loop always involves all $2N$

MZMs, e.g. fermions cannot pass directly between the various quantum dots involved. These couplings gives rise to shifts in the energy spectrum and charge occupation of the dot that depend on the fermionic parity of the MZMs. These shifts can, in turn, be measured using established techniques developed for charge and spin qubits, such as charge sensing or quantum capacitance measurements. Importantly, the measurement is topologically protected in the sense that the operator that is being measured is known up to corrections that are exponentially small in the distance separating the MZMs through the superconducting region (nanowire and spine). However, similar to other quantum non-demolition measurements, the measurement fidelity is limited by the achievable signal-to-noise ratio and decoherence of the qubit in other channels. Additionally, the calibration of a signal's correlation to even or odd parity is a choice of convention whose effect on the final outcome of a compilation is a Pauli factor as we elucidate in Sec. 3.3.

While both hexon architectures considered here allow, in principle, 2-MZM measurements between any pair of MZMs, it is clear that, depending on the layout, certain measurements will be more difficult to perform than others. For example, in the two-sided architecture, some measurements are between MZMs on the same side (left or right) of the island, while others are on opposite sides. For measurements involving MZMs that are in close proximity to each other, such as ones that are on the same side of the hexon, one can adjust the electrostatic gates in the semiconducting regions to define a single quantum dot that the MZMs being measured are coupled together. However, when the MZMs are farther separated (e.g. on opposite sides), enabling coherent single-electron tunneling between these MZMs and a common quantum dot is much more challenging, as their distance may exceed the phase coherence length of realistic semiconducting wires. In such cases, a coherent superconducting link can be used to span the distance, but this increases the complexity of the device and the required tuning necessary to perform such a measurement. In Sec. 2.4, we will discuss the matter of measurement difficulty in more detail and provide a model for assigning "difficulty" weights to different measurements, which will be incorporated in our gate synthesis optimization strategies.

Multiple hexons can be arranged into an array, and multi-qubit operations are performed by weakly coupling MZMs from different islands to common quantum dots. Since the coupling between MZMs and quantum dots is weak, the charging energy protection against quasi-particle poisoning remains effective during such operations. This restricts the operators that can be measured to ones that commute with the charging energy (or total parity) on each island, which are precisely the measurements involving an even number of Majorana operators on each island. We will focus on 4-MZM measurements, as measurements involving larger numbers of MZMs appear unrealistic to achieve in practice.

In the multi-hexon arrays shown in Fig. 2, we see that the most realistic 4-MZM measurements involving pairs of hexons give rise to rectangular lattice connectivity graphs of qubits. Even within this rectangular lattice connectivity, certain 4-MZM measurements will be more difficult to perform than others. This can lead to better or worse connectivity between qubits in the four different directions (up, down, left, and right), and may even prevent some 4-MZM measurements from having realistic implementations. In Sec. 2.4, we will illustrate some of the measurements that will be utilized (see Figs. 3 and 4).

## 2.1 Single-hexon state space and operators

We label the positions of the six MZMs in a hexon $1,\ldots,6$, and associate a Majorana fermionic operator $\gamma_j$ to the MZM at the $j$th position. These operators obey the usual fermionic anticommutation relations $\{\gamma_j, \gamma_k\} = 2\delta_{jk}$. For any ordered pair of MZMs $j$ and $k$, their joint fermionic parity operator is given by $i\gamma_j\gamma_k = -i\gamma_k\gamma_j$, which has eigenvalues $p_{jk} = \pm 1$ for even and odd parity, respectively. (The conventions in this paper will differ

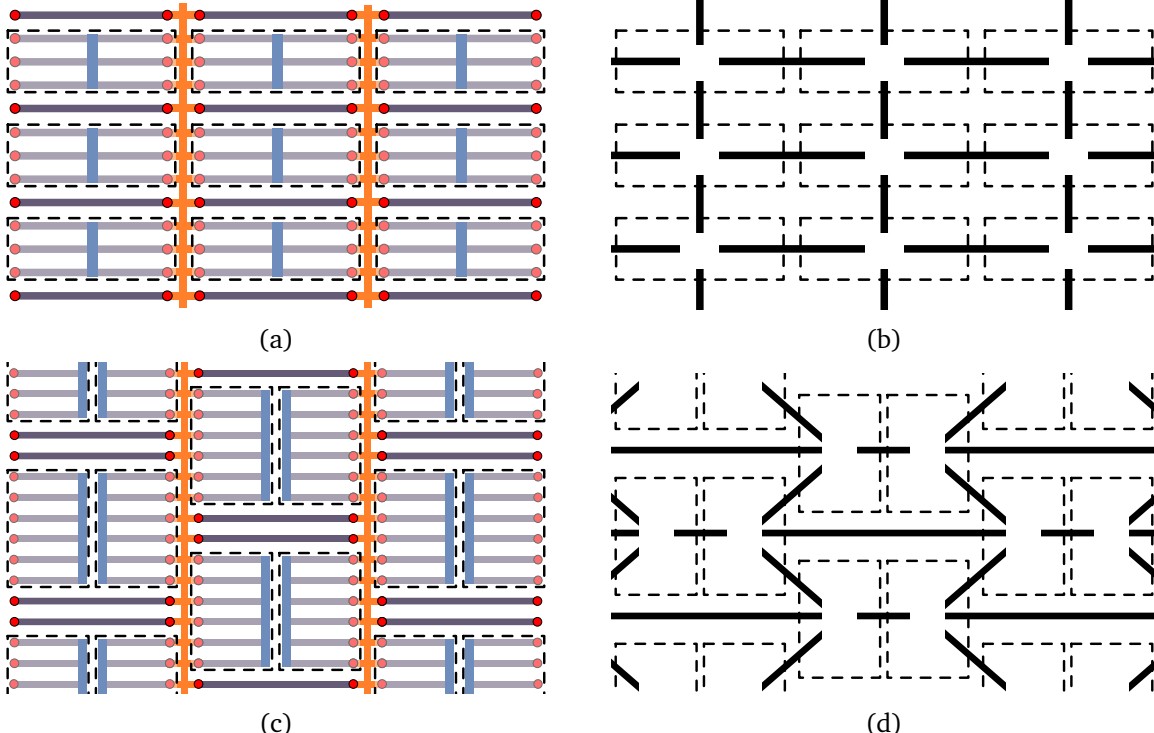

Figure 2: Arrays of hexons, where each hexon is shown enclosed in a dashed-line rectangle. (a) Two-sided hexons can be tiled regularly onto a rectangular lattice. (b) The connectivity of this two-sided hexon array, indicating which pairs of hexons can be acted on by joint 4-MZM measurements, is shown by solid black lines connecting the dashed rectangles. (c) One-sided hexons can be tiled onto a squashed rectangular lattice, with left-facing on one sublattice and right-facing hexons on the other. (d) The connectivity of this one-sided hexon array, indicating which pairs of hexons can be acted on by joint 4-MZM measurements, is shown by solid black lines connecting the dashed rectangles. Examples of measurements that yield the shown connectivity can be found in Figs. 3 and 4. In both architectures, the ability to physically implement two-qubit gates will not be equally difficult in the different directions. For example, utilizing coherent links will generally increase the difficulty.

slightly from those of Ref. [9].) The corresponding projection operator onto the subspace with parity $s = p_{jk} = \pm 1$ is given by

$$\Pi_s^{(jk)} = \Pi_{-s}^{(kj)} = \frac{1}{2}\left(\mathbb{1} + s\, i\gamma_j\gamma_k\right). \tag{1}$$

The operator $i\gamma_j\gamma_k$ can then be expressed as

$$i\gamma_j\gamma_k = \Pi_+^{(jk)} - \Pi_-^{(jk)}, \tag{2}$$

where we use the shorthand $\pm$ for $\pm 1$, the even-parity (vacuum) and odd-parity (fermion) channels, respectively.

In this way, we can write basis states $|p_{12}, p_{34}, p_{56}\rangle$ for a system of six MZMs in terms of the fermionic parities for some choice of how to pair them together. Due to the finite charging energy of the island, the system generically has ground states only in either the even or the odd collective fermion parity sector, which can be tuned using the gate voltage;

without loss of generality, we here assume that the system is tuned to have ground states with even collective fermionic parity, i.e. $p_{12}p_{34}p_{56} = +1$, while states with odd collective parity are excited states associated with quasiparticle poisoning. (The discussion and results for $p_{12}p_{34}p_{56} = -1$ is straightforwardly similar, but we will not focus on it in this paper.) In this way, the low-energy state space of the hexon is 4-dimensional, with basis states

$$|+,+,+\rangle, \tag{3}$$
$$|-,+,-\rangle = i\gamma_2\gamma_5 |+,+,+\rangle, \tag{4}$$
$$|+,-,-\rangle = i\gamma_4\gamma_5 |+,+,+\rangle, \tag{5}$$
$$|-,-,+\rangle = i\gamma_2\gamma_3 |+,+,+\rangle. \tag{6}$$

Viewing this as a two-qubit system with the first qubit encoded in $p_{34}$ and the second qubit encoded in $p_{12}$, the above basis states are $|0,0\rangle, |0,1\rangle, |1,0\rangle, |1,1\rangle$, in order. We can then express the MZM parity operators in terms of Pauli operators on these two qubits as

$$\begin{aligned}
i\gamma_1\gamma_2 &= \mathbb{1} \otimes Z, & i\gamma_1\gamma_3 &= X \otimes Y, & i\gamma_1\gamma_4 &= -Y \otimes Y, & i\gamma_1\gamma_5 &= Z \otimes Y, \\
i\gamma_1\gamma_6 &= \mathbb{1} \otimes X, & i\gamma_2\gamma_3 &= X \otimes X, & i\gamma_2\gamma_4 &= -Y \otimes X, & i\gamma_2\gamma_5 &= Z \otimes X, \\
i\gamma_2\gamma_6 &= -\mathbb{1} \otimes Y, & i\gamma_3\gamma_4 &= Z \otimes \mathbb{1}, & i\gamma_3\gamma_5 &= Y \otimes \mathbb{1}, & i\gamma_3\gamma_6 &= X \otimes Z \\
i\gamma_4\gamma_5 &= X \otimes \mathbb{1}, & i\gamma_4\gamma_6 &= -Y \otimes Z, & i\gamma_5\gamma_6 &= Z \otimes Z,
\end{aligned} \tag{7}$$

where the Pauli matrices are

$$X = \begin{bmatrix} 0 & 1 \\ 1 & 0 \end{bmatrix}, \quad Y = \begin{bmatrix} 0 & -i \\ i & 0 \end{bmatrix}, \quad Z = \begin{bmatrix} 1 & 0 \\ 0 & -1 \end{bmatrix}. \tag{8}$$

We use the convention in which the MZMs 3 and 4 serve as the ancillary MZMs with definite joint parity, e.g. $p_{34} = +1$, and the computational qubit is encoded in $p_{12}$. The remaining parity is correlated with the other two as $p_{56} = p_{12}p_{34}$, so when the ancillary pair has $p_{34} = +1$, the computational basis states are

$$|0\rangle = |p_{12} = p_{56} = +\rangle, \quad |1\rangle = |p_{12} = p_{56} = -\rangle, \tag{9}$$

and when $p_{34} = -1$, the computational basis states are

$$|0\rangle = |p_{12} = -p_{56} = +\rangle, \quad |1\rangle = |p_{12} = -p_{56} = -\rangle. \tag{10}$$

Another way to view this is that a hexon is a Majorana stabilizer code which encodes a single logical qubit in six MZMs [19, 20]. In this language, logical qubits are defined to be in the simultaneous +1 eigenspace of a group of operators, called the stabilizer group. The logical gates which act on this space are operators which commute with the stabilizer group but are not themselves stabilizers. For the case of a hexon, the stabilizer group is generated by the total parity of the island $i^3\gamma_1\gamma_2\gamma_3\gamma_4\gamma_5\gamma_6$ and the parity of the ancillary pair $i\gamma_3\gamma_4$. The logical Pauli operators are taken to be $\bar{Z} = i\gamma_1\gamma_2$ and $\bar{X} = i\gamma_1\gamma_6$. We will initially focus on the case where we require $p_{34} = +1$ for the (initial and final) computational basis, but will allow the ancillary qubit to have either parity in Sec. 4.

We will often make use of a diagrammatic calculus, which allows us to perform algebra in the topological state space by manipulating diagrams, see e.g. Refs [21–23]. In this diagrammatic formalism, isotopy invariance allows us to freely stretch or slide around strands so long as the topology of diagrams remains fixed, i.e. open endpoints of lines are held fixed and trivalent junction do not pass each other when slid along lines. Additional rules for reconnection and braiding of diagrams is incorporated by the so-called $F$-symbols and $R$-symbols.

In the diagrammatic formalism, the projectors can be represented as[2]

$$\Pi_+^{(jk)} = \smile, \quad \Pi_-^{(jk)} = \smile, \quad \Pi_s^{(jk)} = \smile, \tag{11}$$

where $+$ (vacuum) is diagrammatically represented as no line, $-$ (fermion) is represented by a wavy red line, and an unspecified fusion channel $s = \pm 1$ is represented by a magenta line.

The $p_{34} = +1$ computational qubit basis states of hexons are diagrammatically represented as

$$\left| \frac{1-a}{2} \right\rangle = |a, +, a\rangle = \qquad , \tag{12}$$

where $a = \pm 1$. A general computational qubit state $|\Psi\rangle$ will be denoted as

$$|\Psi\rangle := \qquad := \sum_{a=+1,-1} \Psi_a \qquad . \tag{13}$$

Operator multiplication is given by stacking diagrams. The identity operator acting on two MZMs is written as

$$\mathbb{1}_{jk} = \left| \; \right| = \smile + \smile, \tag{14}$$

so the lines are just extended when identity is applied.

The fermionic parity operator $i\gamma_j\gamma_k$ is diagrammatically represented as a fermion line connecting strands $j$ and $k$; it can also be written as an antisymmetric combination of its two projectors (cf. Eq. (2))

$$i\gamma_j\gamma_k = \left| \sim \right| = \smile - \smile. \tag{15}$$

Since $\gamma_j^2 = \mathbb{1}$, a fermion line connecting a single strand to itself with no additional fermion lines connected in between can be freely removed

$$\left| = \right|. \tag{16}$$

From $(i\gamma_j\gamma_k)(i\gamma_k\gamma_l) = -(i\gamma_k\gamma_l)(i\gamma_j\gamma_k) \propto i\gamma_j\gamma_l$ we see that sliding endpoints of fermion lines past one another along a MZM line incurs a minus sign, and also that fermion lines compose

$$\left| \sim \sim \right| = - \left| \sim \sim \right| \propto \left| \sim \sim \right|. \tag{17}$$

$(i\gamma_j\gamma_k)^2 = \mathbb{1}$ is expressed diagrammatically as

$$\left| \sim \sim \right| = \left| \; \right|. \tag{18}$$

Lastly, we pick up a phase of $i$ when we flip the side of a MZM line to which a fermion attaches

$$\left| \sim \right| = i \left| \sim \right|. \tag{19}$$

---

[2]In this paper, we use the diagrammatic normalizations such that a closed loop of either fermion line or MZM line evaluates to 1. Consequently, straightening out bends in the MZM lines will yield nontrivial constant factors, but these will always result in overall constants that can be neglected in the context where they occur in this paper.

## 2.2 Single-qubit gates through measurements

Single-qubit Clifford gates can be implemented on the encoded qubit in a topologically protected manner via a "measurement-only" braiding protocol [10]. The braiding transformations are represented in term of Majorana operators as

$$R^{(jk)} = \frac{1}{\sqrt{2}}(\mathbb{1} + \gamma_j \gamma_k) = \quad , \tag{20}$$

for the counterclockwise exchange of MZMs at positions $j$ and $k$. Using the measurement-only protocol, the single-qubit braiding gates are realized by sequentially measuring the joint fermionic parity of MZM pairs, subject to the following constraints: (1) the first measurement must involve exactly one MZM from the ancillary pair, (2) subsequent measurements must involve exactly one MZM from the preceding measured pair, and (3) the final measurement must involve the (original) ancillary pair and the measurement outcome must equal the ancillary pair's initial joint parity, which (for now) is taken to be $p_{34} = +$. As such, sequential measurements will correspond to anticommuting parity operators, i.e. measurements of pairs $(jk)$ and $(lm)$ are allowed to follow one another if and only if $i\gamma_j\gamma_k i\gamma_l\gamma_m = -i\gamma_l\gamma_m i\gamma_j\gamma_k$. These conditions ensure that the measurements do not read-out any information about the state of the encoded computational qubit. Another way of viewing this process is that one is performing a sequence of anyonic teleportations [10], where, in each step, the encoded qubit state is being re-encoded in a different set of MZMs and the measured pair of MZMs temporarily becomes the ancillary pair. In this view, the sequence of teleportations defines the braiding "path" and enacts the corresponding braiding transformation on the encoded state.

In order to ensure that the final measurement outcome of the ancillary pair is the same as its initial value, and to deterministically control which computational gate is produced by such a process involving measurements, one may use a "forced-measurement" protocol for each measurement step [10]. This is a repeat-until-success procedure involving the ancillary degrees of freedom that allows one to end with a desired measurement outcome. In other words, a forced-measurement of $i\gamma_j\gamma_k$ onto a specific fusion channel $s$ effectively acts on the state space as the projector $\Pi_s^{(jk)}$. In this protocol, if we get an undesired result, we can perform a different measurement that effectively "resets" the state of the system to allow for the target measurement to be performed again with a new probability of obtaining the desired outcome (see Sec. 3 for more details). As the measurement outcomes involved in this process should (ideally) have equal probability of both outcomes, the probability of needing more than some number of attempts to succeed is exponentially suppressed in the number of attempts. The average number of attempts needed to achieve the desired outcome is 2. Thus, while probabilistically determined, the number of measurements needed for a given forced measurement can be treated as a constant, on average.

In Sec. 3, we will describe the forced-measurement procedure in more detail, as well as a refinement of the strategy of applying forced measurements at each measurement step. However, since we have this repeat-until-success method of effectively producing a desired measurement outcome (via a sequence of physical measurements) at each step, we will initially discuss the measurement-only gate synthesis in terms of the projectors corresponding to the desired measurement outcomes, rather than the full sequence of physical measurements involved.

A sequence of projectors on a hexon subject to the above constraints generates a single-

qubit Clifford gate acting on the encoded computational qubit. For example,

$$\Pi_+^{(34)}\Pi_+^{(13)}\Pi_+^{(23)}\Pi_+^{(34)} \propto \begin{bmatrix} 1 & 0 \\ 0 & 0 \end{bmatrix} \otimes \begin{bmatrix} 1 & 0 \\ 0 & i \end{bmatrix} = \Pi_+^{(34)} \otimes S, \tag{21}$$

where $S$ is the $\pi/4$-phase gate. (Here, we have included an initial $\Pi_+^{(34)}$, which is redundant when assuming the ancillary MZMs are properly initialized, but which is convenient for evaluating the operator the sequence will effect.) This relation can be checked algebraically in terms of the Majorana operators by expanding each projector. Similarly, the gate $B = S^\dagger H S^\dagger$ (where $H$ is the Hadamard gate) acting on the qubit can be produced from the projector sequence

$$\Pi_+^{(34)}\Pi_-^{(35)}\Pi_-^{(23)}\Pi_+^{(34)} \propto \begin{bmatrix} 1 & 0 \\ 0 & 0 \end{bmatrix} \otimes \frac{1}{\sqrt{2}}\begin{bmatrix} 1 & -i \\ -i & 1 \end{bmatrix} = \Pi_+^{(34)} \otimes B. \tag{22}$$

We note that the gate set $\{S, B\}$ generates all single-qubit Clifford gates $C_1$.

The Clifford gates can be directly related to the braiding transformations, for example $S = R^{(12)}$ and $B = R^{(25)}$. These relations can be made more visually transparent by viewing the projector sequences applied to the hexon in the diagrammatic representation

$$\Pi_+^{(34)}\Pi_+^{(13)}\Pi_+^{(23)}|\Psi\rangle = \quad \vcenter{\hbox{[diagram]}} \tag{23}$$

and

$$\Pi_+^{(34)}\Pi_+^{(35)}\Pi_+^{(23)}|\Psi\rangle = \quad \vcenter{\hbox{[diagram]}}. \tag{24}$$

Here, the ancillary MZM pair is explicitly initialized to $p_{34} = +$, so the redundant initial projector $\Pi_+^{(34)}$ is not included.

## 2.3 Multi-hexon operations

The Hilbert space of two hexon units is the tensor product of that of the two individual hexons. We label the first hexon's MZMs $1, \ldots, 6$ and the second hexon's MZMs by $1', \ldots, 6'$, and ascribe Majorana operators to them, accordingly. In this way, we have the two-hexon qubit basis states

$$|a, b\rangle = |a\rangle \otimes |b\rangle = |p_{12} = a, p_{34} = +, p_{56} = a\rangle \otimes |p_{1'2'} = b, p_{3'4'} = +, p_{5'6'} = b\rangle. \tag{25}$$

In order to generate entangling two-qubit gates, we need to include measurements of the collective fermionic parity of four MZMs, two (labeled $j$ and $k$) from the first hexon and

two (labeled $l'$ and $m'$) from the second hexon. We write the 4-MZM joint parity projector as

$$\Pi_s^{(jk;l'm')} = \frac{1}{2}\left(\mathbb{1} - s\,\gamma_j\gamma_k\gamma_{l'}\gamma_{m'}\right) = \Pi_+^{(jk)}\Pi_s^{(l'm')} + \Pi_-^{(jk)}\Pi_{-s}^{(l'm')}, \tag{26}$$

where we use semicolons to separate labels corresponding to different hexons. We re-emphasize that the order of MZM labels matters, since the Majorana operators anti-commute. We also emphasize that these projectors will not change the total fermionic parity of either hexon island. Diagrammatically, these projectors can be represented as

$$\Pi_+^{(jk;l'm')} = \qquad \qquad + \qquad \qquad \tag{27a}$$

$$\Pi_-^{(jk;l'm')} = \qquad \qquad + \qquad \qquad , \tag{27b}$$

where the first projector in each term acts on the MZMs at positions $j$ and $k$ of the first hexon, while the second projector acts on MZMs $l'$ and $m'$ of the second hexon.

Two-qubit gates can similarly be generated from sequences of 2-MZM and 4-MZM projection operators. Since the particle number on each island should be preserved, all measurement operators need to involve an even number of Majorana operators on each island. In the case of two-qubit operations, the measurement sequences must also begin and end with both ancillary pairs in their initialized state. In other words, the sequences of projectors begin and end with $\mathbf{\Pi}_+^{(\mathrm{anc})} = \Pi_+^{(34)}\Pi_+^{(3'4')}$. However, if either $\Pi_+^{(34)}$ or $\Pi_+^{(3'4')}$ commutes with every term in the measurement sequence, then the final measurement of the ancillary pairs does not need to involve the corresponding pair of MZMs, since they will already be in the desired final ancillary state.

More generally, a system of $N$ hexons encodes $N$ computational qubits and $N$ ancillary qubits. When we specify an ordered set $\mathcal{M}$ of $2r$ MZMs, we define the corresponding fermionic parity and projection operator to be

$$\Gamma_{\mathcal{M}} = i^r \prod_{a\in\mathcal{M}} \gamma_a, \tag{28}$$

$$\Pi_s^{(\mathcal{M})} = \frac{1}{2}\left(\mathbb{1} + s\Gamma_{\mathcal{M}}\right), \tag{29}$$

where the order of Majorana operators in the product respects the order of the set. Multi-hexon measurements only ever need to involve two MZMs from each hexon involved, since the overall fermionic parity of each hexon island is fixed, giving the relation $i^3\gamma_1\gamma_2\gamma_3\gamma_4\gamma_5\gamma_6 = +1$ on the ground state space. This allows the product of four of the MZMs from a hexon to be replaced by the product of the other two (with appropriate phase factors).

The general condition for a sequence of fermionic parity measurements involving $N$ hexons to compile to a unitary gate acting on the computational qubits is that the measurements (which range from 2-MZM to $2N$-MZM measurements) should not read information out of the computational state, i.e. the corresponding projectors should not reduce the rank of any encoded computational state. Any subsequence of the projector sequence must therefore *not* multiply out to an operator of rank less than $2^N$. Additionally, the final measurement in a sequence must project the ancillary MZMs into the initialized state.

In order to translate this general condition into more explicit constraints on the allowed measurements, it is helpful for the case of MZMs to utilize the stabilizer formalism, as may be adapted from Ref. [12]. (This, of course, also works for the single-qubit measurement-only gates, but is overkill for that case.) In this picture, we view the system of $N$ hexons as a Majorana stabilizer code that encodes $N$ logical qubits in $6N$ MZMs. Each hexon island

has a fixed total parity throughout the measurement-only sequence, which translates into the fixed stabilizer $i^3 \gamma_1 \gamma_2 \gamma_3 \gamma_4 \gamma_5 \gamma_6$. Each hexon island initially has an additional ancillary qubit stabilizer corresponding to the parity operator $i \gamma_3 \gamma_4$. Thus, these are the generators of the initial stabilizer group of a hexon $S_0 = \langle i^3 \gamma_1 \gamma_2 \gamma_3 \gamma_4 \gamma_5 \gamma_6, i \gamma_3 \gamma_4 \rangle$, which is isomorphic to $\mathbb{Z}_2 \times \mathbb{Z}_2$. The corresponding logical Pauli operators (acting on the logical qubit) for a hexon island are $\bar{X} = [i \gamma_1 \gamma_6]$, $\bar{Y} = [-i \gamma_2 \gamma_6]$, and $\bar{Z} = [i \gamma_1 \gamma_2]$, where the equivalence classes contain all parity operators related by multiplication by a stabilizer, that is $[\Gamma_{\mathcal{M}}] = \{\Gamma_{\mathcal{N}}, \mid \exists Q \in S : \Gamma_{\mathcal{N}} = Q \Gamma_{\mathcal{M}}\}$. The initial stabilizer group and operators for the $N$ hexon system is obtained by taking products of each hexon's stabilizer group and operators, i.e. $S = \prod_{\alpha=1}^{N} S_0^{(\alpha)} \cong \mathbb{Z}_2^{2N}$, so there are $4^N$ stabilizers.

In order for a measurement to neither act trivially on nor read information out of the encoded logical state, the operator being measured must not commute with all of the stabilizers. Since the measured parity operator $\Gamma_{\mathcal{M}}$ and the stabilizers are all products of Majorana operators, this means $\Gamma_{\mathcal{M}}$ must commute with exactly half of the stabilizers and anticommute with the other half. After performing such a measurement, the stabilizer group and logical operators must be updated. The updated stabilizer group is obtained by removing all of the stabilizers that anticommute with $\Gamma_{\mathcal{M}}$, and then adding $\Gamma_{\mathcal{M}}$ as a new stabilizer and using them to generate the new stabilizer group. We can write this in terms of the following steps [12]:

1. Write $S = S_C \cup S_A$, where $S_C$ is the subgroup of stabilizers that commute with $\Gamma_{\mathcal{M}}$ and $S_A$ is the set of stabilizers that anticommute with $\Gamma_{\mathcal{M}}$.

2. Update the stabilizer group to: $S' = S_C \times \langle \Gamma_{\mathcal{M}} \rangle$.

3. Write each logical Pauli operator $\bar{P}$ as $\bar{P} = \bar{P}_C \cup \bar{P}_A$, where $\bar{P}_C$ is the subset of parity operators in the equivalence class that commute with $\Gamma_{\mathcal{M}}$ and $\bar{P}_A$ is the subset of parity operators in the equivalence class that anticommute with $\Gamma_{\mathcal{M}}$.

4. Update each logical Pauli operator to: $\bar{P}' = \bar{P}_C \cup \bar{P}_C \Gamma_{\mathcal{M}} = [P_C]'$, for any $P_C \in \bar{P}_C$, where $[\cdot]'$ is the equivalence class under multiplication by the updated stabilizer $S'$.

In this way, each step in the measurement-only sequence may be viewed as a deformation of the Majorana stabilizer code (updating the stabilizer group and logical operators) of the $N$ hexon system [24].

For computational purposes, it is typically more convenient to work with a minimal set of generators of the stabilizer group and a single representative of the logical operators. Let $J$ be a minimal set of generators of the stabilizer, i.e. $\langle J \rangle = S$ and $|J| = 2N$. Let $P \in \bar{P}$ be a representative element of the logical Pauli operator. These objects are updated after measuring $\Gamma_{\mathcal{M}}$ according to the following steps:

1. Identify all elements $A_1, \ldots, A_n \in J$ that anticommute with $\Gamma_{\mathcal{M}}$.

2. Update the generating set the stabilizer group to:
   $J' = J \cup \{\Gamma_{\mathcal{M}}, A_1 A_2, \ldots, A_1 A_n\} \setminus \{A_1, \ldots, A_n\}$.

3. Update the representative element of each logical Pauli operator to: $P' = P$ if $P$ commutes with $\Gamma_{\mathcal{M}}$, or to $P' = A_1 P$ if $P$ anticommutes with $\Gamma_{\mathcal{M}}$.

It should be clear that $S' = \langle J' \rangle$, $|J'| = |J|$, and $P' \in \bar{P}' = [P']'$. We emphasize that the labeling order of the elements $A_1, \ldots, A_n$ is arbitrary and the choice of $A_1$ is not special.

For a measurement-only sequence applied to a single hexon, each measurement step may select from 8 possible pairs of MZMs to measure. For two hexons, there are 16 possible 2-MZM measurements and 176 4-MZM measurements that are allowed to select from at

each step. If a sequence of measurements ends with the final stabilizer group equal to the initial stabilizer group, then the sequence yields a logical gate acting on the original logical state space, which is determined by the transformation of the logical Pauli operators.

We will use $\mathcal{G}$ to denote a specific sequence of projectors, corresponding to a specific sequence of measurements and outcomes (or forced measurements), used to generate a gate with a measurement-only protocol, as

$$\mathcal{G} = \mathbf{\Pi}_+^{(\text{anc})} \Pi_{s_{n-1}}^{(\mathcal{M}_{n-1})} \dots \Pi_{s_1}^{(\mathcal{M}_1)} \mathbf{\Pi}_+^{(\text{anc})}, \tag{30}$$

where the labels $\mathcal{M}_\mu$ are used to denote an allowed ordered set of (an even number of) MZMs whose joint fermionic parity is being projected onto corresponding parity $s_\mu$ at the $\mu$th projector in the sequence. The ancillary projector gives the projection of all involved hexons' ancillary pair of MZMs into the $+$ state, that is

$$\mathbf{\Pi}_+^{(\text{anc})} = \Pi_+^{(34)} \otimes \dots \otimes \Pi_+^{(3'\dots'4'\dots')}. \tag{31}$$

The resulting unitary gate acting on the encoded computational state space will be written as $G$, where

$$\mathcal{G} \propto \Pi_+^{(\text{anc})} \otimes G. \tag{32}$$

We emphasize that the relation between projection operator sequences and computational gates is many-to-one.

An example of a two-qubit entangling gate generated from 2-MZM and 4-MZM projectors is

$$W = \begin{bmatrix} 1 & 0 & 0 & 0 \\ 0 & i & 0 & 0 \\ 0 & 0 & i & 0 \\ 0 & 0 & 0 & 1 \end{bmatrix}, \tag{33}$$

which can be obtained from the sequence of projectors, as in Ref. [9]:

$$\Pi_+^{(34)} \Pi_{s_3}^{(35)} \Pi_{s_2}^{(56;1'2')} \Pi_{s_1}^{(45)} \mathbf{\Pi}_+^{(\text{anc})} \propto \mathbf{\Pi}_+^{(\text{anc})} \otimes W^{-s_1 s_2 s_3}, \tag{34}$$

where either $W$ ($-s_1 s_2 s_3 = +1$) or its inverse ($-s_1 s_2 s_3 = -1$) is obtained, depending on the measurement outcomes. The first term in the tensor product acts on the ancillary qubits and the second acts on the computational qubits. Note that $\Pi_+^{(3'4')}$ commutes with every term above, so the final projector only needs to act on MZMs 3 and 4. For example, diagrammatically,

$$W |\Psi_1, \Psi_2\rangle \propto \Pi_+^{(34)} \Pi_-^{(35)} \Pi_+^{(56;1'2')} \Pi_+^{(45)} |\Psi_1, \Psi_2\rangle$$

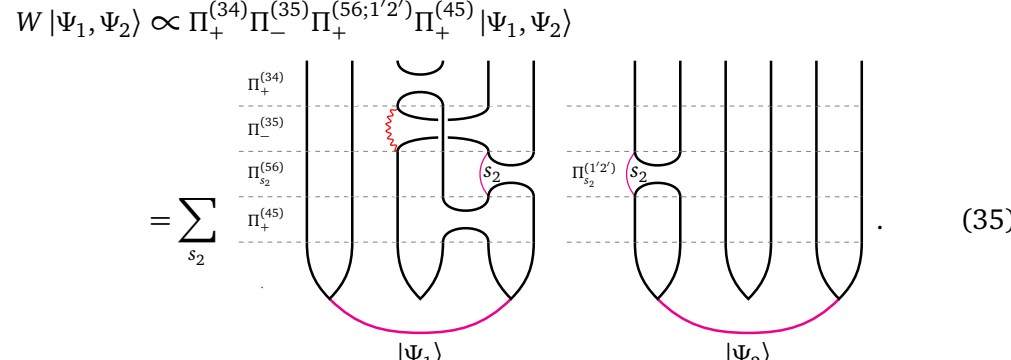

$$\tag{35}$$

To see this relation, we first use the fact that $\Pi_+^{(3'4')}$ commutes with every other projector in the sequence, and hence can be factored out and ignored for this calculation. We expand

$$
\begin{aligned}
\Pi_+^{(34)}\Pi_{s_3}^{(35)}\Pi_{s_2}^{(56;1'2')}\Pi_{s_1}^{(45)}\Pi_+^{(34)} &= \Pi_+^{(34)}\frac{\mathbb{1}+is_3\gamma_3\gamma_5}{2}\frac{\mathbb{1}-s_2\gamma_5\gamma_6\gamma_{1'}\gamma_{2'}}{2}\frac{\mathbb{1}+is_1\gamma_4\gamma_5}{2}\Pi_+^{(34)}\\
&= 2^{-3}\Pi_+^{(34)}\big(\mathbb{1}+is_1\gamma_4\gamma_5+is_3\gamma_3\gamma_5-s_2\gamma_5\gamma_6\gamma_{1'}\gamma_{2'}+s_1s_3\gamma_3\gamma_4\\
&\qquad +is_1s_2\gamma_4\gamma_6\gamma_{1'}\gamma_{2'}-is_2s_3\gamma_3\gamma_6\gamma_{1'}\gamma_{2'}+s_1s_2s_3\gamma_3\gamma_4\gamma_5\gamma_6\gamma_{1'}\gamma_{2'}\big)\Pi_+^{(34)}\\
&= 2^{-3}\Pi_+^{(34)}\big(\mathbb{1}-is_1s_3\mathbb{1}-s_2\gamma_5\gamma_6\gamma_{1'}\gamma_{2'}-is_1s_2s_3\gamma_5\gamma_6\gamma_{1'}\gamma_{2'}\big)\Pi_+^{(34)}\\
&= 2^{-5/2}\Pi_+^{(34)}e^{-i\frac{\pi}{4}s_1s_3}\big(\mathbb{1}-is_1s_2s_3\gamma_5\gamma_6\gamma_{1'}\gamma_{2'}\big)\Pi_+^{(34)}\\
&= 2^{-2}e^{i\frac{\pi}{4}s_1s_3(s_2-1)}\Pi_+^{(34)}\otimes W^{-s_1s_2s_3}.
\end{aligned}
\tag{36}
$$

Here, we used the facts that $\Pi_+^{(34)}\gamma_3\gamma_j = \gamma_3\gamma_j\Pi_-^{(34)}$ and $\Pi_+^{(34)}\gamma_4\gamma_j = \gamma_4\gamma_j\Pi_-^{(34)}$ for $j\neq 3,4$, that $\Pi_+^{(34)}$ projects onto the subspace with $\gamma_3\gamma_4 = -i$, and $(i\gamma_5\gamma_6)(i\gamma_{1'}\gamma_{2'}) = Z\otimes Z\otimes\mathbb{1}\otimes Z$. We note that, using the methods of our paper, we can find more efficient projector sequences for this gate, such as

$$
\Pi_+^{(34)}\Pi_{s_2}^{(35)}\Pi_{s_1}^{(36;1'2')}\mathbf{\Pi}_+^{(\mathrm{anc})} \propto \mathbf{\Pi}_+^{(\mathrm{anc})}\otimes W^{s_1s_2}.
\tag{37}
$$

The gate set $\{S,B,W\}$, where the single-qubit gates can act on any qubit and the two-qubit gates can act on any (nearest-neighbor) pair of qubits, generates all $N$-qubit Clifford gates $\mathsf{C}_N$. For instance, the controlled-$Z$ gate can be obtained as $\mathsf{C}(Z) = (S^\dagger\otimes S^\dagger)W$, and $\mathsf{C}(X)$ can be obtained from $\mathsf{C}(Z)$ by conjugating the target qubit by $H = SBS$. It is well-known that $\{S,H,\mathsf{C}(Z)\}$ generates the entire set of $N$-qubit Clifford gates for any $N$, so $\{S,B,W\}$ does as well.

## 2.4 Not all measurements are created equal

Experimentally, certain measurements will be more difficult to perform than others. For example, measurements on nearby MZMs can be expected to be less faulty and require less resources than measurements involving distant MZMs. We can account for this by using a cost function that assigns "difficulty" weights to the specific measurement operations that are utilized throughout a computation. In this way, a sequence of measurements, used e.g. to generate computational gates, will have a corresponding difficulty weight.

We use the ambiguous term "difficulty" primarily as a stand-in for error-rate, but also to encapsulate resource requirements and other complexities, until a more accurate picture of these matters is obtained through physical experiments. We will provide extremely rough, but systematic and physically motivated estimates of the difficulty weights for the measurements, to provide quantitative demonstrations of our methodology.

*Cutter gates* — In the hexon architecture, measurements are performed by coupling different MZMs to quantum dots, which effectively form interference loops delineated by the paths connecting the MZMs through the hexon and the paths connecting MZMs through the dots, as shown in Figs. 3 and 4. To select the interference paths, electrostatic depletion gates are tuned which effectively connect or disconnect different parts of the semiconductor, and define quantum dots in it. We will refer to these gates as *cutter gates*. These cutter gates affect the measurement difficulty in two ways: (i) It appears likely that disorder in the region where the cutters are deposited will locally decrease the phase coherence of the semiconductor, and thus reduce the visibility of the measurement. (ii) The overall length of the semiconducting path will affect phase coherence, and its volume affects properties of the dot such as its charging energy and level spacing. In general, the measurement

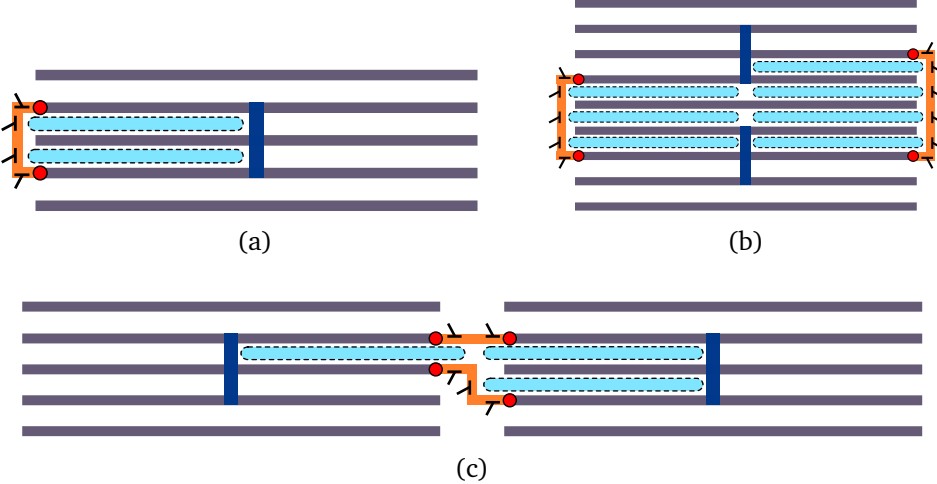

Figure 3: Various fermionic parity measurement configurations for the two-sided hexon architecture. (a) A 2-MZM measurement with $n_c = 2$ vertical cutter gates opened, $n_a = 2$ units of area enclosed by the interference loop, and $n_t = 2$ tunneling junctions to MZMs. (b) A 4-MZM measurement on vertically displaced hexons, with $n_c = 7, n_a = 7$, and $n_t = 4$. (c) A 4-MZM measurement on horizontally displaced hexons, with $n_c = 1$, $n_a = 3$, and $n_t = 4$.

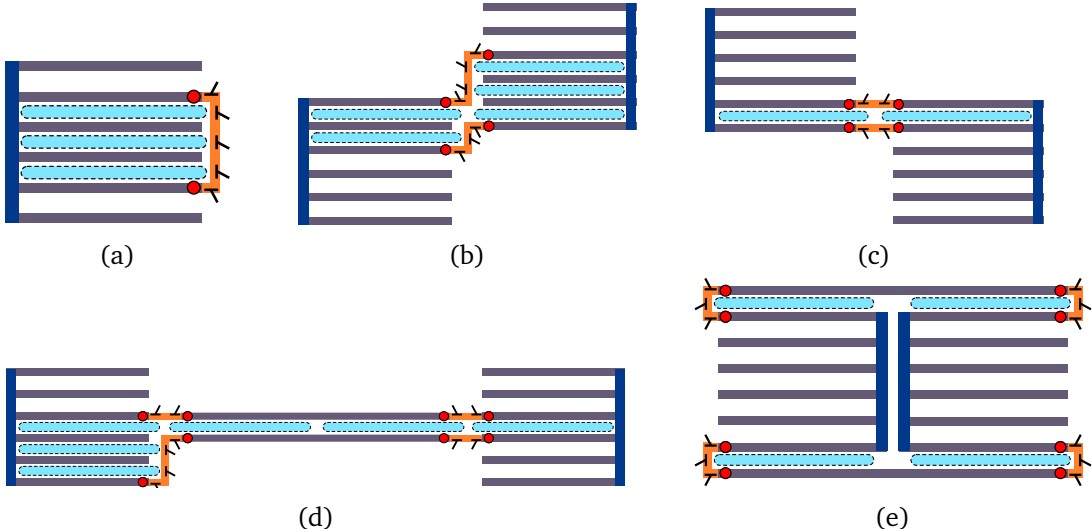

Figure 4: Various fermionic parity measurement configurations for the one-sided hexon architecture. (a) A 2-MZM measurement with $n_c = 3$ vertical cutter gates opened, $n_a = 3$ units of area enclosed by the interference loop, and $n_t = 2$ tunneling junctions to MZMs. (b) A 4-MZM measurement in the upward direction, with $n_c = 3, n_a = 5$, and $n_t = 4$. (c) A 4-MZM measurement in the downward direction, with $n_c = 0$, $n_a = 2$, and $n_t = 4$. (d) A 4-MZM measurement in the rightward direction, with $n_c = 2, n_a = 6$, and $n_t = 8$. (e) A 4-MZM measurement in the leftward direction, with $n_c = 4$, $n_a = 4$, and $n_t = 8$.

will be easier for smaller dots. We use the number of *vertical* cutter gates involved in a measurement as simple placeholder for the length of the semiconducting region.

*Tunnel junctions* — Wherever a MZM couples to the semiconductor, the coupling must

be carefully tuned by a depletion gate forming a tunnel junction. In contrast to cutter gates between semiconducting regions, which will generally be either fully opened or closed, it is important to tune the coupling to MZMs carefully such that its ratio with the charging energy $E_C$ is in a favorable regime where the effect on the quantum dot is quickly and reliably measurable, while not suppressing the charging energy of the dot and increasing the probability of quasiparticle poisoning. Realistically, the visibility of the signal will be reduced with each tunnel junction, and noise in the tunnel gate can affect the measurement signal. Furthermore, as part of the measurement protocol, this coupling must be tuned from 0 to its target value on a time-scale that is at the same time fast compared to the measurement time and slow enough to avoid inducing diabatic corrections; this must be achieved even in the presence of non-monotonic pinch-off curves due to bound states near the gate. Finally, note that MZMs that are far away from each other can be connected using superconducting coherent links, themselves made from topological superconductors and requiring additional tunnel junctions. The number of tunneling junctions is equal to the number of MZMs involved in a measurement, which may be larger than the number of MZMs being measured if using superconducting coherent links. It is also equal to the number of horizontal cutter gates.

*Flux noise* — The energy shift of the quantum dot depends on the magnetic flux enclosed in the loop. Noise in the enclosed flux, either from noise in the background field or any flux lines used to tune local fields, will make the measurement more challenging. As the flux noise will depend on the enclosed area, we account for this area, assuming that the geometries are such that the relevant areas for such errors are approximately partitioned into integer multiples of some unit area.

*Number of islands* — The difficulty of a measurement will also depend on the number $N$ of hexons involved. This is because the measurement visibility will be significantly affected by how well the system can be tuned to the resonant tunneling point, and also because the operations utilized in a measurement can cause errors that transfer fermions between the different hexons.

Given the factors described above, we define the difficulty weight of a fermionic parity measurement of $2N$-MZMs $\mathcal{M} = (jk; l'm'; \dots)$ involving $N$ hexons to be

$$w(\mathcal{M}) = w_c^{n_c(\mathcal{M})} w_t^{n_t(\mathcal{M})} w_a^{n_a(\mathcal{M})} f(N), \tag{38}$$

where $n_c$ is the number of vertical cutter gates that are opened for the measurement, $n_t$ is the number of tunneling junctions involved in the measurement, which is equal to the number of MZMs involved in the measurement (including those of coherent links), and $n_a$ is the (integer) amount of unit area enclosed by the interferometry loop delineated by the measurement. The quantities $w_c$, $w_t$, and $w_a$ are the difficulty weights associated with the corresponding factors described above. (The weights $w_t$ associated with the tunneling junctions will also include the contribution from horizontal cutter gates, since these are used to control the tunneling in a manner somewhat different from the way the vertical cutter gates are used to define the quantum dots.) The difficulty associated with the number of hexons involved in the measurement is likely a more complicated (though quickly growing) function of $N$ that we denote as $f(N)$. All of these quantities must be determined by the experimental setup being utilized.

## 2.5 Relabeling Majorana zero modes

In our discussion thus far, we have labeled the MZMs in a hexon $1, \dots, 6$ and assigned them particular roles according to these labels. For example, in the computational basis, the MZMs labeled as 3 and 4 serve as the ancillary pair, while MZMs 1, 2, 5, and 6 collectively

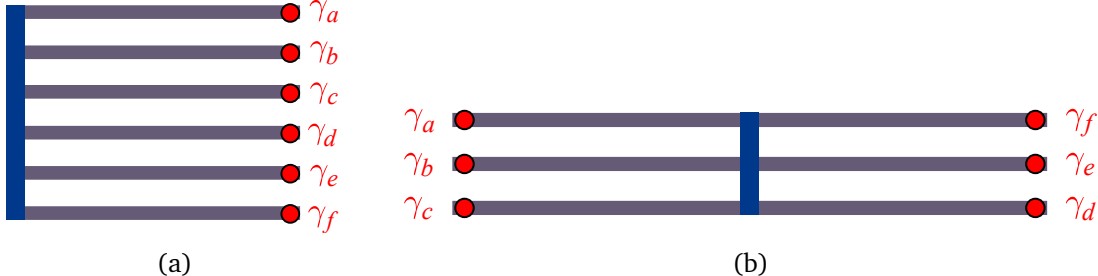

Figure 5: A labeling configuration $\langle a,b,c,d,e,f \rangle$ of MZMs shown for (a) one-sided hexons, which follow the labeling order from top to bottom, and (b) two-sided hexons, which follow the labeling order counterclockwise from top-left to top-right.

encode the computational qubit. However, we are free to choose how the six labels are assigned to the physical MZMs of a hexon. We will now discuss briefly how this choice can affect the difficulty of different measurements, and hence measurement-only gate synthesis.

Let $\langle a,b,c,d,e,f \rangle$ denote the configuration of MZMs within a hexon, where (i) for one-sided hexons, the labeling goes from top to bottom, and (ii) for two-sided hexons, the labeling goes counterclockwise from the top-left to the top-right. A possible configuration for either hexon architecture, which was used in Ref. [9], is $\langle 1,2,3,4,5,6 \rangle$. Here, MZMs 1 and 6 are on opposite ends of the hexon. On the other hand, in the configuration $\langle 1,6,2,3,4,5 \rangle$, these two MZMs are adjacent. In this way, different configurations of MZMs will result in different assignments of difficulty weights to a measurement. For example, a measurement of MZMs (16) will have weights $w(16)_{\langle 1,6,2,3,4,5 \rangle} < w(16)_{\langle 1,2,3,4,5,6 \rangle}$ for these two configurations. Thus, if this measurement occurs very frequently in a computation, the configuration $\langle 1,6,2,3,4,5 \rangle$ may be advantageous. We will take this into account when we numerically optimize measurement sequences in Sec. 5 and discuss examples of optimal configurations of the labels under certain assumptions about the weights.

Note that there are certain symmetry relations for each architecture, which reduce the number of inequivalent configurations that must be considered. A two-sided hexon has horizontal and vertical reflection symmetry, reducing the number of inequivalent configurations from $6! = 720$ to 180. One-sided hexons have horizontal reflection symmetry, so the number of configurations that we consider is reduced from 720 to 360.

In order for the gate generation methods to be scalable, the full array of hexons in the system should utilize labeling configurations that are periodic in the array. In this paper, we consider the simplest case, where each hexon in the array uses the same labeling configuration. However, one could imagine finding benefits from assigning different configurations to different hexons, e.g. one configuration for all right-facing one-sided hexons and a different configuration for all left-facing one-sided hexons.

Depending on the architecture and labeling configuration used, the different 4-MZM measurements can have significantly different difficulty weights. Moreover, the measurements involving hexon pairs that are neighbors in different directions may have different difficulty levels. For example, in the case of one-side hexon arrays, the measurements connecting vertical neighbors shown in Fig. 4(b) and (c) will generally be less difficult than those connecting horizontal neighbors shown in Fig. 4(d) and (e). However, the geometry can make certain 4-MZM measurements essentially impossible (or prohibitively difficult). For example, the measurements involving vertical neighbors must always involve the top-

most MZM of one hexon and the bottom-most MZM of the other hexon.

# 3  Forced-Measurement Methods

In the measurement-only approach to topological quantum computation, the desired sequences of projection operators that yield computational gates are physically generated by performing measurements on the system. When the joint fermionic parity operator $\Gamma_{\mathcal{M}}$ of an ordered set $\mathcal{M}$ of MZMs is measured in a system in a pure state $|\Psi\rangle$, the measurement outcome $s = \pm$ will be obtained with probability $p_s = \langle\Psi|\Pi_s^{(\mathcal{M})}|\Psi\rangle$, and one obtains the corresponding post-measurement state

$$|\Psi\rangle \mapsto \frac{1}{\sqrt{p_s}}\Pi_s^{(\mathcal{M})}|\Psi\rangle. \tag{39}$$

For general states described by a density matrix $\rho$, the measurement outcome $s$ is obtained with probability $p_s = \text{Tr}\left[\Pi_s^{(\mathcal{M})}\rho\right]$, and the post-measurement state is

$$\rho \mapsto \frac{1}{p_s}\Pi_s^{(\mathcal{M})}\rho\,\Pi_s^{(\mathcal{M})}. \tag{40}$$

The probabilistic nature of measurements can be dealt with in the measurement-only approach (where ancillary degrees of freedom are being utilized) by using forced-measurement protocols. When the outcome of a measurement in a measurement-only sequence is a non-Abelian anyon, the use of a forced-measurement protocol is necessary. On the other hand, when the measurement outcomes is an Abelian anyon (different from the "desired" measurement outcome at a given step), then one can use tracking methods as a more efficient alternative, as will be described in Sec. 4. In the case of MZMs, one can always use tracking methods instead of forced-measurements. However, we will nonetheless use the example of MZMs to discuss forced-measurement methods in this section, since the basic ideas carry over to more general non-Abelian anyons, with straightforward modifications.

## 3.1  Forced-measurement protocols for 2-MZM measurements

In order to get a desired projector $\Pi_s^{(jk)}$ in the measurement-only scheme, we can utilize a repeat-until-success "forced-measurement" procedure. When the measurement of $i\gamma_j\gamma_k$ is performed, the probability of obtaining the desired outcome is 1/2 (except for the initial projector on the ancillary MZMs, which should have deterministic outcome). If an undesired measurement outcome is obtained, we can essentially undo this measurement by performing a parity measurement on the pair of MZMs measured in the previous step, and then perform the measurement of $i\gamma_j\gamma_k$ again. Each such measurement of $i\gamma_j\gamma_k$ yields a new probability of 1/2 of obtaining the desired measurement outcome. This repeated attempt and reset process does not collapse the encoded computational state, because we are utilizing ancillary MZMs and measurements in a manner similar to the quantum state teleportation protocol. In other words, the measurements simply alter which subset of the system encodes the computational state. On average, the number of attempts needed (including the first one) to obtain the desired measurement outcome in this way is 2. The likelihood of not succeeding to obtain the desired outcome within $n$ attempts is $2^{-n}$, so failure is exponentially suppressed.

For example, suppose we wish to implement the $S$ gate via the sequence of projection operators $\Pi_+^{(34)}\Pi_+^{(23)}\Pi_+^{(13)}$:

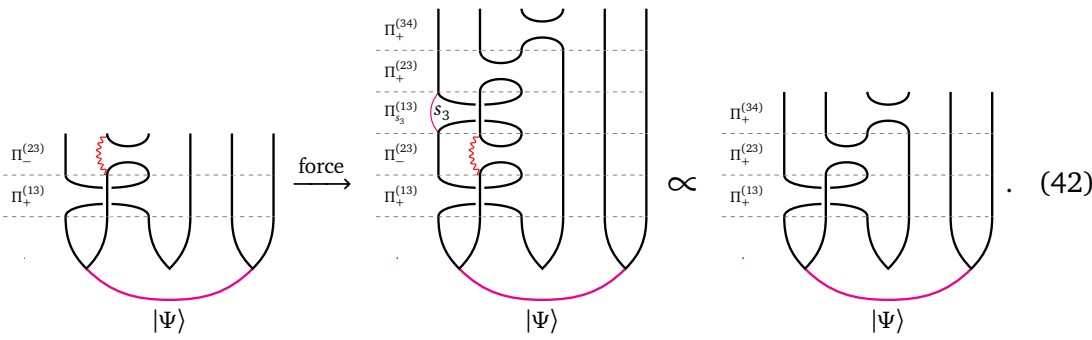

$$\tag{41}$$

Imagine that we perform the first step's measurement of $i\gamma_1\gamma_3$ with the desired outcome $s_1 = +$, but for the second step's measurement of $i\gamma_2\gamma_3$, we obtain the undesired outcome $s_2 = -$. At this point, we can repeat the measurement of $i\gamma_1\gamma_3$ (the outcome of which is irrelevant) and then repeat the measurement of $i\gamma_2\gamma_3$, with another $1/2$ probability of obtaining the desired outcome $s_2 = +$. If the undesired measurement outcome is obtained again, we repeat this process until the desired outcome is obtained. This example is depicted in the following (recall that the purple line indicates unspecified measurement outcomes, here the outcome does not affect the result):

$$\tag{42}$$

Notice that the measurements corresponding to $\Pi_-^{(23)}$ and $\Pi_{s_3}^{(13)}$ are rendered inconsequential by the forcing procedure. Diagrammatically, this can be verified by applying isotopy invariance (bending and straightening lines) on the the diagrams and using the fact that fermions (red wavy lines) whose ends both connect to the same MZM line give rise to an overall phase at most, and can thus be removed without changing the state. Algebraically, this can be verified by checking that

$$\Pi_s^{(jk)}\Pi_p^{(kl)}\Pi_q^{(jk)}\Pi_r^{(kl)} \propto \Pi_s^{(jk)}\Pi_r^{(kl)}, \tag{43}$$

through a straightforward manipulation of Majorana operators.

In order to distinguish the application of a forced-measurement operation from projectors associated with a physical measurement, we denote the application of this forced-measurement protocol applied to the MZM pair $(jk)$ in a sequence following a measurement of $(kl)$ as $\overset{\text{\tiny ⊄}}{\Pi}_s^{(jk)}$. In terms of the sequence of projectors with the desired measurement outcome $s$ obtained at the $n$th attempt, we have

$$\overset{\text{\tiny ⊄}}{\Pi}_s^{(jk)}\Pi_r^{(kl)} = \Pi_s^{(jk)}\Pi_{r_{n-1}}^{(kl)}\Pi_{s_{n-1}}^{(jk)}\cdots\Pi_{r_3}^{(kl)}\Pi_{s_2}^{(jk)}\Pi_{r_2}^{(kl)}\Pi_{s_1}^{(jk)}\Pi_r^{(kl)}, \tag{44}$$

where $s_a \neq s$ for $a = 1,\ldots,n-1$, and the measurement outcomes $r_a$ are irrelevant.

The difficulty weight of a sequence of measurements is simply the product of difficulty weights of each measurement in the sequence. Since a forced measurement involves a probabilistically determined number of measurements, we define the difficulty weight associated with an application of a forced measurement to be the geometric mean (over the

distribution for $n$) of the difficulty weight of the sequence. In other words, the difficultly weight of this forced measurement is taken to be

$$\overset{\wp}{w}(jk) = \exp\left(\sum_{n=1}^{\infty} 2^{-n} \ln\left[w(jk)^n \, w(kl)^{n-1}\right]\right) = w(jk)^2 \, w(kl). \tag{45}$$

This is equal to the difficulty weight of the average case sequence, i.e. $\langle n \rangle = 2$ attempts. [3]

There is an alternative to this forced-measurement protocol that similarly achieves the desired measurement outcome within a measurement-only sequence. When the measurement of the MZM pair $(jk)$ immediately following a measurement of the MZM pair $(kl)$ yields an undesired outcome, instead of resetting by repeating the previous measurement of $(kl)$, we can instead reset by measuring the MZM pair $(jl)$. This is shown in the following diagrammatic representation for the desired projector sequence $\Pi_+^{(34)}\Pi_+^{(36)}\Pi_+^{(23)}\Pi_+^{(13)}$, when an undesired measurement outcome occurs for the measurement of MZMs $(36)$:

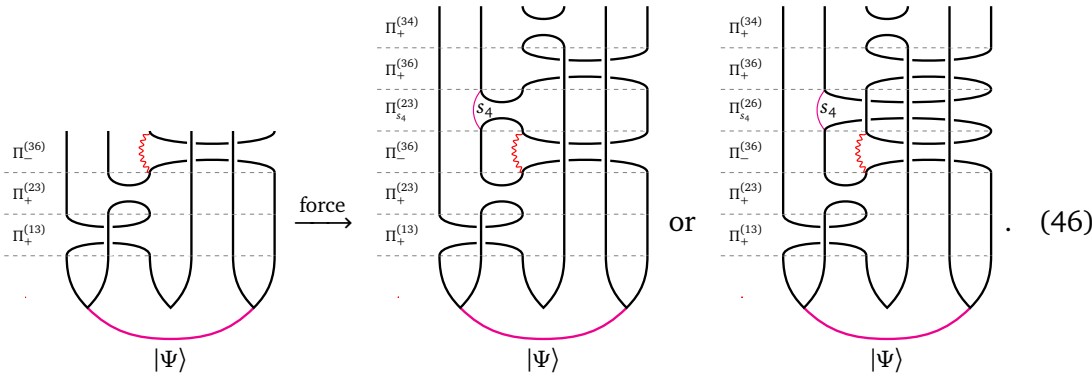

That this procedure works as claimed can be verified diagrammatically by applying isotopy invariance and removing fermion lines, as allowed. Algebraically, this can be verified by checking that

$$\Pi_s^{(jk)}\Pi_p^{(jl)}\Pi_q^{(jk)}\Pi_r^{(kl)} \propto \Pi_s^{(jk)}\Pi_r^{(kl)}, \tag{47}$$

through a straightforward manipulation of Majorana operators.

In order to differentiate the application of this alternative forced-measurement protocol from the previous one (and from an ordinary projector), we denote the application of this forced-measurement protocol applied to the MZM pair $(jk)$ in a sequence following a measurement of $(kl)$ as $\widehat{\Pi}_s^{(jk)}$. In terms of the sequence of projectors with the desired measurement outcome $s$ obtained at the $n$th attempt, we have

$$\widehat{\Pi}_s^{(jk)}\Pi_r^{(kl)} = \Pi_s^{(jk)}\Pi_{p_{n-1}}^{(jl)}\Pi_{s_{n-1}}^{(jk)}\cdots\Pi_{p_3}^{(jl)}\Pi_{s_2}^{(jk)}\Pi_{p_2}^{(jl)}\Pi_{s_1}^{(jk)}\Pi_r^{(kl)}, \tag{48}$$

where $s_a \neq s$ for $a = 1, \ldots, n-1$, and the measurement outcomes $p_a$ are irrelevant. Similar to the case of the previous forced-measurement protocol, the difficulty weight associated with an application of this alternative forced measurement is defined to be the geometric mean of the difficulty weight of the sequence, and is equal to the difficulty weight of the average case sequence, i.e. $\langle n \rangle = 2$ attempts. This is given by

$$\widehat{w}(jk) = w(jk)^2 \, w(jl). \tag{49}$$

This alternative forcing protocol would be preferable to the previous one in situations where parity measurements of MZMs $(jl)$ are physically less difficult to perform that those of MZMs $(kl)$, i.e. when $w(jl) < w(kl)$.

---

[3]For more general non-Abelian anyons, the probability factors $2^{-n}$ and corresponding average number of attempts $\langle n \rangle = 2$ need to be replaced with the outcome probabilities particular to the type of anyons involved.

## 3.2 Forced-measurement protocols involving $2N$-MZM measurements

We now discuss similar forced-measurement strategies for $2N$-MZM measurements, in particular 4-MZM measurements, as well as 2-MZM measurements that follow a 4-MZM measurement.

In general, the required condition for a forced measurement on $\mathcal{M}_2$ following a measurement of $\mathcal{M}_1$ to be possible is the following:

$$\Pi_{s_4}^{(\mathcal{M}_2)}\Pi_{s_3}^{(\mathcal{M}_3)}\Pi_{s_2}^{(\mathcal{M}_2)}\Pi_{s_1}^{(\mathcal{M}_1)} \propto \Pi_{s_4}^{(\mathcal{M}_2)}\Pi_{s_1}^{(\mathcal{M}_1)}, \tag{50}$$

for some choice of $\mathcal{M}_3$. This, of course, assumes the subsequent projectors in this sequence do not commute, so $\Gamma_{\mathcal{M}_1}\Gamma_{\mathcal{M}_2} = -\Gamma_{\mathcal{M}_2}\Gamma_{\mathcal{M}_1}$ and $\Gamma_{\mathcal{M}_2}\Gamma_{\mathcal{M}_3} = -\Gamma_{\mathcal{M}_3}\Gamma_{\mathcal{M}_2}$, as otherwise they would not interact in a way for which a forcing protocol can be actualized. As such, we see that

$$\begin{aligned}
\Pi_{s_4}^{(\mathcal{M}_2)}\Pi_{s_3}^{(\mathcal{M}_3)}\Pi_{s_2}^{(\mathcal{M}_2)}\Pi_{s_1}^{(\mathcal{M}_1)} &= \Pi_{s_4}^{(\mathcal{M}_2)}\frac{\mathbb{1} + s_3\Gamma_{\mathcal{M}_3}}{2}\Pi_{s_2}^{(\mathcal{M}_2)}\Pi_{s_1}^{(\mathcal{M}_1)} \\
&= \frac{1}{2}\Pi_{s_4}^{(\mathcal{M}_2)}\left(\Pi_{s_2}^{(\mathcal{M}_2)} + \Pi_{-s_2}^{(\mathcal{M}_2)}s_3\Gamma_{\mathcal{M}_3}\right)\Pi_{s_1}^{(\mathcal{M}_1)} \\
&= \frac{1}{2}\Pi_{s_4}^{(\mathcal{M}_2)}\left(s_3\Gamma_{\mathcal{M}_3}\right)^{\frac{1-s_2s_3}{2}}\Pi_{s_1}^{(\mathcal{M}_1)}.
\end{aligned} \tag{51}$$

It is clear that Eq. (50) will hold if either

$$\mathcal{M}_3 = \mathcal{M}_1 \quad \text{or} \quad \mathcal{M}_3 = (\mathcal{M}_1\bigcup\mathcal{M}_2)\setminus(\mathcal{M}_1\bigcap\mathcal{M}_2), \tag{52}$$

i.e. if $\Gamma_{\mathcal{M}_3} = \Gamma_{\mathcal{M}_1}$ or $\Gamma_{\mathcal{M}_3} \propto \Gamma_{\mathcal{M}_2}\Gamma_{\mathcal{M}_1}$, since the projectors will then allow $\Gamma_{\mathcal{M}_3}$ to be replaced by a constant.

This provides a generalization of the two different forcing protocols described in the previous subsection. Note that the latter condition can lead to invalid measurement sequences that would collapse the qubit states or to measurements of greater than $2N$-MZMs if $\mathcal{M}_1$ and $\mathcal{M}_2$ contain more than two elements each. This is a case we want to avoid, as the cost of doing multi-MZM measurements grows dramatically in the number of MZMs. On the other hand, the case of $\mathcal{M}_3 = \mathcal{M}_1$ is always permissible and so forced measurements are always possible when needed.

More explicitly, for measurement sequences involving 4-MZM measurements, some of the possible forced-measurement protocols include

$$\overset{\frown}{\Pi}_s^{(ac)}\Pi_r^{(ab;x'y')} = \Pi_s^{(ac)}\Pi_{p_{n-1}}^{(bc;x'y')}\Pi_{s_{n-1}}^{(ac)}\cdots\Pi_{p_2}^{(bc;x'y')}\Pi_{s_1}^{(ac)}\Pi_r^{(ab;x'y')}, \tag{53a}$$

$$\overset{\leftrightarrow}{\Pi}_s^{(ac)}\Pi_r^{(ab;x'y')} = \Pi_s^{(ac)}\Pi_{r_{n-1}}^{(ab;x'y')}\Pi_{s_{n-1}}^{(ac)}\cdots\Pi_{r_2}^{(ab;x'y')}\Pi_{s_1}^{(ac)}\Pi_r^{(ab;x'y')}, \tag{53b}$$

$$\overset{\leftrightarrow}{\Pi}_s^{(ab;x'y')}\Pi_r^{(ac)} = \Pi_s^{(ab;x'y')}\Pi_{r_{n-1}}^{(ac)}\Pi_{s_{n-1}}^{(ab;x'y')}\cdots\Pi_{r_2}^{(ac)}\Pi_{s_1}^{(ab;x'y')}\Pi_r^{(ac)}, \tag{53c}$$

$$\overset{\leftrightarrow}{\Pi}_s^{(ac;w'z')}\Pi_r^{(ab;x'y')} = \Pi_s^{(ac;w'z')}\Pi_{r_{n-1}}^{(ab;x'y')}\Pi_{s_{n-1}}^{(ac;w'z')}\cdots\Pi_{r_2}^{(ab;x'y')}\Pi_{s_1}^{(ac;w'z')}\Pi_r^{(ab;x'y')}, \tag{53d}$$

which have the corresponding difficulty weights

$$\overset{\frown}{w}(ac) = w(ac)^2 w(bc;x'y'), \tag{54a}$$

$$\overset{\leftrightarrow}{w}(ac) = w(ac)^2 w(ab;x'y'), \tag{54b}$$

$$\overset{\leftrightarrow}{w}(ab;x'y') = w(ab;x'y')^2 w(ac), \tag{54c}$$

$$\overset{\leftrightarrow}{w}(ac;w'z') = w(ac;w'z')^2 w(ab;x'y'). \tag{54d}$$

## 3.3 Procrastination methods

The forced-measurement protocols of the previous subsections provides control over which fermionic parities are projected upon at each step, which allows us to effectively implement a projector sequence that generates a specified target computational gate. In principle, one can apply a forced-measurement protocol for every projector in a given projector sequence. In practice, this turns out to be an inefficient strategy, since the different projectors in the sequence may have a correlated effect on the resulting gate. This subsection outlines theoretical tools for determining which projectors in a sequence have a correlated effect and, therefore, which specific measurements can tolerate any outcome and which are required to be forced in order to obtain the intended computational gate. We will show that the measurement outcomes can only change the final gate by an overall Pauli operator for the case of MZMs. By the anti-commutation properties of Pauli operators, we need only apply a forced-measurement protocol for at most 3 of the projectors for each hexon in a measurement-only projector sequence in order to realize a specified target gate.

Diagrammatically, this can be understood by recalling that a measurement with outcome $s$, corresponding to the projector $\Pi_s^{(jk)}$, is represented by a cap and cup in the MZM lines corresponding to $\gamma_j$ and $\gamma_k$, with a fermion line connecting the cap and cup for outcomes $s = -$, as shown in Eq. (11). For every $s = -$ projector in a measurement-only sequence of projectors, we can slide the corresponding fermion line (that terminates on two MZM lines) up to the top of the diagram using the diagrammatic rules. Each such fermion line that has been slid to the top of the diagram simply connects two MZM lines $a$ and $b$, i.e. it results in a parity operator $i\gamma_a\gamma_b$. If every measurement sequences starts and ends with a forced $\Pi_+^{(\text{anc})}$, the fermion lines will not connect to the ancillary MZMs' lines when pushed to the top of the diagram, i.e. $a$ and $b$ do not correspond to ancillary MZMs. Thus, the femion lines slid to the top of the diagram correspond to the following Pauli operators (cf. Eq. (7)):

$$
\begin{array}{c|c}
i\gamma_a\gamma_b & \text{Pauli} \\
\hline
i\gamma_1\gamma_2 & \mathbb{1} \otimes Z \\
i\gamma_1\gamma_5 & Z \otimes Y \\
i\gamma_1\gamma_6 & \mathbb{1} \otimes X \\
i\gamma_2\gamma_5 & Z \otimes X \\
i\gamma_2\gamma_6 & -\mathbb{1} \otimes Y \\
i\gamma_5\gamma_6 & Z \otimes Z
\end{array} \quad . \tag{55}
$$

In other words, the complete operation effected on the computational subspace by a measurement-only sequence will be a braiding transformation (hence a Clifford gate) determined by which MZMs were measured in the sequence, followed by a Pauli gate determined by the measurement outcomes.

Thus, we see that a single hexon projector sequence

$$
\mathcal{G} = \Pi_+^{(34)} \Pi_{s_{n-1}}^{(\mathcal{M}_{n-1})} \dots \Pi_{s_1}^{(\mathcal{M}_1)} \Pi_+^{(34)}, \tag{56}
$$

(with projection channels $s_\mu$ that need not all be $+$) compiling to gate $G$ can be rewritten as

$$
\mathcal{G} = \left( i\gamma_{j_q}\gamma_{k_q} \cdots i\gamma_{j_1}\gamma_{k_1} \right) \mathcal{G}_+ \tag{57}
$$

$$
\propto (Z^p \otimes P)\left( \Pi_+^{(\text{anc})} \otimes G_+ \right) = \Pi_+^{(\text{anc})} \otimes P G_+ \tag{58}
$$

where

$$
\mathcal{G}_+ = \Pi_+^{(34)} \Pi_+^{(\mathcal{M}_{n-1})} \dots \Pi_+^{(\mathcal{M}_1)} \Pi_+^{(34)}, \tag{59}
$$

is the projector sequence obtained from $\mathcal{G}$ by switching all its projectors to have $s_\mu = +$, and $q$ is the number of $s_\mu = -$ projectors in the sequence $\mathcal{G}$. Furthermore, the product of fermionic parity operators corresponding to the fermion lines after sliding them to the top of the diagram is equal to $i\gamma_{j_q}\gamma_{k_q}\cdots i\gamma_{j_1}\gamma_{k_1} = Z^p \otimes P$, where $p$ is an integer and $P \in \{\mathbb{1}, X, Y, Z\}$ is a Pauli gate. Thus, the effect of the measurement outcomes $s_\mu$ in a single hexon projector sequence is to change the resulting compiled gate by at most a Pauli gate.

A useful example to consider is the following projector sequence, which can realize any of the Pauli gates, depending on the measurement outcomes:

$$\mathcal{P} = \Pi_+^{(34)}\Pi_{s_5}^{(23)}\Pi_{s_4}^{(13)}\Pi_{s_3}^{(23)}\Pi_{s_2}^{(34)}\Pi_{s_1}^{(35)} \propto \mathbf{\Pi}_+^{(\text{anc})} \otimes P, \tag{60}$$

$$P = Z^{\frac{1-s_5}{2}}Z^{\frac{1-s_3}{2}}X^{\frac{1-s_2}{2}}. \tag{61}$$

Notice that the resulting gate $P$ is independent of $s_1$ and $s_4$. Diagrammatically, this result is easily obtained, as follows.

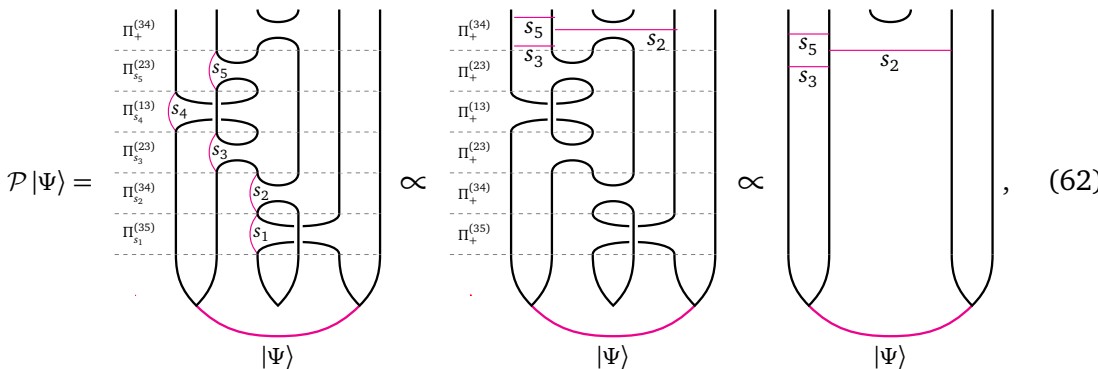

$$\tag{62}$$

where the equality is up to overall phases. Notice that isotopy of the MZM lines allows them to be straightened out, leaving no nontrivial braiding, and hence $P_+ = \mathbb{1}$, where $\mathcal{P}_+$ is the sequence $\mathcal{P}$ with all measurement outcomes $s_\mu = +$ and $P_+$ is the gate $\mathcal{P}_+$ compiles to. Also notice that both ends of the $s_1$ line connect to the $j = 5$ MZM line when straightened, and both ends of the $s_4$ line connect to the $j = 1$ MZM line when straighten, so the $s_1$ and $s_4$ lines can be removed without affecting the resulting computational gate, i.e. $\gamma_1\gamma_1 = \gamma_5\gamma_5 = \mathbb{1}$. Finally, after straightening out the MZM lines and sliding the $s_\mu$ lines to the top of the diagram, we see that $s_2 = -$ would contribute the operator $i\gamma_2\gamma_5 = Z \otimes X$, $s_3 = -$ would contribute $i\gamma_1\gamma_2 = \mathbb{1} \otimes Z$, and $s_5 = -$ would contribute $i\gamma_1\gamma_2 = \mathbb{1} \otimes Z$. Thus, the compiled gate is $P = Z^{\frac{1-s_5}{2}}Z^{\frac{1-s_3}{2}}X^{\frac{1-s_2}{2}}$, as claimed.

Some specific realizations include

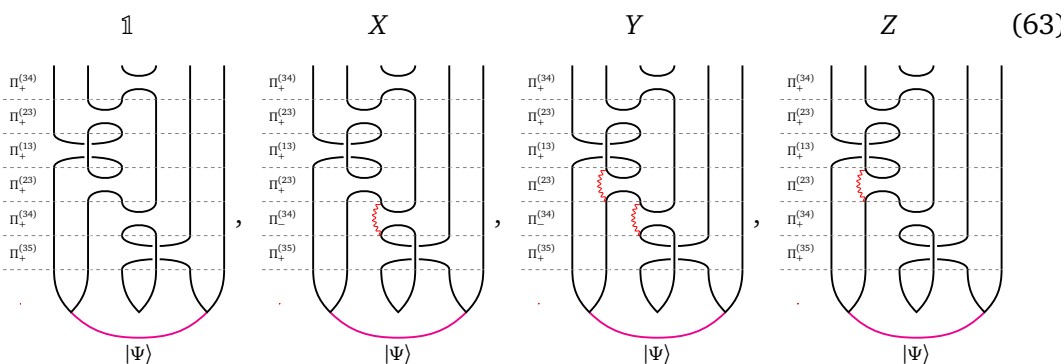

$$\tag{63}$$

Similar arguments apply for the case of multi-hexon projector sequences, which demonstrate that the different choices of projection channels $s_\mu$ change the compiled gate by at most a multi-qubit Pauli gate. A more general argument that verifies this is given in Sec. 4.

Finally, by tracking the effects of the projection channels $s_\mu$ on the resulting compiled gate in this manner, we can see which measurements in the sequence need to be forced in order to obtain the desired gate. In particular, for a single hexon, when we slide all the fermion lines in a projector sequence to the top of the diagram, each line can either be removed or end up in one of the six configurations connecting MZM lines represented by the fermion parity operators $i\gamma_j\gamma_k$ listed in Eq. (55). In turn, this determines which Pauli operator a given measurement outcome contributes to $P$ in the decomposition $G = PG_+$. In this way, it is clear that a measurement sequence generating a Clifford gate requires *at most* three of its measurements in each hexon to be forced – one of which is needed to end with the proper final state $+$ of the ancillary MZMs (via the projector $\Pi_+^{(\text{anc})}$) and at most two of which are needed to ensure that the desired $P$ is obtained in the sequence. For instance, in the example above, we see that sequence $\mathcal{P}$ can generate a particular desired Pauli gate for any values of $s_1$, $s_3$, and $s_4$, by choosing $s_2$ and $s_5$ appropriately, i.e. by applying forced measurements for the corresponding steps.

### 3.4 Adaptive methods

While forced measurements and procrastination are, strictly speaking, adaptive protocols, it is worth considering adaptive methods that change the sequence of projectors/forced measurements in a more complex manner. This could potentially find utility when the projector sequence requires a measurement that is particularly difficult, but which we wish to avoid including in forcing protocols, as doing so would increase the number of times this costly measurement will need to be performed, on average. However, this strategy generally increases the total number of measurements needed, so the most likely instances that could benefit from its use would involve multi-hexon measurements whose difficulty outweighs that of several single-hexon measurements. For an example of such an adaptive approach proving beneficial, see Appendix A.

## 4 Majorana-Pauli Tracking

When using MZMs for measurement-only topological quantum computing, it is possible to forego the use of forced measurements by instead tracking the measurement outcomes, the different possibilities of which only change the resulting transformation by Pauli gates. [25] More generally, a similar tracking strategy can be employed when the measurement outcomes are always guaranteed to be Abelian anyons, e.g. when using Parafendleyons (parafermion zero modes), as was applied for measurement-only braiding transformations in Ref. [26]. The tracking methods allow for the use of fewer physical measurement operations and makes the sequence of measurement operations used for topological gate operations completely deterministic. The cost of using such methods is the need to classically track the measurement outcomes and utilize adaptive methods when non-Clifford gates are introduced.

For a system of $N$ hexons, we now write a sequence of projection operators that compiles to a gate $G_{(s_n,\vec{s},s_0)}$ acting on the computational state space as

$$\mathcal{G}_{(s_n,\vec{s},s_0)} = \Pi_{s_n}^{(\text{anc})}\Pi_{s_{n-1}}^{(\mathcal{M}_{n-1})}\dots\Pi_{s_1}^{(\mathcal{M}_1)}\Pi_{s_0}^{(\text{anc})} \propto \Upsilon_{s_n s_0}\,\Pi_{s_0}^{(\text{anc})} \otimes G_{(s_n,\vec{s},s_0)}, \tag{64}$$

where $\mathcal{M}_\mu$ are the ordered sets of (up to $2N$) MZMs whose collective fermionic parity is being projected onto $s_\mu = \pm$ (collectively denoted as $\vec{s}$, where $s_0$ and $s_n$ are themselves vectors), and the ancillary projectors take the form

$$\Pi_{s_n}^{(\text{anc})} = \Pi_{s_{n,1}}^{(34)} \otimes \cdots \otimes \Pi_{s_{n,N}}^{(3'\dots'4'\dots')}. \tag{65}$$

Since we are allowing for the initial and final ancillary projectors to be inequivalent, we introduced the operator

$$\Upsilon_{s_n s_0} = \bigotimes_{j=1}^{N} \left( i\gamma_{4,j}\gamma_{5,j} \right)^{\frac{1-s_{n,j}s_{0,j}}{2}} = \bigotimes_{j=1}^{N} \left( X_j \otimes \mathbb{1}_j \right)^{\frac{1-s_{n,j}s_{0,j}}{2}}, \tag{66}$$

where $\gamma_{a,j}$ is the $a$th MZM of the $j$th hexon. This operator flips the state of each ancillary qubit whose initial and final projections differ. In other words, $\Upsilon_{s_n s_0} \Pi^{(\mathrm{anc})}_{s_0} \Upsilon_{s_n s_0} = \Pi^{(\mathrm{anc})}_{s_n}$.

It is straightforward to show using the diagrammatic formalism that the sequence of projectors in Eq. (64) must reduce to an operator with the form of the right hand side of that expression. The only task is to determine the operator $G_{(s_n,\vec{s},s_0)}$. By definition, we only consider a projector sequence to be a valid measurement-only sequence if $G_{(s_n,\vec{s},s_0)}$ is unitary, i.e. does not reduce the rank of the computational subspace.

In the following, we show that different projection channels $(\mathbf{s_n},\vec{s},\mathbf{s_0})$ for a fixed sequence of MZM sets $\mathcal{M}_\mu$ will, at most, change the compiled gate $G_{(s_n,\vec{s},s_0)}$ by a multi-qubit Pauli gate, assuming it does not reduce the rank. [4] These Pauli gate differences are determined by the corresponding sequences of projections. In other words, for the same $\mathcal{M}_\mu$ with another sequence of projection channels $(\mathbf{r_n},\vec{r},\mathbf{r_0})$ that does not project to zero, we have

$$G_{(r_n,\vec{r},r_0)} = P_{(r_n,\vec{r},r_0;s_n,\vec{s},s_0)} G_{(s_n,\vec{s},s_0)}, \tag{67}$$

where $P_{(r_n,\vec{r},r_0;s_n,\vec{s},s_0)}$ is an $N$-qubit Pauli gate.

Thus, if we perform a measurement-only sequence of measurements for a desired gate and track the measurement outcomes, rather than using forced measurements, we will have a known Pauli gate correction. If the non-Clifford gates that we utilize in a quantum computation are single qubit phase gates (in any of the Pauli bases), we can also push the Pauli gate correction through the phase gates with at most a Clifford gate correction that can be dealt with by updating the subsequent Clifford gate in the computation to absorb the Clifford correction. When non-Clifford phase gates are implemented by injecting states, such a Clifford correction will be necessary anyway, so this would not be a significantly greater burden.

## 4.1 Proof of Majorana-Pauli tracking

We now prove Eq. (67) by taking the product

$$
\begin{aligned}
\mathcal{G}_{(r_n,\vec{r},r_0)}\mathcal{G}^\dagger_{(s_n,\vec{s},s_0)} &= \Pi^{(\mathrm{anc})}_{r_n}\Pi^{(\mathcal{M}_{n-1})}_{r_{n-1}}\cdots\Pi^{(\mathcal{M}_1)}_{r_1}\Pi^{(\mathrm{anc})}_{r_0}\Pi^{(\mathrm{anc})}_{s_0}\Pi^{(\mathcal{M}_1)}_{s_1}\cdots\Pi^{(\mathcal{M}_{n-1})}_{s_{n-1}}\Pi^{(\mathrm{anc})}_{s_n} \\
&\propto \delta_{r_0,s_0}\Upsilon_{r_n s_n}\Pi^{(\mathrm{anc})}_{s_n}\otimes G_{(r_n,\vec{r},r_0)}G^\dagger_{(s_n,\vec{s},s_0)},
\end{aligned} \tag{68}
$$

and recursively using relations that will reduce the product of projectors.

For this, we will utilize the relation

$$\Pi^{(\mathcal{B}_1)}_{q_1}\cdots\Pi^{(\mathcal{B}_k)}_{q_k} + z\Pi^{(\mathcal{B}_1)}_{-q_1}\cdots\Pi^{(\mathcal{B}_k)}_{-q_k} = \left( q_1\Gamma_{\mathcal{B}_1} \right)^{\frac{1-z}{2}}\Pi^{(\mathcal{B}_{1:2})}_{q_{1:2}}\cdots\Pi^{(\mathcal{B}_{k-1:k})}_{q_{k-1:k}}, \tag{69}$$

that holds for ordered sets $\mathcal{B}_\mu$ of even numbers of MZMs such that the $\Pi^{(\mathcal{B}_\mu)}_{q_\mu}$ all commute with each other, i.e. $|\mathcal{B}_\mu \bigcap \mathcal{B}_\nu|$ is even for all $\mu$ and $\nu$, where $z, q_\alpha = \pm 1$. In this expression,

---

[4]It is possible that changing the projection channels will yield a sequence that projects to zero. This merely indicates that such a sequence of projection channels cannot occur as a result of measurements, i.e. it would have probability zero. A trivial example of this would be if we let $\mathcal{M}_\mu = \mathcal{M}_{\mu+1}$ and $s_\mu = -s_{\mu+1}$, but it is possible for more subtle cancellations to occur in a measurement-only sequence.

we define $q_{\mu:\nu} = \pm 1$ and the ordered sets $\mathcal{B}_{\mu:\nu}$ obtained by taking the symmetric difference $(\mathcal{B}_\mu \bigcup \mathcal{B}_\nu) \setminus (\mathcal{B}_\mu \bigcap \mathcal{B}_\nu)$, and ordering its elements such that

$$\Pi_{q_{\mu:\nu}}^{(\mathcal{B}_{\mu:\nu})} = \frac{\mathbb{1} + q_{\mu:\nu}\Gamma_{\mathcal{B}_{\mu:\nu}}}{2} = \frac{\mathbb{1} + q_\mu q_\nu \Gamma_{\mathcal{B}_\mu}\Gamma_{\mathcal{B}_\nu}}{2}. \tag{70}$$

We notice that, since $|\mathcal{B}_\mu \bigcap \mathcal{B}_\nu|$ is even, the operator $\frac{1}{2}(\mathbb{1} + q_\mu q_\nu \Gamma_{\mathcal{B}_\mu}\Gamma_{\mathcal{B}_\nu})$ will always be a projector for a joint fermionic parity operator $\Gamma_{\mathcal{B}_{\mu:\nu}}$.

We can establish Eq. (69) inductively, starting by noticing that

$$\Pi_q^{(\mathcal{B})} + z\Pi_{-q}^{(\mathcal{B})} = \frac{\mathbb{1} + q\Gamma_\mathcal{B}}{2} + z\frac{\mathbb{1} - q\Gamma_\mathcal{B}}{2} = (q\Gamma_\mathcal{B})^{\frac{1-z}{2}}. \tag{71}$$

For $k = 2$, we see that

$$\begin{aligned}
\Pi_{q_1}^{(\mathcal{B}_1)}\Pi_{q_2}^{(\mathcal{B}_2)} + z\Pi_{-q_1}^{(\mathcal{B}_1)}\Pi_{-q_2}^{(\mathcal{B}_2)} &= \frac{\mathbb{1} + q_1\Gamma_{\mathcal{B}_1}}{2}\frac{\mathbb{1} + q_2\Gamma_{\mathcal{B}_2}}{2} + z\frac{\mathbb{1} - q_1\Gamma_{\mathcal{B}_1}}{2}\frac{\mathbb{1} - q_2\Gamma_{\mathcal{B}_2}}{2} \\
&= \frac{1+z}{2}\frac{\mathbb{1} + q_1 q_2 \Gamma_{\mathcal{B}_1}\Gamma_{\mathcal{B}_2}}{2} + \frac{1-z}{2}\frac{q_1\Gamma_{\mathcal{B}_1} + q_2\Gamma_{\mathcal{B}_2}}{2} \\
&= \left(q_1\Gamma_{\mathcal{B}_1}\right)^{\frac{1-z}{2}}\Pi_{q_{1:2}}^{(\mathcal{B}_{1:2})}. \tag{72}
\end{aligned}$$

If Eq. (69) holds for $k \geq 2$, then

$$\begin{aligned}
\left(q_1\Gamma_{\mathcal{B}_1}\right)^{\frac{1-z}{2}}\Pi_{q_{1:2}}^{(\mathcal{B}_{1:2})}\cdots\Pi_{q_{k:k+1}}^{(\mathcal{B}_{k:k+1})} &= \left(\Pi_{q_1}^{(\mathcal{B}_1)}\cdots\Pi_{q_k}^{(\mathcal{B}_k)} + z\Pi_{-q_1}^{(\mathcal{B}_1)}\cdots\Pi_{-q_k}^{(\mathcal{B}_k)}\right)\Pi_{q_{k:k+1}}^{(\mathcal{B}_{k:k+1})} \\
&= \left(\Pi_{q_1}^{(\mathcal{B}_1)}\cdots\Pi_{q_k}^{(\mathcal{B}_k)} + z\Pi_{-q_1}^{(\mathcal{B}_1)}\cdots\Pi_{-q_k}^{(\mathcal{B}_k)}\right)\left(\Pi_{q_k}^{(\mathcal{B}_k)}\Pi_{q_{k+1}}^{(\mathcal{B}_{k+1})} + \Pi_{-q_k}^{(\mathcal{B}_k)}\Pi_{-q_{k+1}}^{(\mathcal{B}_{k+1})}\right) \\
&= \Pi_{q_1}^{(\mathcal{B}_1)}\cdots\Pi_{q_{k+1}}^{(\mathcal{B}_{k+1})} + z\Pi_{-q_1}^{(\mathcal{B}_1)}\cdots\Pi_{-q_{k+1}}^{(\mathcal{B}_{k+1})} \tag{73}
\end{aligned}$$

shows that it holds for $k + 1$, and this completes the induction argument.

Returning to the product of projectors in Eq. (68), each step of the recursion involves a product in the middle of the string of projectors that takes the form

$$\Pi_r^{(\mathcal{M})}\Pi_{q_1}^{(\mathcal{A}_1)}\cdots\Pi_{q_k}^{(\mathcal{A}_k)}\Pi_{p_1}^{(\mathcal{C}_1)}\cdots\Pi_{p_l}^{(\mathcal{C}_l)}\Pi_s^{(\mathcal{M})}, \tag{74}$$

where $|\mathcal{A}_\alpha \bigcap \mathcal{A}_{\alpha'}|$, $|\mathcal{A}_\alpha \bigcap \mathcal{C}_\beta|$, $|\mathcal{C}_\beta \bigcap \mathcal{C}_{\beta'}|$, and $|\mathcal{M} \bigcap \mathcal{C}_\beta|$ are all even, while $|\mathcal{M} \bigcap \mathcal{A}_\alpha|$ are all odd. In other words, the projectors $\Pi_{q_\alpha}^{(\mathcal{A}_\alpha)}$ and $\Pi_{p_\beta}^{(\mathcal{C}_\beta)}$ all commute with each other, $\Pi_s^{(\mathcal{M})}$ commutes with $\Pi_{p_\nu}^{(\mathcal{C}_\beta)}$, and $\Gamma_\mathcal{M}$ anticommutes with all $\Gamma_{\mathcal{A}_\alpha}$. From this, we find

$$\begin{aligned}
\Pi_r^{(\mathcal{M})}\Pi_{q_1}^{(\mathcal{A}_1)}\cdots\Pi_{q_k}^{(\mathcal{A}_k)}\Pi_{p_1}^{(\mathcal{C}_1)}\cdots\Pi_{p_l}^{(\mathcal{C}_l)}\Pi_s^{(\mathcal{M})} &= \Pi_r^{(\mathcal{M})}\Pi_{q_1}^{(\mathcal{A}_1)}\cdots\Pi_{q_k}^{(\mathcal{A}_k)}\Pi_s^{(\mathcal{M})}\Pi_{p_1}^{(\mathcal{C}_1)}\cdots\Pi_{p_l}^{(\mathcal{C}_l)} \\
&= \frac{1}{2}\left(\Pi_{q_1}^{(\mathcal{A}_1)}\cdots\Pi_{q_k}^{(\mathcal{A}_k)} + rs\Pi_{-q_1}^{(\mathcal{A}_1)}\cdots\Pi_{-q_k}^{(\mathcal{A}_k)}\right)\Pi_s^{(\mathcal{M})}\Pi_{p_1}^{(\mathcal{C}_1)}\cdots\Pi_{p_l}^{(\mathcal{C}_l)} \\
&= \frac{1}{2}\left(q_1\Gamma_{\mathcal{A}_1}\right)^{\frac{1-rs}{2}}\Pi_{q_{1:2}}^{(\mathcal{A}_{1:2})}\Pi_{q_{2:3}}^{(\mathcal{A}_{2:3})}\cdots\Pi_{q_{k-1:k}}^{(\mathcal{A}_{k-1:k})}\Pi_s^{(\mathcal{M})}\Pi_{p_1}^{(\mathcal{C}_1)}\cdots\Pi_{p_l}^{(\mathcal{C}_l)}, \tag{75}
\end{aligned}$$

where we expanded $\Pi_r^{(\mathcal{M})} = \frac{\mathbb{1} + r\Gamma_\mathcal{M}}{2}$ and anticommuted $\Gamma_\mathcal{M}$ through the $\Gamma_{\mathcal{A}_\alpha}$ to obtain the second line, and then used Eq. (69). We notice that all the projectors in the last line of Eq. (75) commute with each other, since $|\mathcal{A}_{\alpha:\alpha+1} \bigcap \mathcal{M}|$ is even.

Recursively applying Eq. (75) to Eq. (68) and moving extra fermionic parity operators (e.g. $\Gamma_{\mathcal{A}_1}$) through the remaining projectors to the left (which will flip the projection channels when $|\mathcal{A}_1 \bigcap \mathcal{M}_\mu|$ is odd), we find a final result of the form

$$\mathcal{G}_{(r_n,\vec{r},r_0)}\mathcal{G}_{(s_n,\vec{s},s_0)}^\dagger \propto \delta_{r_0,s_0}\Gamma_\mathcal{B}\Pi_{p_1}^{(\mathcal{C}_1)}\cdots\Pi_{p_m}^{(\mathcal{C}_m)}, \tag{76}$$

where the projectors all commute with each other and $\Gamma_{\mathcal{B}}$ is the fermionic parity operator corresponding to some ordered set of MZM labels $\mathcal{B}$ that is determined by the projector sequences. When this does not project to zero, it must be proportional to $\Upsilon_{r_n s_n} \Pi^{(\text{anc})}_{s_n} \otimes G_{(r_n, \vec{r}, s_0)} G^{\dagger}_{(s_n, \vec{s}, s_0)}$, which implies that $\Pi^{(\mathcal{C}_1)}_{p_1} \cdots \Pi^{(\mathcal{C}_m)}_{p_m} = \Pi^{(\text{anc})}_{s_n}$ and that $G_{(r_n, \vec{r}, s_0)} G^{\dagger}_{(s_n, \vec{s}, s_0)}$ is a multi-qubit Pauli operator.

Applying the same argument to $\mathcal{G}^{\dagger}_{(r_n, \vec{r}, r_0)} \mathcal{G}_{(s_n, \vec{s}, s_0)}$ shows that, when this sequence does not project to zero, $G^{\dagger}_{(s_n, \vec{r}, r_0)} G_{(s_n, \vec{s}, s_0)}$ is a multi-qubit Pauli operator. Combining the results for these two cases establishes that when $G_{(r_n, \vec{r}, r_0)}$ and $G_{(s_n, \vec{s}, s_0)}$ are nonzero, they are related by a multi-qubit Pauli gate. This proves Eq. (67).

# 5 Brute-force Optimization of Measurement-Only Generation of Gates

In this section, we discuss optimization strategies for measurement-generated gates and then carry out numerical searches for the optimal measurement-only realizations of gates. We exhaustively search all valid projector sequences, i.e. those that do not collapse the encoded computational state, up to some pre-determined length for single-qubit and two-qubit gates. This is used to determine the optimal measurement sequences for all single-qubit gates. For two-qubit gates, the search space is much larger and we limit our focus on optimization of the controlled-Pauli, $W$, and SWAP gates.

In Appendix B, we will discuss techniques whose computational costs scale better than brute-force search, but which are not guaranteed to find (globally) optimal measurement sequences.

## 5.1 Optimization

There are many possible strategies and layers of optimization that may be employed in an effort to optimize the implementation of computational gates.

The crucial first step is deciding on the metric with respect to which optimization is performed. A simple choice would be the length of measurement sequences, which would provide useful results if all measurements are approximately equally difficult to implement. The difficulty weights introduced in Sec. 2.4 provide a more physically realistic cost function for optimization. The difficulty weight $w(\mathcal{M})$ in Eq. (38) provides a systematic estimation of the error and resource costs of a joint parity measurement of MZMs $\mathcal{M}$. For each measurement-only gate implemented by a sequence of physical measurements corresponding to $\mathcal{M}_1, \ldots, \mathcal{M}_n$, we assign the sequence a difficulty weight defined as the product of its component measurements' weights:

$$w(\{\mathcal{M}_1, \ldots, \mathcal{M}_n\}) = \prod_{\mu=1}^{n} w(\mathcal{M}_{\mu}). \tag{77}$$

Here, $\mathcal{M}_n$ constitutes the ancillary MZMs that actually need to be measured at the final step, i.e. the ones whose projectors do not commute with the rest of the projector sequence, and actually may represent multiple measurements, since each hexon's ancillary pair are projected/measured separately. (We do not include a contribution for the initial ancillary measurement at step $\mu = 0$, since that is provided by previous operations.) The individual weight factors in Eq. (38) will need to be determined through experimental characterization of the physical systems.

Another key aspect of optimization is deciding which set of computational gates to optimize, as all gates cannot be simultaneously optimized. This choice should take into consideration how the quantum computing system is primarily going to be used. For example, if it is implementing certain algorithms or error-correction protocols that call certain gates with high frequency, then it would be natural to optimize the implementation for that set of gates. Some typical choices include the controlled-Pauli gates, the Hadamard gate, and/or all single-qubit Clifford gates. When averaging the sequence weights over the target set, we use the geometric mean due to the multiplicative nature of the weights.

In determining how to appropriately search for optimal measurement sequences, one needs to decide whether one is utilizing Majorana-Pauli tracking methods or forced-measurement methods, as the optimization goals and relation between projector sequences and measurement sequences differ between these two cases, which we detail in the following. In most quantum computing contexts, it will be preferable to utilize the tracking methods, as they generally provide significantly better efficiency than forced-measurement methods. We will demonstrate our optimization methods for both approaches.

As seen in Sec. 2.5, the difficulty weights for different measurements will depend on the MZM labeling configuration used for a given hexon architecture. Thus, the labeling configurations represents another set of parameters over which one can optimize. We carry out the gate optimization analysis within a fixed labeling configuration, and then do so for each possible labeling configuration, up to symmetry. In this way, we can compare and determine which configuration(s) provide the optimal implementation of the relevant gate set.

### 5.1.1 Majorana-Pauli tracking methods

In the case where we use Majorana-Pauli tracking, when we write the measurement-only compilation of a gate $G$ in terms of a projector sequence

$$\mathcal{G}_{(s_n,\vec{s},s_0)} = \mathbf{\Pi}_{s_n}^{(\text{anc})}\Pi_{s_{n-1}}^{(\mathcal{M}_{n-1})}\dots\Pi_{s_1}^{(\mathcal{M}_1)}\mathbf{\Pi}_{s_0}^{(\text{anc})}, \tag{78}$$

the sequence of physical measurements that will be performed is exactly the sequence $\mathcal{M}_1,\dots,\mathcal{M}_n$ specified in the projector sequence. When the physical measurement outcomes do not match the specified projector channels $s_\mu$, the resulting gate will differ from $G$ by at most a Pauli gate, which we track and compensate for at a later time in a more efficient manner. As such, this measurement-only realization of $G$ is assigned the difficulty weight

$$w(\mathcal{G}) = \prod_{\mu=1}^{n} w(\mathcal{M}_\mu), \tag{79}$$

where $\mathcal{M}_n$ corresponds to the measurements of ancillary MZMs.

When Majorana-Pauli tracking is being utilized, it is useful to group together Clifford gates into their Pauli cosets, given by the collections of Clifford gates that are equivalent up to multiplication by an overall (multi-qubit) Pauli gate, i.e. the Pauli coset of a $N$-qubit Clifford gate $G$ is defined to be

$$[G] = \left\{ G' \in \mathsf{C}_N \mid \exists P \in \mathsf{P}_N : G' = PG \right\}. \tag{80}$$

When using tracking, we do not need to be able to generate every Clifford gate; we only need one gate from each Pauli coset, as all differences by Pauli gates are dealt with by the tracking methods. Thus, we can use the most easily realized gate in a given Pauli coset to implement the entire class of gates. In this way, optimization of $[G]$ is carried out by

optimizing all of its elements and selecting the one with lowest difficulty weight to use when any of the gates in $[G]$ is called in a computation. Thus, we define

$$w([G]) = \min_{G \in [G]} w(\mathcal{G}). \tag{81}$$

### 5.1.2 Forced-measurement methods

If, for some reason, one wanted to implement the braiding Clifford gates exactly instead of up to a Pauli correction, forced-measurement protocols would be utilized to ensure the desired gate. (More generally, forced-measurement protocols are actually necessary when the fusion channels include non-Abelian anyons, which is not the case for MZMs.) In this case, one must decide which of the forced-measurement methods to utilize. In our demonstrations, we utilize both forced-measurement protocols and procrastination, but not more complicated adaptive methods.

Using these forced-measurement methods, when we write the measurement-only compilation of a gate $G$ in terms of a projector sequence

$$\mathcal{G}_{(+,\vec{s},+)} = \mathbf{\Pi}_+^{(\text{anc})} \Pi_{s_{n-1}}^{(\mathcal{M}_{n-1})} \ldots \Pi_{s_1}^{(\mathcal{M}_1)} \mathbf{\Pi}_+^{(\text{anc})}, \tag{82}$$

we must determine the minimal set of projectors in the sequence that must be forced in order to generate the desired gate (there are at most three per hexon involved in the gate). We then follow the procrastination method outlined in Sec. 3.3 and convert the projector sequence into a measurement sequence by utilizing forced-measurement protocols only for the generation of these projectors that must be forced, and standard measurements for the rest. For each of the projectors that must be forced, we assess which of the two forced-measurement protocols has the smaller difficulty weight, as given by Eqs. (45), (49), and (54), and we use the lesser weight protocol to implement that forced projector.

The corresponding physical measurement sequence obtained from the projector sequence will be probabilistically determined. As such, we consider the geometric average of the difficulty weight of the physical measurement sequence. This is obtained by starting with the expression for the difficulty weight of the projector sequence and replacing the weights of the projectors that must be forced with the average difficulty weight corresponding to the forced-measurement protocol used. This gives the average difficulty weight of this forced-measurement implementation of $G$:

$$w(\mathcal{G}_{(+,\vec{s},+)}) = \prod_{\mu=1}^{n} w(\mathcal{M}_\mu) \prod_{\mu \in F_1} \frac{\overleftrightarrow{w}(\mathcal{M}_\mu)}{w(\mathcal{M}_\mu)} \prod_{\mu \in F_2} \frac{\widehat{w}(\mathcal{M}_\mu)}{w(\mathcal{M}_\mu)}, \tag{83}$$

where $F_1$ is the set of projectors in the sequence to be implemented by forced measurements of the first type and $F_2$ is the set of projectors in the sequence to be implemented by forced measurements of the second type.

In this way, the optimization analysis when using forced-measurement methods is still processed via projector sequences.

## 5.2 Gate search

**Single-qubit gates**  For single-qubit gates, we first determine which sequences of measurements/projectors are valid, i.e. which sequences of $\mathcal{M}_\mu$ do not collapse the computational state. (For single-qubit gates, this is irrespective of the corresponding projection channels $s_\mu$ at each step.) As discussed in Sec. 2.2, valid single-qubit measurement sequences have the constraint that consecutive 2-MZM measurements must have exactly one

MZM in common, so each measurement step involves choosing one MZM from the previous measurement pair and one from the four remaining MZMs, leading to 8 possible measurements to choose from. The $n$th measurement in the sequence is fully constrained, as it must be of the ancillary pair of MZMs (3,4). The penultimate measurement is also constrained, as it must involve one MZM from the antepenultimate measurement pair and one from the ancillary pair (3,4), leading to 4 possible choices for the penultimate measurement. Thus, the size of the search space for single-hexon measurement sequences of length $n$ is $2^{3n-4}$. Even though this scaling is exponential in $n$, we are able to consider sufficiently long sequences for all single-qubit gates in order to determine their optimal measurement-only sequences.

Once we have determined which measurement/projector sequences are valid, we evaluate the resulting computational gates $G_{(s_n, \vec{s}, s_0)}$ for all possible measurement outcomes/projection channels $s_\mu$.

For the single-qubit gates, when forced-measurement methods are being utilized, we can determine the minimal set of projectors that need to be forced using the tools developed in Sec. 3.3. Moreover, when Majorana-Pauli tracking methods are being utilized, the same methods allow us to determine the overall Pauli gate correction.

In the following, we searched up to $n = 9$ and found that the lowest weight sequences occur at $n \leq 5$. For the estimated weight factors used in this paper, this constitutes an exhaustive search for the minimal weight single-qubit gates, because longer sequences for the same gates are guaranteed to have larger difficulty weight values. In other words, we have found the globally optimal measurement-only implementations of the single-qubit gates.

**Two-qubit gates**   The set of two-qubit Clifford gates has 11,520 elements or 720 Pauli cosets, making it impractical to report optimal sequences for each element. For the purpose of this paper, we focus on controlled-Pauli gates $\{C(X), C(Y), C(Z)\}$ and these will be the only two-qubit gates with respect to which we optimize the labeling configurations. We then also report results for the $W$ and SWAP gates within these labeling configurations.

As discussed in Sec. 2.3, valid measurements on two-hexons must be of operators that anticommute with at least one stabilizer of the code. There are thus 16 valid 2-MZM measurements and 176 valid 4-MZM measurements at every step. As before, the final set of stabilizers must match the initial one. This condition fixes the final measurement and also partially fixes the penultimate measurement in a sequence. Thus, the search space is roughly $176^k 16^{n-k-1}$ depending on the number $k$ of 4-MZM measurements involved in a sequence and their placements. In addition to helping reduce the size of the search space, limiting the number of 4-MZM projectors can be physically motivated by the assumption that they would typically be significantly more costly than 2-MZM measurements.

We have carried out such a search up to length $n = 5$ and included at most $k = 3$ 4-MZM measurements. For each measurement sequence that compiles to $C(X), C(Y), C(Z)$ up to a Pauli operator, we calculate its difficulty weight and compare it with other sequences that realize these gates up to Pauli cosets, recording the minimal weight sequence found for each labeling configuration.

Within each $C(P)$ optimized labeling configuration we also search for minimal weight sequences compiling to $W$ and SWAP. Note that certain 4-MZM measurements are not possible in the one-sided geometries. In these cases, gates such as SWAP require longer measurement sequences though, at the same time, the restriction allows us to search to a greater length $n = 6$.

Both minimal weight forced-measurement sequences and minimal weight tracked-measurement sequences for $C(P)$ gates were found at $n = 4$ involving only a single

4-MZM measruement. No $C(P)$ gates were found for sequences of length $n < 4$ though $W$ gates occur at $n = 3, k = 1$. A naïve compilation for SWAP is SWAP $= C(X)_{12} C(X)_{21} C(X)_{12}$ which in the present compilation would require three 4-MZM measurements. However, our search reveals a more direct compilation for SWAP requiring only two 4-MZM measurements.

In the case of two-qubit gates, we do not have a simple way of determining the Pauli corrections or which projectors need to be forced, so we use a brute-force method. In particular, for a given measurement sequence that can realize a desired gate, we evaluate the sequence using all possible measurement/projector channels $s_\mu$. This immediately gives the Pauli correction gate for Majorana-Pauli tracking, and can be used to determine which projectors need to be forced when using forced-measurement methods. This is done by first grouping together projector sequences that yield the same gate. For each such set of projector sequences, we first check which projector channels $s_\mu$ are the same across all elements of the set; these projectors must be forced. We then look for correlations between the remaining measurement outcomes, which may require further forced measurements. We start from the first projector that does not have fixed projection channel, which we denote as $s_\nu$, and consider separately the subsets of projector sequences where this outcome is $s_\nu = +1$ or $-1$. Within each subset, we check if any subsequent measurement has fixed outcome; if so, it must be forced onto a channel that is correlated with $s_\nu$, and if not we recursively apply the procedure to this measurement. For example, we find that the measurement sequence $\Pi_+^{(34)}\Pi_{s_3}^{(35)}\Pi_{s_2}^{(56)}\Pi_{s_1}^{(35;1'6')}$ compiles to $C(X)$ exactly when $s_2 = +$ and $s_3 = s_1$, i.e. the following two projector sequences yield the same gate:

$$\Pi_+^{(34)}\Pi_+^{(35)}\Pi_+^{(56)}\Pi_+^{(35;1'6')}, \tag{84a}$$

$$\Pi_+^{(34)}\Pi_-^{(35)}\Pi_+^{(56)}\Pi_-^{(35;1'6')}, \tag{84b}$$

which indicates that the $\mu = 2, 3, 4$ projectors need to be forced.

## 5.3 Demonstration of Methods

We now demonstrate the use of our methods for the various cases of interest. For the purposes of producing a quantitative demonstration, we will very roughly estimate the difficulty weight factors to be: $w_c = 1.25$, $w_t = 1.65$, $w_a = 1.01$, and $f(N) = (\prod_{n=1}^{N} n!)^{(N-1)!}$. The results obtained for these weight factor values should not be misconstrued as being universal. For practical applications, the analysis will need to be performed again using weight factors that are more accurately estimated from experiments on the physical system being utilized.

We note that multiple measurement-only sequences may yield the same computational gate with the same difficulty weight. When this is the case for minimal weight sequences, we only present one representative of the set of minimal weight sequences for a gate or Pauli class. Similarly, multiple MZM labeling configurations may yield equally optimal minimal difficulty weights for the relevant gates, and we will only present one of the optimal configurations.

For two-sided hexon architectures, in both the case of using forced-measurement methods and the case of using Majorana-Pauli tracking methods, we find that the MZM labeling configuration $\langle 3, 4, 1, 2, 6, 5 \rangle$ yields the optimal results within our search for each of the following gates or gate sets, independently: the single-qubit Hadamrd gate, the geometric average of all single-qubit Clifford gates, the geometric average of $C(X)$ acting in all four directions, and in the geometric average over all $C(P)$ gates in all four directions.

For one-sided hexon architectures, in the case of using forced-measurement methods, we find that: (a) the MZM labeling configuration $\langle 1, 2, 6, 3, 4, 5 \rangle$ yields the optimal results

Table 1: Optimal MZM labeling configurations for two-sided $\langle 3,4,1,2,6,5\rangle_2$ and one-sided hexon architectures $\langle 1,2,6,3,4,5\rangle_1, \langle 3,4,1,2,6,5\rangle_1$ when using forced-measurement methods. The difficulty weights or geometric average of weights are reported for the gates: the Hadamard gate $H$, the set of single-qubit Clifford gates $C_1$, the controlled-not gate $C(X)$, the set of controlled-Pauli gates $C(P)$, the $W$ gate, and the SWAP gate. The weights of the two-qubit gates are averaged over the four connectivity directions.

| Configuration | $H$ | $C_1$ | $C(X)$ | $C(P)$ | $W$ | SWAP |
|---|---|---|---|---|---|---|
| $\langle 3,4,1,2,6,5\rangle_2$ | $1.39\times10^8$ | $7.72\times10^6$ | $8.10\times10^8$ | $7.78\times10^8$ | $3.81\times10^6$ | $1.50\times10^{12}$ |
| $\langle 1,2,6,3,4,5\rangle_1$ | $9.99\times10^5$ | $1.45\times10^5$ | $2.85\times10^8$ | $3.02\times10^8$ | $3.92\times10^6$ | $2.95\times10^{14}$ |
| $\langle 3,4,1,2,6,5\rangle_1$ | $9.99\times10^5$ | $1.89\times10^5$ | $2.69\times10^8$ | $2.39\times10^8$ | $6.25\times10^6$ | $5.93\times10^{14}$ |

Table 2: Optimal MZM labeling configurations for two-sided $\langle 3,4,1,2,6,5\rangle_2$ and one-sided hexon architectures $\langle 1,2,6,3,4,5\rangle_1, \langle 3,4,1,2,6,5\rangle_1$ when using Majorana-Pauli tracking methods. The difficulty weights or geometric average of weights are reported for the Pauli cosets of gates: the Hadamard gate $H$, the set of single-qubit Clifford gates $C_1$, the controlled-not gate $C(X)$, the set of controlled-Pauli gates $C(P)$, the $W$ gate, and the SWAP gate. The weights of the two-qubit gates are averaged over the four connectivity directions.

| Configuration | $[H]$ | $[C_1]$ | $[C(X)]$ | $[C(P)]$ | $[W]$ | $[\text{SWAP}]$ |
|---|---|---|---|---|---|---|
| $\langle 3,4,1,2,6,5\rangle_2$ | $1.76\times10^2$ | $5.44\times10^2$ | $4.20\times10^3$ | $4.20\times10^3$ | $1.05\times10^3$ | $7.42\times10^4$ |
| $\langle 1,2,6,3,4,5\rangle_1$ | $5.13\times10^1$ | $9.66\times10^1$ | $4.25\times10^3$ | $4.42\times10^3$ | $1.10\times10^3$ | $1.18\times10^6$ |
| $\langle 3,4,1,2,6,5\rangle_1$ | $8.17\times10^1$ | $1.16\times10^2$ | $4.78\times10^3$ | $4.59\times10^3$ | $1.39\times10^3$ | $1.48\times10^6$ |

for the Hadamard gate and the geometric average of all single-qubit Clifford gates; (b) the MZM labeling configuration $\langle 3,4,1,2,6,5\rangle$ yields the optimal results within our search for the geometric average over $C(X)$ acting in all four directions and the geometric average over all $C(P)$ gates in all four directions.

For one-sided hexon architectures, in the case of using Majorana-Pauli tracking methods, we find that the MZM labeling configuration $\langle 1,2,6,3,4,5\rangle$ yields the optimal results within our search for each of the following gates or gate sets, independently: the single-qubit Hadamrd gate, the geometric average of all single-qubit Clifford gates, the geometric average of $C(X)$ acting in all four directions, and in the geometric average over all $C(P)$ gates in all four directions.

In Table 1, we present a summary of the minimal difficulty weights of gates for the case when forced-measurement methods (including procrastination) are being utilized for the mentioned configurations. In Table 2, we present a summary of the minimal difficulty weights of Pauli cosets of gates for the case when Majorana-Pauli tracking methods are being utilized for the mentioned configurations. Details of the measurement-only sequences and corresponding difficulty weights for the specific gates or Pauli cosets of gates can be found in Appendix C. We also provide the detailed Pauli gate corrections that arise for the presented optimal measurement-only gate sequences when using Majorana-Pauli tracking methods.

## 5.4 Comparative Analysis

The methods in this paper can be used to compare different approaches and architectures to determine preferences between them. Here, we discuss some of the comparative analyses that can be made.

**Measurements: forced vs. tracked**   It is clear without a detailed analysis that utilizing the Majorana-Pauli tracking methods will be more efficient than utilizing forced-measurement methods. Our optimization analysis serves to more precisely quantify the difference, when such a comparison is desired. This can be done for our demonstration by comparing the results in Tables 1 and 2, which exhibit substantial benefit for using tracking methods.

**Scalable architectures: one-sided hexons vs. two-sided hexons**   It will be important to eventually determine which scalable architectures are preferable. Our methods can help this assessment, once sufficient experimental data is collected for all architectures under consideration to provide an accurate comparison between the different options. (We emphasize that the difficulty weight factors $w_c$, $w_t$, $w_a$, and $f(N)$ might even differ between different architectures.) An important aspect of this comparison is also knowing how the quantum computing device will be utilized, i.e. which gates are relevant to the optimization problem. This can already be observed in the results of our demonstration (with the caution that the speculative weight factors were assumed to be identical for one-sided and two-sided hexon architectures). For example, in the case where tracking methods are utilized, Table 2 shows that the one-sided hexon architecture has a notable advantage for single-qubit gates, but that the two-sided hexon architecture has a slight advantage for controlled-Pauli gates and a major advantage for SWAP gates.

**Measurement-only gate synthesis: measurements vs. gates/braids**   The primary premise of this paper is that, for measurement-only topological quantum computing, there will be a significant benefit by optimizing gate synthesis with respect to the physical measurements, rather than optimizing with respect to a generating set of gates or braiding operators, each of which is implemented through a measurement-only sequence. In order to make this benefit quantitative, we perform a similar analysis using the "natural" generating set of Clifford gates $\langle S, H, C(Z) \rangle$ or braiding gates $\langle S, B, W \rangle$, where the difficulty weights of these generators are determined by their optimal measurement-only sequence realizations. The detailed comparison is presented in the tables in Appendix C. Here, we summarize the comparison for the case where Majorana-Pauli tracking methods are used for two-sided hexon architectures in Table 3 and for one-sided hexon architectures in Table 4. The benefit is even more dramatic when forced-measurement methods are utilized.

Table 3: Difficulty weights of Pauli cosets of gates for the case where Majorana-Pauli tracking methods are utilized for two-sided hexon architectures with the $\langle 3, 4, 1, 2, 6, 5 \rangle$ labeling configuration. We compare the weights for gates synthesized from the generating sets of operations given by MZM measurements $\langle \Pi_s^{(jk)}, \Pi_s^{(jk;l'm')} \rangle$, Clifford gates $\langle S, H, \mathsf{C}(Z) \rangle$, or braiding operations $\langle S, B, W \rangle$, respectively.

| Generating set | $[H]$ | $[\mathsf{C}_1]$ | $[\mathsf{C}(X)]$ | $[\mathsf{C}(P)]$ | $[\mathrm{SWAP}]$ | $[W]$ |
|---|---|---|---|---|---|---|
| $\langle \Pi_s^{(jk)}, \Pi_s^{(jk;l'm')} \rangle$ | $1.76 \times 10^2$ | $5.44 \times 10^2$ | $4.20 \times 10^3$ | $4.20 \times 10^3$ | $7.42 \times 10^4$ | $1.05 \times 10^3$ |
| $\langle S, H, \mathsf{C}(Z) \rangle$ | $1.76 \times 10^2$ | $1.10 \times 10^4$ | $1.30 \times 10^8$ | $1.30 \times 10^8$ | $2.21 \times 10^{24}$ | $1.30 \times 10^8$ |
| $\langle S, B, W \rangle$ | $6.88 \times 10^6$ | $1.33 \times 10^4$ | $4.98 \times 10^{16}$ | $1.38 \times 10^{12}$ | $1.23 \times 10^{50}$ | $1.05 \times 10^3$ |

Table 4: Difficulty weights of Pauli cosets of gates for the case where Majorana-Pauli tracking methods are utilized for one-sided hexon architectures with the $\langle 1, 2, 6, 3, 4, 5 \rangle$ labeling configuration. We compare the weights for gates synthesized from the generating sets of operations given by MZM measurements $\langle \Pi_s^{(jk)}, \Pi_s^{(jk;l'm')} \rangle$, Clifford gates $\langle S, H, \mathsf{C}(Z) \rangle$, or braiding operations $\langle S, B, W \rangle$, respectively.

| Generating set | $[H]$ | $[\mathsf{C}_1]$ | $[\mathsf{C}(X)]$ | $[\mathsf{C}(P)]$ | $[\mathrm{SWAP}]$ | $[W]$ |
|---|---|---|---|---|---|---|
| $\langle \Pi_s^{(jk)}, \Pi_s^{(jk;l'm')} \rangle$ | $5.13 \times 10^1$ | $9.66 \times 10^1$ | $4.25 \times 10^3$ | $4.42 \times 10^3$ | $1.18 \times 10^6$ | $1.10 \times 10^3$ |
| $\langle S, H, \mathsf{C}(Z) \rangle$ | $5.13 \times 10^1$ | $1.20 \times 10^3$ | $1.12 \times 10^7$ | $1.12 \times 10^7$ | $1.40 \times 10^{21}$ | $1.12 \times 10^7$ |
| $\langle S, B, W \rangle$ | $1.70 \times 10^5$ | $1.44 \times 10^3$ | $3.19 \times 10^{13}$ | $1.04 \times 10^{10}$ | $3.25 \times 10^{40}$ | $1.10 \times 10^3$ |

# 6 Example: stabilizer measurements of the surface code

## 6.1 Overview and motivation

An important class of error correcting codes are stabilizer codes, which we have briefly mentioned in the Introduction. In error correcting codes, the *logical* qubit state is encoded into a carefully chosen subspace of the Hilbert space of many *physical* qubits. In the case of stabilizer codes, this subspace is defined as the simultaneous +1 eigenspace of some number of commuting multi-qubit Pauli operators, referred to as the stabilizers. Errors are detected by repeatedly measuring the stabilizers; deviations from the expected outcome of +1 indicate errors.

Within the class of stabilizer codes, the surface code [18] is one of the most promising proposals for large-scale error correction. The simplest realization is defined on a rectangular lattice of qubits, whose plaquettes are divided into two sublattices in a checkerboard pattern. There is one stabilizer for each plaquette: for one sublattice, it is given by the product of the four Pauli $X$ operators of the data qubits around a plaquette; for the other sublattice, it is given by the corresponding product of four Pauli $Z$ operators.

Since the measurement of Pauli operators translates into topologically protected parity measurements in the MZM-based architectures discussed in this paper, Pauli stabilizer codes map ideally onto such architectures. Some of the ideas for using these architectures build upon those of Ref. [27], which suggested an implementation of a particular stabilizer code using MZMs, but relies on a physical 8-MZM measurement involving four neighbor-

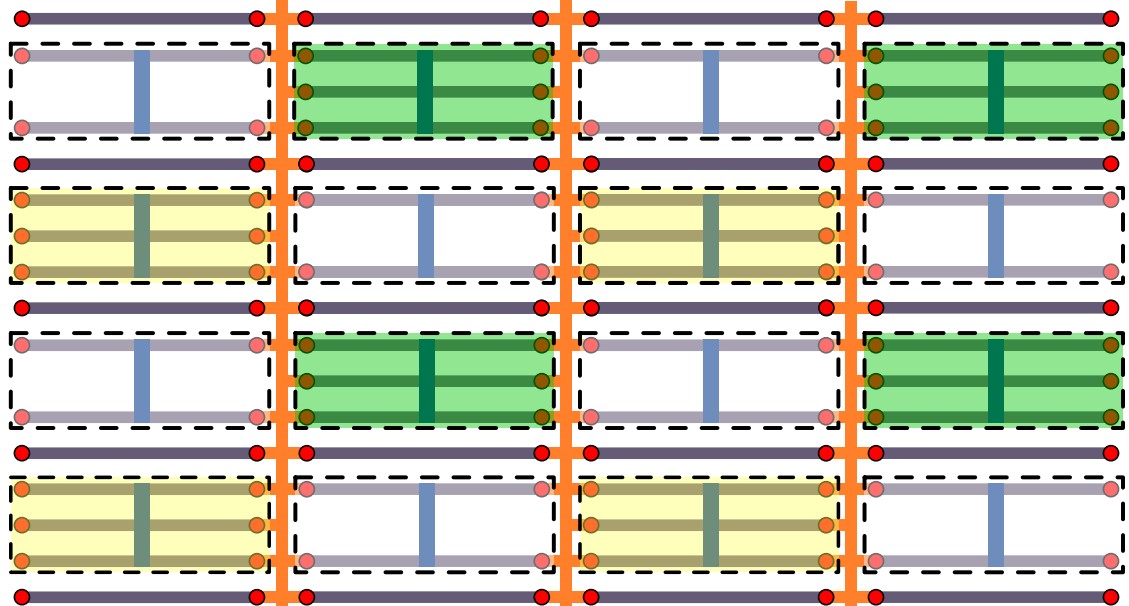

Figure 6: A proposed architecture layout for implementing a surface code. The tetrons (shown enclosed in unshaded dashed rectangles) play the role of data qubits in the surface code, while hexons (shown enclosed in shaded dashed rectangles) play the role of ancillary qubits used to facilitate the stabilizer measurement. The yellow and green shading of rectangles correspond to the $M_X$ and $M_Z$ hexons, respectively, which facilitate measuring the $X^{\otimes 4}$ and $Z^{\otimes 4}$ stabilizers on their nearest-neighboring data qubits. Coherent links (shown between vertical neighboring islands) are a necessary aid to enable the full set of Pauli measurements.

ing topological islands. This approach was generalized in Ref. [28], which however still relies on higher-weight measurements for the implementation of stabilizers. Since such measurements are likely to be prohibitively difficult to implement, it is worth seeking a MZM-based surface code implementation that utilizes physical measurements involving at most 4 MZMs (two topological islands) at a time.

In most practical proposals for implementing the surface code, the measurement of the product of four Pauli operators is achieved by adding an additional ancillary qubit to each plaquette, entangling it in a particular way with its adjacent data qubits, and finally performing a single-qubit measurement on the ancillary qubit. In this section, we propose a specific MZM-based architecture layout, sketched in Fig. 6, that can be used to implement precisely such a scheme in an efficient and topologically protected fashion. The required measurements are all 2-MZM or 4-MZM measurements on single and nearest-neighbor islands, respectively, and are natural to carry out in the architecture. We will use the techniques for optimizing compilations introduced in Sec. 5 as well as App. B to obtain an optimized measurement sequence implementing the stabilizer measurements.

The proposed architecture makes use of an additional MZM-based qubit design referred to as tetron. As opposed to a hexon, which has 6 MZMs on a single island, a tetron has 4 MZMs on a single island. Therefore, its state space in a fixed total parity sector of the island is two-dimensional instead of the four-dimensional state space of a hexon. As such, it does not accommodate a data and auxiliary qubit and, thus, cannot be used to perform Clifford operations on its own. However, pairs of tetrons together can be used for such an end, as discussed in Ref. [9]. We will see that for the purpose of implementing a surface code, a

mixed architecture of tetron and hexon islands is sufficient and has certain advantages. In our proposal, tetrons will play the role of data qubits in the surface code, while the hexons are used as ancillary qubits that facilitate unitary operations and the implementation of the $X^{\otimes 4}$ and $Z^{\otimes 4}$ stabilizer measurements. In order to avoid confusion between the term "ancillary qubit" used in reference to the second qubit encoded within a hexon and in reference to the qubits used to facilitate stabilizer measurements in the surface code, we will refer to the ancillary qubits of the surface code explicitly as ancillary hexons (or, when generalizing to tetrons or hexons, as ancillary islands). We refer to the ancillary hexons that facilitate measurements of the $X^{\otimes 4}$ stabilizers as "$M_X$-hexons" and the ones that facilitate measurements of the $Z^{\otimes 4}$ stabilizers as "$M_Z$-hexons."

It is worth pointing out that the tetrons can trivially be replaced by hexons in the proposed architecture; one can simply ignore the extra degrees of freedom, or perhaps utilize them in beneficial way. Depending on how logical gate operations are performed, it may be favorable to utilize hexons for the data qubits. For example, if transversal gates are used, the ability of hexons to perform single-qubit Clifford gates using only 2-MZM measurements may be useful.

## 6.2 Measurement circuit and example compilation

The two surface code stabilizers are measured as follows.

For the $X^{\otimes 4}$ stabilizers, the protocol is:

1. Initialize a $M_X$-hexon qubit into the $|X = +1\rangle$ state, with the hexon's ancillary qubit in an arbitrary, but definite state (i.e. into a $|i\gamma_1\gamma_6 = +, i\gamma_3\gamma_4 = p_{34}\rangle$ state).

2. Apply the sequence of CNOTs: $C(X)_{(h_x, t_4)}C(X)_{(h_x, t_3)}C(X)_{(h_x, t_2)}C(X)_{(h_x, t_1)}$, controlled on the $M_X$-hexon (labeled $h_x$) and targeting the four nearest-neighboring tetrons (labeled $t_j$).

3. Measure the $M_X$-hexon qubit in the $X$-basis (i.e. measure $i\gamma_1\gamma_6$).

The effect of this sequence of steps is a measurement of $X^{\otimes 4}$ of the four data tetrons. The outcome of the final measurement (in step 3), is the outcome of this stabilizer measurement.

For the $Z^{\otimes 4}$ stabilizers, the protocol is:

1. Initialize a $M_X$-hexon qubit into the $|0\rangle$ ($Z = +1$) state, with the hexon's ancillary qubit in an arbitrary, but definite state (i.e. into a $|i\gamma_1\gamma_2 = +, i\gamma_3\gamma_4 = p_{34}\rangle$ state).

2. Apply the sequence of CNOTs: $C(X)_{(t_4, h_z)}C(X)_{(t_3, h_z)}C(X)_{(t_2, h_z)}C(X)_{(t_1, h_z)}$, controlled on the four nearest-neighboring tetrons (labeled $t_j$) and targeting the $M_Z$-hexon (labeled $h_z$).

3. Measure the $M_Z$-hexon in the $Z$-basis (i.e. measure $i\gamma_1\gamma_2$).

The effect of this sequence of steps is a measurement of $Z^{\otimes 4}$ of the four data tetrons. The outcome of the final measurement (in step 3), is the outcome of this stabilizer measurement.

In order to accomplish this as efficiently as possible, we search for optimized compilations of these circuits. Since steps 1 and 3 are simply measurements (two needed for step 1 and one for step 3), they leave no room for optimizing. Thus, we need only focus on step

2, and search for optimal measurement sequences realizing the two sequences of CNOT gates, which we denote as

$$
\begin{aligned}
\mathsf{L}_X &= \mathsf{C}(X)_{(h_x,t_4)}\mathsf{C}(X)_{(h_x,t_3)}\mathsf{C}(X)_{(h_x,t_2)}\mathsf{C}(X)_{(h_x,t_1)} & (85) \\
\mathsf{L}_Z &= \mathsf{C}(X)_{(t_4,h_z)}\mathsf{C}(X)_{(t_3,h_z)}\mathsf{C}(X)_{(t_2,h_z)}\mathsf{C}(X)_{(t_1,h_z)}. & (86)
\end{aligned}
$$

The search space for a system of one hexon and four tetrons is prohibitively large for a brute-force search. Another way to proceed is by first finding measurement sequences compiling the individual CNOT gates $\mathsf{C}(X)_{(h,t)}$ and $\mathsf{C}(X)_{(t,h)}$ separately, use these to construct a full measurement-only circuit for $\mathsf{L}_X$ and $\mathsf{L}_Z$, and finally attempt to reduce the length of the sequence with the methods of Appendix B.

In this case, it is helpful to find measurement sequence compilations by identifying the stabilizers and logical operators in a system comprising a hexon and tetron, and updating them appropriately as a sequence of measurements is performed (as described in Sec. 2.3). If the set of stabilizers at the end of a sequence of measurements is the same as the initial set of stabilizers, the sequence will yield a logical gate that is determined by the transformation of the logical Pauli operators. A given measurement sequence will compile to the target gate $\mathsf{C}(X)_{(a,b)}$ if the logical Pauli operators transform the same way as they do under conjugation by $\mathsf{C}(X)_{(a,b)}$, that is

$$
\begin{array}{c}
X_a I_b \\
Z_a I_b \\
I_a X_b \\
I_a Z_b
\end{array}
\xrightarrow{\mathsf{C}(X)_{(a,b)}}
\begin{array}{c}
X_a X_b \\
Z_a I_b \\
I_a X_b \\
Z_a Z_b
\end{array} . \tag{87}
$$

Recall from Sec. 2.3 that a hexon encodes one logical qubit in six MZMs and is stabilized by the total parity of the island $i^3\gamma_1\gamma_2\gamma_3\gamma_4\gamma_5\gamma_6 = +1$ and restricted to a further ancillary parity sector, which we choose to initialize as $i\gamma_3\gamma_4 = p_{34} = \pm 1$. The set of generators for the initial hexon stabilizer group is therefore $\mathsf{S}_{\text{hex}} = \langle i^3\gamma_1\gamma_2\gamma_3\gamma_4\gamma_5\gamma_6, i\gamma_3\gamma_4 \rangle$. The corresponding logical Pauli operators (acting on the logical qubit) for a hexon island are $\bar{X}_{\text{hex}} = [i\gamma_1\gamma_6]$, $\bar{Y}_{\text{hex}} = [-i\gamma_2\gamma_6]$, and $\bar{Z}_{\text{hex}} = [i\gamma_1\gamma_2]$, where the equivalence classes contain all parity operators related by multiplication by a stabilizer. The 2-MZM parity operators for hexons can be mapped back to Pauli operators via Eq. (7).

Similarly, a tetron encodes one logical qubit in four MZMs and is stabilized by the total parity of the island $i^2\gamma_1\gamma_2\gamma_3\gamma_4$. The stabilizer group is therefore $\mathsf{S}_{\text{tet}} = \langle i^2\gamma_1\gamma_2\gamma_3\gamma_4 \rangle$. The corresponding logical Pauli operators are $\bar{X}_{\text{tet}} = [i\gamma_1\gamma_4]$, $\bar{Y}_{\text{tet}} = [-i\gamma_2\gamma_4]$, and $\bar{Z}_{\text{tet}} = [i\gamma_1\gamma_2]$. The 2-MZM pairty operators for tetrons can be mapped back to Pauli operators via

$$
\begin{array}{lll}
i\gamma_1\gamma_2 = Z, & i\gamma_1\gamma_3 = Y, & i\gamma_1\gamma_4 = X, \\
& i\gamma_2\gamma_3 = X, & i\gamma_2\gamma_4 = -Y, \\
& & i\gamma_3\gamma_4 = Z.
\end{array} \tag{88}
$$

When a measurement of the operator $\Gamma_M$ is performed, the stabilizers and logical operators are updated according to the rules in Sec. 2.3. In discussing stabilizers for the purposes of gate synthesis, we can assume the total parity of each island is always fixed (this is only violated by quasiparticle poisoning errors that flip the parity of an island, which we neglect for the discussion in this paper), so the stabilizers corresponding to total island parity ($i^3\gamma_1\gamma_2\gamma_3\gamma_4\gamma_5\gamma_6 = +1$ for hexons and $i^2\gamma_1\gamma_2\gamma_3\gamma_4 = +1$ for tetrons) will be left implicit.

An example of a measurement sequence realizing $\mathsf{C}(X)_{(h,t)}$ is the following:

| Step | Measurement of | | Stabilizer | | $\bar{X}_{\text{hex}}\bar{I}_{\text{tet}}$ | | $\bar{Z}_{\text{hex}}\bar{I}_{\text{tet}}$ | | $\bar{I}_{\text{hex}}\bar{X}_{\text{tet}}$ | | $\bar{I}_{\text{hex}}\bar{Z}_{\text{tet}}$ | |
|---|---|---|---|---|---|---|---|---|---|---|---|---|
| 0 | — | | 34 | ∘∘ | 16 | ∘∘ | 12 | ∘∘ | ∘∘ | 14 | ∘∘ | 12 |
| 1 | 46 | 14 | 46 | 14 | 25 | ∘∘ | 12 | ∘∘ | ∘∘ | 14 | 34 | 12 |
| 2 | 56 | ∘∘ | 56 | ∘∘ | 13 | 14 | 12 | ∘∘ | ∘∘ | 14 | 34 | 12 |
| 3 | 46 | ∘∘ | 46 | ∘∘ | 13 | 14 | 12 | ∘∘ | ∘∘ | 14 | 12 | 12 |
| 4 | 34 | ∘∘ | 34 | ∘∘ | 25 | 14 | 12 | ∘∘ | ∘∘ | 14 | 12 | 12 |
| | | | | | $\bar{X}_{\text{hex}}\bar{X}_{\text{tet}}$ | | $\bar{Z}_{\text{hex}}\bar{I}_{\text{tet}}$ | | $\bar{I}_{\text{hex}}\bar{X}_{\text{tet}}$ | | $\bar{Z}_{\text{hex}}\bar{Z}_{\text{tet}}$ | |

We use the shorthand ab$|$cd to mean $(i\gamma_a\gamma_b)_{\text{hex}} \otimes (i\gamma_c\gamma_d)_{\text{tet}}$ and ∘∘ to mean that the corresponding hexon or tetron is not involved. As mentioned, the overall island parity stabilizers are left implicit, since they are assumed to be fixed throughout the process. Furthermore, we do not explicitly account for signs in the stabilizers or logical operators. For example, $(i\gamma_1\gamma_2)(i\gamma_1\gamma_3) = -i\gamma_2\gamma_3$, but would be recorded as 23. The effect of these signs is to alter the compiled gate by an overall Pauli operator, which can be determined by Pauli tracking, as discussed in Sec. 4.

We see that the effect of this measurement sequence is to apply a $C(X)_{(h,t)}$ gate controlled on the hexon and targeting a tetron, up to a Pauli operator. We can build up the full $L_X$ circuit by concatenating variations of this circuit for each of the four tetrons. Then we can improve the efficiency by using the sequence manipulation and reduction tools developed in Appendix B. The same can be done for $C(X)_{(t,h)}$ gates and $L_Z$ circuits.

More specifically, we know that reversing a measurement sequence yields the inverse of the compiled gate. Since $C(X)^\dagger = C(X)$, we can freely reverse the corresponding measurement sequence (we assume an initialization of $i\gamma_3\gamma_4$, so all sequences implicitly start with a $i\gamma_3\gamma_4$ stabilizer that we leave implicit from now on)

$$
\begin{array}{cc|c}
46 & 14 \\
56 & \circ\circ \\
46 & \circ\circ \\
34 & \circ\circ
\end{array}
\quad \overset{\text{reverse}}{\longleftrightarrow} \quad
\begin{array}{cc|c}
46 & \circ\circ \\
56 & \circ\circ \\
46 & 14 \\
34 & \circ\circ
\end{array} \; .
$$

Immediate repetitions of the same measurement can be reduced, since $\Pi_r^{(M)}\Pi_s^{(M)} = \delta_{r,s}\Pi_s^{(M)}$. Furthermore, triplets of measurements of $M_1$, $M_2$, and then $M_1$, where $\{\Gamma_{M_1}, \Gamma_{M_2}\} = 0$ can be reduced, since $\Pi_r^{(M_1)}\Pi_s^{(M_2)}\Pi_t^{(M_1)} \propto \left(\delta_{r,t} + s\Gamma_{M_2}\delta_{-r,t}\right)\Pi_t^{(M_1)}$ for such measurements.

A full $L_X$ circuit can then be compiled and reduced in the following way:

```
46 | 14 | oo | oo | oo        46 | 14 | oo | oo | oo
56 | oo | oo | oo | oo        56 | oo | oo | oo | oo
46 | oo | oo | oo | oo        46 | oo | oo | oo | oo
34 | oo | oo | oo | oo        34 | oo | oo | oo | oo
------------------------      ------------------------
46 | oo | 14 | oo | oo        46 | oo | oo | oo | oo        46 | 14 | oo | oo | oo
56 | oo | oo | oo | oo        56 | oo | oo | oo | oo        56 | oo | oo | oo | oo
46 | oo | oo | oo | oo        46 | oo | 14 | oo | oo        46 | oo | 14 | oo | oo
34 | oo | oo | oo | oo  →rev  34 | oo | oo | oo | oo  →red  34 | oo | oo | oo | oo
------------------------ blocks ------------------------    ------------------------
46 | oo | oo | 14 | oo  2,4   46 | oo | oo | 14 | oo        46 | oo | oo | 14 | oo
56 | oo | oo | oo | oo        56 | oo | oo | oo | oo        56 | oo | oo | oo | oo
46 | oo | oo | oo | oo        46 | oo | oo | oo | oo        46 | oo | oo | oo | 14
34 | oo | oo | oo | oo        34 | oo | oo | oo | oo        34 | oo | oo | oo | oo
------------------------      ------------------------
46 | oo | oo | oo | 14        46 | oo | oo | oo | oo
56 | oo | oo | oo | oo        56 | oo | oo | oo | oo
46 | oo | oo | oo | oo        46 | oo | oo | oo | 14
34 | oo | oo | oo | oo        34 | oo | oo | oo | oo
```
.

Here, the first column corresponds to the hexon and the next four columns correspond to each of the neighboring tetrons. This reduces the naïve length 16 measurement sequence

to a length 8 measurement sequence, where each tetron is involved in only a single 4-MZM measurement. We conjecture that this is the minimum number of measurements required to implement $L_X$. (It is clearly the minimum number of 4-MZM measurements required.)

The same steps can be applied to construct an optimized implementation of the $L_Z$ circuit. The starting point is a single $C(X)_{(t,h)}$, which can be implemented by

$$
\begin{array}{c|c}
14 & 12 \\
16 & \circ\circ \\
36 & \circ\circ \\
34 & \circ\circ
\end{array}
$$

Following the same steps as for the $L_X$ circuit, i.e. appropriately combining four $C(X)_{(t,h)}$ gates and reducing them yields the following implementation of $L_Z$:

$$
\begin{array}{c|c|c|c|c}
14 & 12 & \circ\circ & \circ\circ & \circ\circ \\
16 & \circ\circ & \circ\circ & \circ\circ & \circ\circ \\
14 & \circ\circ & 12 & \circ\circ & \circ\circ \\
34 & \circ\circ & \circ\circ & \circ\circ & \circ\circ \\
14 & \circ\circ & \circ\circ & 12 & \circ\circ \\
12 & \circ\circ & \circ\circ & \circ\circ & \circ\circ \\
14 & \circ\circ & \circ\circ & \circ\circ & 12 \\
34 & \circ\circ & \circ\circ & \circ\circ & \circ\circ
\end{array}
$$

This also reduces the naïve length 16 measurement sequence to a length 8 sequence, where each tetron is involved in only one 4-MZM measurement.

## 6.3 Circuit optimization

We can apply cost functions, such as the difficulty weight assignment scheme of Sec. 2.4, to find optimized encodings of hexons, tetrons and optimized $L_X$ and $L_Z$ circuit compilations, similar to the optimizations performed the previous section.

For the sequence optimization, we recognize that the $L_X$ and $L_Z$ circuits naturally divide into two segments, each of which involves two applications of $C(X)$ that can be manipulated as a pair and reduced. With this in mind, we first search for all length-4 measurement sequences that alternate between 4-MZM measurements and 2-MZM measurements (each 4-MZM measurement is pairing the hexon with a tetron in a different direction on the lattice, either upwards, rightwards, leftwards, or downwards) and which compile to $C(X)_{(h,t_j)}C(X)_{(h,t_k)}$ and $C(X)_{(t_j,h)}C(X)_{(t_k,h)}$, up to overall Pauli factors. There are 8 possible MZM pairs that can be chosen for the hexon for each measurement step along with $\binom{4}{2} = 6$ MZM pairs for the selected tetron. The search space for a 4-MZM, 2-MZM, 4-MZM, 2-MZM measurement sequence with the constraint that the final 2-MZM measurement is on $i\gamma_3\gamma_4$ of the hexon is therefore over $(8 \times 6) \times (7 \times 24 + 1 \times 48) = 10,368$ measurement-only sequences. For each pair of directions, $j$ and $k$, we find 64 sequences for $C(X)_{(h,t)}$, and similarly for $C(X)_{(t,h)}$. We then combine these to form measurement-only compilations of $L_X$ and $L_Z$. This produces a list of all $L_X$ and $L_Z$ circuits obtained through optimized compilations of $C(X)_{(h,t_j)}C(X)_{(h,t_k)}$ and $C(X)_{(t_j,h)}C(X)_{(t_k,h)}$. A search over all length-8 measurement sequences that alternate between 4-MZM and 2-MZM measurements has yet to be carried out; the search space in this case has is over $(48 \times 8)^2 \times 48 \times 9 \times 24 \times 1 = 1,528,823,808$ measurement-only sequences.

As in the case of hexons (see Sec. 2.5), the MZMs of tetrons may also be relabeled, reflecting a different encoding choice. We use the analogous notation of $\langle a, b, c, d \rangle$ to denote the labeling configuration of MZMs within a two-sided tetron where the labeling goes

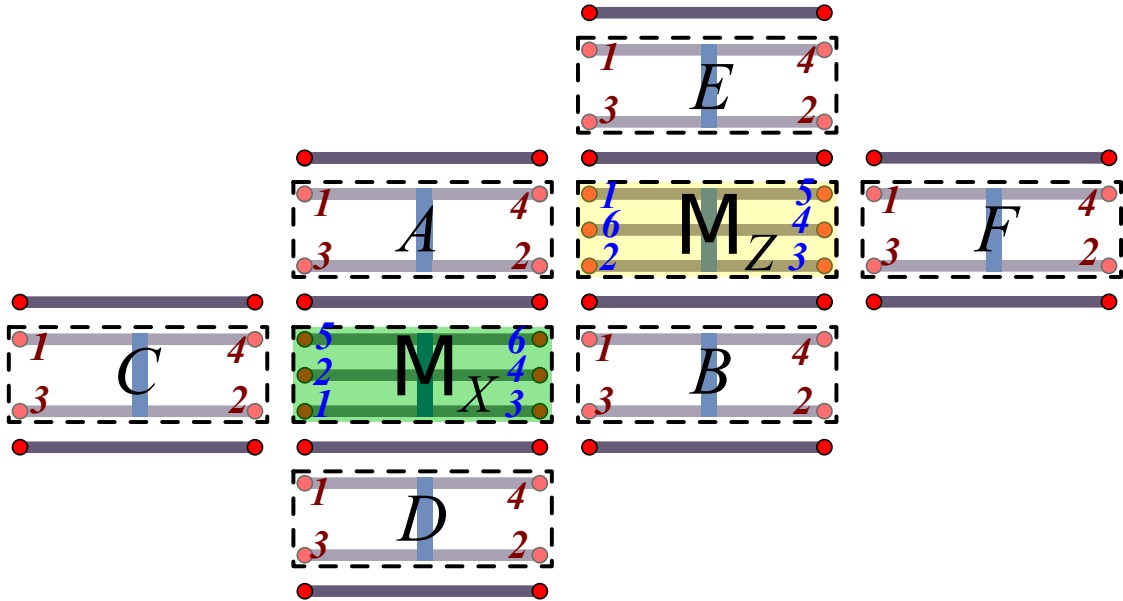

Figure 7: An example of an optimized labeling configuration for the proposed architecture. The tetrons (labeled $A, B, C, D, E, F$) all use the $\langle 1, 3, 2, 4 \rangle$ configuration, the $\mathsf{M}_X$ hexon (green) uses the $\langle 5, 2, 1, 3, 4, 6 \rangle$ configuration, and the $\mathsf{M}_Z$ hexon (yellow) uses the $\langle 1, 6, 2, 3, 4, 5 \rangle$ configuration.

counterclockwise from the top-left to the top-right. The next step in the optimization is the following: for each tetron labeling configuration, search over all hexon labeling configurations and record the lowest weight $\mathsf{L}_X$ sequence and the hexon configuration for which it is realized, and likewise for $\mathsf{L}_Z$. For each tetron labeling configuration, this gives a $\mathsf{M}_X$-hexon labeling configuration, $\mathsf{L}_X$ measurement sequence, and corresponding difficulty weight, as well as a $\mathsf{M}_Z$-hexon labeling configuration, $\mathsf{L}_Z$ measurement sequence and weight. Defining the tetron labeling configuration weight to be the geometric average of its $\mathsf{L}_X$ and $\mathsf{L}_Z$ weights, we can pick out the best configuration.

Doing this for the same choice of weights as in the previous section, we find eight tetron labeling configurations that have difficulty weight $1.67 \times 10^{10}$. This is clearly an improvement over the naïve concatenation of four $\mathsf{C}(X)$ measurement sequences, which has a total difficulty on weight on the order of $10^{14}$ (see Table 2). An example of an optimized labeling configuration is shown in Fig. 7, where the tetrons using the $\langle 1, 3, 2, 4 \rangle$ configuration, the $\mathsf{M}_X$ hexons using the $\langle 5, 2, 1, 3, 4, 6 \rangle$ configuration, and the $\mathsf{M}_Z$ hexons using the $\langle 1, 6, 2, 3, 4, 5 \rangle$ configuration. The associated optimal measurement sequences are

$$
\mathsf{L}_X =
\begin{array}{c|c|c|c|c|c|c|c}
\mathsf{M}_X & \mathsf{M}_Z & A & B & C & D & E & F \\
\hline
24 & \circ\circ & 23 & \circ\circ & \circ\circ & \circ\circ & \circ\circ & \circ\circ \\
12 & \circ\circ & \circ\circ & \circ\circ & \circ\circ & \circ\circ & \circ\circ & \circ\circ \\
13 & \circ\circ & \circ\circ & \circ\circ & \circ\circ & 14 & \circ\circ & \circ\circ \\
34 & \circ\circ & \circ\circ & \circ\circ & \circ\circ & \circ\circ & \circ\circ & \circ\circ \\
13 & \circ\circ & \circ\circ & 23 & \circ\circ & \circ\circ & \circ\circ & \circ\circ \\
12 & \circ\circ & \circ\circ & \circ\circ & \circ\circ & \circ\circ & \circ\circ & \circ\circ \\
13 & \circ\circ & \circ\circ & \circ\circ & 23 & \circ\circ & \circ\circ & \circ\circ \\
34 & \circ\circ & \circ\circ & \circ\circ & \circ\circ & \circ\circ & \circ\circ & \circ\circ \\
\end{array}
\;,\quad
\mathsf{L}_Z =
\begin{array}{c|c|c|c|c|c|c|c}
\mathsf{M}_X & \mathsf{M}_Z & A & B & C & D & E & F \\
\hline
\circ\circ & 13 & \circ\circ & \circ\circ & \circ\circ & \circ\circ & \circ\circ & 34 \\
\circ\circ & 16 & \circ\circ & \circ\circ & \circ\circ & \circ\circ & \circ\circ & \circ\circ \\
\circ\circ & 13 & 34 & \circ\circ & \circ\circ & \circ\circ & \circ\circ & \circ\circ \\
\circ\circ & 34 & \circ\circ & \circ\circ & \circ\circ & \circ\circ & \circ\circ & \circ\circ \\
\circ\circ & 14 & \circ\circ & \circ\circ & \circ\circ & \circ\circ & 12 & \circ\circ \\
\circ\circ & 16 & \circ\circ & \circ\circ & \circ\circ & \circ\circ & \circ\circ & \circ\circ \\
\circ\circ & 36 & \circ\circ & 12 & \circ\circ & \circ\circ & \circ\circ & \circ\circ \\
\circ\circ & 34 & \circ\circ & \circ\circ & \circ\circ & \circ\circ & \circ\circ & \circ\circ \\
\end{array}
\;,
$$

where the first column is the $\mathsf{M}_X$ hexon, the second column is the $\mathsf{M}_Z$ hexon, and the remaining columns correspond to tetrons $A, B, C, D, E, F$ as shown, for example, in Fig. 7.

## 6.4 Boundary circuits

The stabilizer measurements at the boundaries of a surface code will involve fewer data qubits than in the bulk, so we consider these for completeness. For the case of stabilizer measurements involving two data qubits, the tetrons may be measured directly, or via the sequence reduced $C(X)$ circuits studied above, or through the use of GHZ states as described in [28]. For the case of stabilizer measurements involving three data qubits, a direct measurement is hypothesized to be significantly difficult due to the number of islands involved. Further, the sequence reduction techniques utilized in the previous section are not applicable for this case. A brute-force search may, however, be performed for length six measurement sequences that alternate between 4-MZM and 2-MZM measurements. An example of a measurement sequence for applying three $C(X)_{(h,t)}$ operations (i.e. a three data qubit version of $\mathsf{L}_X$) found in this way is:

| 13 | 23 | ∘∘ | ∘∘ |
| 12 | ∘∘ | ∘∘ | ∘∘ |
| 15 | ∘∘ | 23 | ∘∘ |
| 56 | ∘∘ | ∘∘ | ∘∘ |
| 45 | ∘∘ | ∘∘ | 23 |
| 34 | ∘∘ | ∘∘ | ∘∘ |

# 7 Final Remarks

The methods introduced in this paper can be applied more generally to topological quantum computation with other non-Abelian anyons or defects. For example, the difficulty analysis and optimization can be applied for different and mixed architectures, such as tetron, octons, etc., systems with different topological orders, and to other measurement-based operations, such as the injection of non-Clifford gates.

The procrastination and tracking methods can only be applied when the measurement outcomes correspond to fusion channels that are Abelian [26, 29], e.g. for Ising anyons, MZMs, and Parafendleyons (parafermionic zero modes). When fusion channels may be non-Abelian, leaving the corresponding projectors in a measurement-only sequence of measurements will eventually lead to measurements extracting information regarding the computational state, (at least partially) collapsing it. Thus, when the measurement have non-Abelian fusion channels, one must use forced-measurement protocols to ensure all the projection channels are Abelian.

# Acknowledgements

We thank M. Beverland, N. Delfosse, J. Haah, T. Karzig, C. Knapp, D. Pikulin, and M. Silva for enlightening discussions.

# A  Example of Adaptive Forced-Measurement Protocol

For an example of the adaptive method described in Sec. 3.4, consider the following measurement-only sequence for compiling $C(Z)$ between horizontal neighboring one-sided

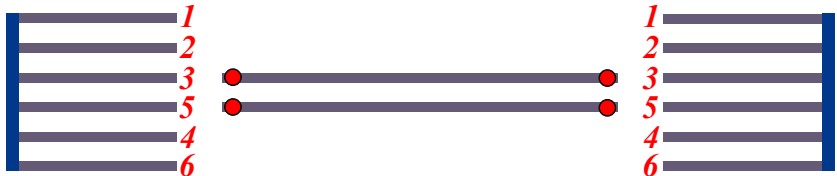

Figure 8: Two one-sided hexons in the MZM labeling configuration $\langle 1, 2, 3, 5, 4, 6 \rangle$.

hexons for the MZM labeling configuration $\langle 1, 2, 3, 5, 4, 6 \rangle$, as shown in Fig. 8:

$$\overset{\leftrightarrow}{\Pi}{}^{(34)}_{+} \overset{\leftrightarrow}{\Pi}{}^{(3'4')}_{+} \overset{\frown}{\Pi}{}^{(46)}_{-s_2 s_1} \overset{\leftrightarrow}{\Pi}{}^{(56)}_{+} \Pi^{(4'6')}_{s_2} \Pi^{(35;3'5')}_{s_1}. \tag{89}$$

This sequence has difficulty weight $1.44 \times 10^{11}$. Notice that in the forced measurement $\overset{\leftrightarrow}{\Pi}{}^{(56)}_{+}$, the 4-MZM measurement of $(35; 3'5')$ must be repeated when the undesired outcome of the (56)-measurement is obtained.

On the other hand, searching further out in the number of measurements, the measurement-only sequence

$$\overset{\leftrightarrow}{\Pi}{}^{(34)}_{+} \overset{\leftrightarrow}{\Pi}{}^{(3'4')}_{+} \overset{\leftrightarrow}{\Pi}{}^{(4'6')}_{s_3 s_2} \Pi^{(4'5')}_{s_5} \overset{\frown}{\Pi}{}^{(46)}_{-s_3 s_2 s_1} \Pi^{(56)}_{s_3} \Pi^{(4'6')}_{s_2} \Pi^{(35;3'5')}_{s_1}. \tag{90}$$

also compiles to $C(Z)$ and has difficulty weight $4.05 \times 10^{10}$. Notice that the forcing procedures in this alternative sequence no longer involve the 4-MZM measurement of $(35; 3'5')$.

Simply using this alternative sequence is already an improvement over the first, but it is possible to achieve better results by combining the two in a more complex way. Notice that the first three measurements in these two sequences are identical if $s_3 = +$. This suggests a more optimal protocol for synthesizing $C(Z)$ is: (1) Perform the first three measurements giving the projector sequence $\Pi^{(56)}_{s_3} \Pi^{(4'6')}_{s_2} \Pi^{(35;3'5')}_{s_1}$. (2a) If $s_3 = +$, finish the measurement sequence as in Eq.( 89). (2b) If $s_3 = -$, finish the measurement sequence as in Eq.( 90). This protocol gives a geometric average weight of $3.43 \times 10^9$ for the $C(Z)$ gate.

$$\begin{array}{c|c} \text{Sequence} & \text{Weight} \\ \hline \overset{\leftrightarrow}{\Pi}{}^{(34)}_{+} \overset{\leftrightarrow}{\Pi}{}^{(3'4')}_{+} \overset{\frown}{\Pi}{}^{(46)}_{-s_2 s_1} \Pi^{(56)}_{+} \Pi^{(4'6')}_{s_2} \Pi^{(35;3'5')}_{s_1} & 2.90 \times 10^8 \\ \overset{\leftrightarrow}{\Pi}{}^{(34)}_{+} \overset{\leftrightarrow}{\Pi}{}^{(3'4')}_{+} \overset{\leftrightarrow}{\Pi}{}^{(4'6')}_{-s_2} \Pi^{(4'5')}_{s_5} \overset{\frown}{\Pi}{}^{(46)}_{s_2 s_1} \Pi^{(56)}_{-} \Pi^{(4'6')}_{s_2} \Pi^{(35;3'5')}_{s_1} & 4.05 \times 10^{10} \\ \text{Average case} & 3.43 \times 10^9 \end{array} \tag{91}$$

In general, such adaptive protocols will be relevant whenever a projector sequence has a forced measurement that involves a particularly high weight measurement (for example, a 4-MZM measurement that uses coherent links) and extending the sequence removes having to repeat that costly measurement.

# B Sequence Morphology

In this section, we develop some tools to aid in the optimization of measurement-only gate compilation. We provide a method for generating alternate projector sequences for a specified gate from a given projector sequence and a method for generating projector sequences for all gates in the same conjugacy class as the specified gate. We also provide a protocol for reducing the lengths of projector sequences obtained through gate synthesis.

## B.1 Some general formulas

Recall that 2-MZM projectors are defined as $\Pi_s^{(jk)} = \frac{\mathbb{1} + si\gamma_j\gamma_k}{2}$ and obey the usual properties of complete orthogonal projectors

$$\Pi_s^{(jk)}\Pi_t^{(jk)} = \delta_{s,t}\Pi_s^{(jk)} \tag{92a}$$

$$\Pi_+^{(jk)} + \Pi_-^{(jk)} = \mathbb{1}. \tag{92b}$$

From the definition, it also follows that

$$\Pi_+^{(jk)} - \Pi_-^{(jk)} = i\gamma_j\gamma_k, \tag{93}$$

$$(i\gamma_j\gamma_k)\Pi_s^{(jk)} = \Pi_s^{(jk)}(i\gamma_j\gamma_k) = s\Pi_s^{(jk)}. \tag{94}$$

If $i\gamma_{a_1}\gamma_{a_2}$ and $i\gamma_{b_1}\gamma_{b_2}$ anti-commute with each other, we have the relations

$$\Pi_{s_2}^{(a_1 a_2)}\Pi_{s_1}^{(b_1 b_2)} = \Pi_{-s_1}^{(b_1 b_2)}\Pi_{s_2}^{(a_1 a_2)} + \frac{s_1}{2}i\gamma_{b_1}\gamma_{b_2} \tag{95a}$$

$$= \Pi_{s_1}^{(b_1 b_2)}\Pi_{-s_2}^{(a_1 a_2)} + \frac{s_2}{2}i\gamma_{a_1}\gamma_{a_2}. \tag{95b}$$

It follows that we can reduce the triplet of projections

$$\Pi_{s_3}^{(a_1 a_2)}\Pi_{s_2}^{(b_1 b_2)}\Pi_{s_1}^{(a_1 a_2)} = \begin{cases} \frac{1}{2}\Pi_{s_1}^{(a_1 a_2)} & \text{if } s_3 = s_1 \\ \frac{s_2}{2}(i\gamma_{b_1}\gamma_{b_2})\Pi_{s_1}^{(a_1 a_2)} & \text{if } s_3 = -s_1 \end{cases}, \tag{96}$$

when $i\gamma_{a_1}\gamma_{a_2}$ and $i\gamma_{b_1}\gamma_{b_2}$ anti-commute. Diagrammatically, this gives identities of the form

$$\begin{array}{c}\Pi_{s_3}^{(12)} \\ \Pi_{s_2}^{(23)} \\ \Pi_{s_1}^{(12)}\end{array} \quad\propto\quad \tag{97}$$

where the magenta line labeled by $s_\mu$ between a cup and a cap indicates a projector with unspecified projection channel $s_\mu = \pm 1$, and a magenta line labeled by $s_\mu$ connecting MZM lines $j$ and $k$ corresponds to the operator $(i\gamma_j\gamma_k)^{\frac{1-s_\mu}{2}}$.

In general, for multi-hexon MZM measurements $\Pi_s^{(\mathcal{M})}$

$$\Pi_s^{(\mathcal{M}_1)}\Pi_p^{(\mathcal{M}_2)}\Pi_r^{(\mathcal{M}_1)} = \Pi_s^{(\mathcal{M}_1)}\frac{\mathbb{1} + p\Gamma_{\mathcal{M}_2}}{2}\Pi_r^{(\mathcal{M}_1)}$$

$$= \begin{cases} \frac{1}{2}\Pi_r^{(\mathcal{M}_1)} & \text{if } s = r \\ \frac{p}{2}\Gamma_{\mathcal{M}_2}\Pi_r^{(\mathcal{M}_1)} & \text{if } s = -r, \end{cases} \tag{98}$$

whenever $\Gamma_{\mathcal{M}_1}$ anticommutes with $\Gamma_{\mathcal{M}_2}$, On the other hand, when $\Gamma_{\mathcal{M}_1}$ commutes with $\Gamma_{\mathcal{M}_2}$

$$\Pi_s^{(\mathcal{M}_1)}\Pi_p^{(\mathcal{M}_2)}\Pi_r^{(\mathcal{M}_1)} = \delta_{s,r}\Pi_s^{(\mathcal{M}_1)}\Pi_p^{(\mathcal{M}_2)} = \delta_{s,r}\Pi_p^{(\mathcal{M}_2)}\Pi_r^{(\mathcal{M}_1)}. \tag{99}$$

Additionally, we have the following identities (for $a$, $b$, $c$ all distinct)

$$\Pi_{s_3}^{(bc)}\Pi_{s_2}^{(ac)}\Pi_{s_1}^{(ab)} = \frac{1 + is_1 s_2 s_3}{2}\Pi_{s_3}^{(bc)}\Pi_{s_1}^{(ab)}, \tag{100a}$$

$$\Pi_{s_4}^{(ab)}\Pi_{s_3}^{(bc)}\Pi_{s_2}^{(ac)}\Pi_{s_1}^{(ab)} = \begin{cases} \frac{1+is_1 s_2 s_3}{4}\Pi_{s_1}^{(ab)} & \text{if } s_4 = s_1 \\ s_3(i\gamma_b\gamma_c)\frac{1+is_1 s_2 s_3}{4}\Pi_{s_1}^{(ab)} & \text{if } s_4 = -s_1 \end{cases}. \tag{100b}$$

The first follows from $i\gamma_a\gamma_c = i(i\gamma_b\gamma_c)(i\gamma_a\gamma_b)$ and the second follows from the first and Eq. (96). Diagrammatically, these relations take the form

$$\tag{101a}$$

$$\tag{101b}$$

## B.2 Sequence Morphology

Given a projection operator sequence $\mathcal{G}$ compiling to $G$, it is useful to develop tools for constructing alternate sequences compiling to $G$. This is because measuring certain MZMs may be easier than measuring others. Thus, we wish to come up with as many ways of obtaining a gate $G$ as possible so that we can then pick out the one that is easiest to implement. Furthermore, the ability to compile some gate $H$ when given a compilation for a different gate $G$ is beneficial for expanding our range of operations. This subsection details methodologies for both of these tasks.

For a system of $N$ hexons, we write a sequence of projection operators that compiles to the gate $G$ acting on the computational state space as

$$\mathcal{G} = \Pi_+^{(\text{anc})}\Pi_{s_{n-1}}^{(\mathcal{M}_{n-1})}\dots\Pi_{s_1}^{(\mathcal{M}_1)}\Pi_+^{(\text{anc})} \propto \Pi_+^{(\text{anc})}\otimes G, \tag{102}$$

where $\mathcal{M}_\mu$ are the sets of (up to $2N$) MZMs whose collective fermionic parity is being projected onto $s_\mu = \pm$. The first term in the tensor product acts on the ancillary qubits and the second term acts on the computational qubits. Given a projector sequence $\mathcal{G}$ that compiles to the target gate $G$, one can easily construct sequences that compile to the complex conjugate gate $G^*$, the inverse gate $G^{-1} = G^\dagger$, the transposed gate $G^T = G^{*\dagger}$, and nontrivial alternate sequences for $G$.

### B.2.1 Space-time reflections

By reversing a projector sequence, one generates a compilation for the inverse gate $G^{-1} = G^\dagger$, since the projectors are Hermitian, that is

$$\mathcal{G}^{\text{rev}} = \Pi_+^{(\text{anc})}\Pi_{s_1}^{(\mathcal{M}_1)}\dots\Pi_{s_{n-1}}^{(\mathcal{M}_{n-1})}\Pi_+^{(\text{anc})} \propto \Pi_+^{(\text{anc})}\otimes G^\dagger. \tag{103}$$

Diagrammatically, this can be seen by first applying $\mathcal{G}$ then $\mathcal{G}^\dagger$ and noting that the stacked diagram can be straightened out and fermion lines canceled, yielding the identity operator;

for example

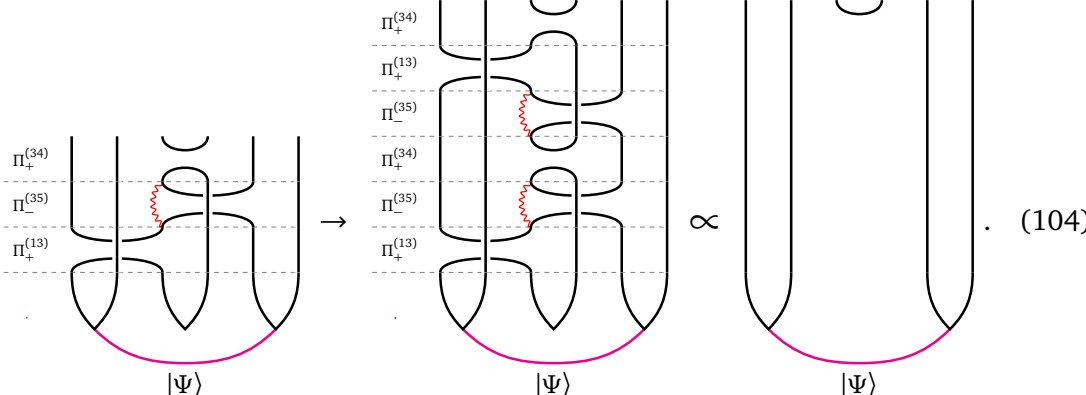

$$(104)$$

The complex conjugated gate can be constructed by complex conjugating each term in the projector sequence, as follows

$$\mathcal{G}^* = \Pi_+^{(\text{anc})^*} \Pi_{s_{n-1}}^{(\mathcal{M}_{n-1})^*} \dots \Pi_{s_1}^{(\mathcal{M}_1)^*} \Pi_+^{(\text{anc})^*} \propto \mathbf{\Pi}_+^{(\text{anc})} \otimes G^*. \tag{105}$$

In the choice of basis that we are using, complex conjugating the fermionic parity projectors has the effect of potentially changing which parity is being projected onto. In particular, we find that $\Pi_s^{(\mathcal{M})^*} = \Pi_{\pm s}^{(\mathcal{M})}$ when $\Gamma_{\mathcal{M}}^* = \pm\Gamma_{\mathcal{M}}$. In other words, when $\Gamma_{\mathcal{M}}$ is written as a tensor product of Pauli matrices, $\Pi_s^{(\mathcal{M})^*} = \Pi_s^{(\mathcal{M})}$ when the tensor product involves an even number of $Y$ matrices, and $\Pi_s^{(\mathcal{M})^*} = \Pi_{-s}^{(\mathcal{M})}$ when the tensor product involves an odd number of $Y$ matrices. It is straightforward to check that, in our choice of basis, the later occurs for a single hexon projector $\Pi_s^{(jk)}$ whenever $|j-k|$ is even. We emphasize that the action of complex conjugation is basis dependent.

A different nontrivial way to arrive at the complex conjugated gate is by creating the mirror image of the braid sequence. For example,

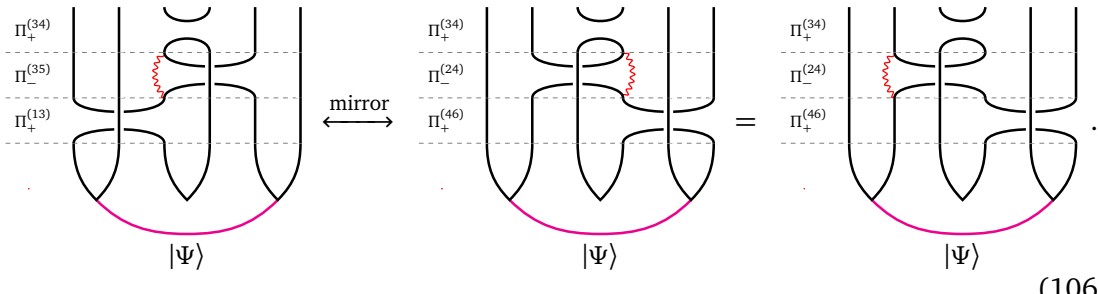

$$(106)$$

On the level of projector sequences, this is implemented for a single hexon via the "mirror-ing" operation $\Pi_s^{(jk)^M} = \Pi_s^{((7-k)(7-j))}$ applied to each projector in the sequence, where the MZMs are numbered $1, \dots, 6$ from left to right. We note that the ancillary qubit's projector is invariant under mirroring, i.e. $\Pi_+^{(34)^M} = \Pi_+^{(34)}$. Thus, we have

$$\mathcal{G}^M = \Pi_+^{(\text{anc})^M} \Pi_{s_{n-1}}^{(\mathcal{M}_{n-1})^M} \dots \Pi_{s_1}^{(\mathcal{M}_1)^M} \Pi_+^{(\text{anc})^M} \propto \mathbf{\Pi}_+^{(\text{anc})} \otimes G^*. \tag{107}$$

Generalizing to the $N$ hexon case, the mirroring operator can be applied to each hexon independently to generate $2^N$ (potentially) different projector sequences.

Combining the two ways of complex conjugating, we can construct an alternate compilation for a gate $G$, given $\mathcal{G}$. Specifically, we can first mirror the sequence to get a compilation $\mathcal{G}^M$ for $G^*$ and then we can complex conjugate each projector to get a projector

sequence $\mathcal{G}^{M*}$. The result compiles to the gate $G^{**} = G$. We dub this operation "mirror-conjugating" a sequence.

For example, both of the following projector sequences will compile to the same gate $ZH$

$$ZH\,|\Psi\rangle \propto \quad \begin{array}{c}\Pi_+^{(34)}\\ \Pi_-^{(35)}\\ \Pi_+^{(13)}\end{array} \qquad = \qquad \begin{array}{c}\Pi_+^{(34)}\\ \Pi_+^{(24)}\\ \Pi_-^{(46)}\end{array} \qquad . \tag{108}$$

### B.2.2 Paulimorphism

It follows from the definition of the Clifford group that, given a sequence $\mathcal{G}$ compiling to gate $G$, one can construct alternate compilations of $G$, as well as compilations for any other gate $G' \in \mathrm{Conj}_{C_N}(G)$ in the same conjugacy class as $G$. We define the (projective) stabilizer of a gate $G \in C_N$ to be

$$\mathrm{Stab}_{C_N}(G) = \{A \in C_N \,|\, AGA^{-1} = e^{i\phi}G\} \tag{109}$$

and the (projective) conjugacy class of $G \in C_N$ to be

$$\mathrm{Conj}_{C_N}(G) = \{G' \in C_N \,|\, \exists K \in C_N \text{ s.t. } G' = e^{i\phi}KGK^{-1}\}, \tag{110}$$

where the equivalences are up to arbitrary overall phases $e^{i\phi}$ (since we are considering gates, not group elements).

We define the *sequence stabilizer* and *sequence conjugacy class* for a MZM projector sequence $\mathcal{G}$, acting on the $2N$ qubits corresponding to $N$ hexons, that compiles to the gate $G$ as

$$\mathrm{Stab}(\mathcal{G}) = \left\{\mathcal{A} \in C_{2N}\,\middle|\, \mathcal{A}\mathcal{G}\mathcal{A}^{-1} \propto \Pi_+^{(\mathrm{anc})} \otimes G\right\} \tag{111}$$

$$\mathrm{Conj}(\mathcal{G}) = \left\{\mathcal{G}'\,\middle|\, \begin{array}{l}\exists \mathcal{K} \in C_{2N} \text{ s.t. } \mathcal{G}' = e^{i\phi}\mathcal{K}\mathcal{G}\mathcal{K}^{-1},\\ \mathcal{G}' \propto \Pi_+^{(\mathrm{anc})} \otimes G', \, G' \in \mathrm{Conj}_{C_N}(G)\end{array}\right\}. \tag{112}$$

We note that conjugation by Clifford gates maps fermionic parity projectors to fermionic parity projectors, though possibly changes the number and location of MZMs involved in the projection operator. This follows from the observation that conjugation by Clifford gates maps multi-qubit Pauli operators to multi-qubit Pauli operators, up to possible signs, together with the bijection between multi-MZM parity operators and the multi-qubit Pauli operators established by Eq. (7). Thus, conjugating $\mathcal{G}$ by $\mathcal{A} \in \mathrm{Stab}(\mathcal{G})$ yields the (potentially different) projector sequence

$$\tilde{\mathcal{G}} = \mathcal{A}\mathcal{G}\mathcal{A}^{\dagger} = \widetilde{\Pi}_+^{(\mathrm{anc})}\widetilde{\Pi}_{s_{n-1}}^{(\mathcal{M}_{n-1})}\ldots\widetilde{\Pi}_{s_1}^{(\mathcal{M}_1)}\widetilde{\Pi}_+^{(\mathrm{anc})} \propto \Pi_+^{(\mathrm{anc})} \otimes G, \tag{113}$$

where

$$\widetilde{\Pi}_{s_\mu}^{(\mathcal{M}_\mu)} = \mathcal{A}\Pi_{s_\mu}^{(\mathcal{M}_\mu)}\mathcal{A}^{\dagger} = \Pi_{s_\mu}^{(\widetilde{\mathcal{M}}_\mu)}, \tag{114}$$

is the fermionic parity projection operator, corresponding to a new set of MZMs $\widetilde{\mathcal{M}}_\mu$, for which the number, order, and locations of MZMs may be different than those of the original set $\mathcal{M}_\mu$. This is generally determined from the transformation of the fermionic parity operators

$$\Gamma_{\widetilde{\mathcal{M}}} = \mathcal{A}\Gamma_{\mathcal{M}}\mathcal{A}^{\dagger}. \tag{115}$$

For example, for a single hexon ($N = 1$), the pairwise projectors become

$$\widetilde{\Pi}_s^{(jk)} = \mathcal{A}\Pi_s^{(jk)}\mathcal{A}^\dagger = \frac{1}{2}(\mathbb{1} + s\mathcal{A}(i\gamma_j\gamma_k)\mathcal{A}^\dagger) = \Pi_s^{(\widetilde{jk})}, \tag{116}$$

where $(\widetilde{jk})$ is a potentially different pair of MZM labels than $(jk)$, and we allow the order of labels in $(\widetilde{jk})$ to be used to absorb changes in the sign of $s$, i.e. $\Pi_{-s}^{(ab)} = \Pi_s^{(ba)}$. We note that $\mathcal{A} \in \mathrm{Stab}(\mathcal{G})$ has the property that $\mathcal{A}\left(\mathbf{\Pi}_+^{(\mathrm{anc})} \otimes \mathbb{1}_{2^N}\right)\mathcal{A}^\dagger = \mathbf{\Pi}_+^{(\mathrm{anc})} \otimes \mathbb{1}_{2^N}$ and the property that $\left(\mathbf{\Pi}_+^{(\mathrm{anc})} \otimes \mathbb{1}_{2^N}\right)\mathcal{A}\left(\mathbf{\Pi}_+^{(\mathrm{anc})} \otimes \mathbb{1}_{2^N}\right) = \mathbf{\Pi}_+^{(\mathrm{anc})} \otimes A$ for some $A \in \mathrm{Stab}_{\mathrm{C}_N}(G)$.

Conjugating $\mathcal{G}$ and $\mathcal{G}^{M*}$ by the elements of its stabilizer $\mathrm{Stab}(\mathcal{G})$ yields up to $2|\mathrm{Stab}(\mathcal{G})|$ possible compilations for a target gate $G$ with the same sequence length as $\mathcal{G}$. In practice, this generates fewer than $2|\mathrm{Stab}(\mathcal{G})|$ distinct projector sequences, because an entire sequence is often invariant with respect to some subgroup of the stabilizer group.

Likewise, given a compilation for a gate $G$, we also are able to generate compilations for every other element of its conjugacy class $H \in \mathrm{Conj}_{\mathrm{C}_N}(G)$. This is because, by definition, there exists an $X \in \mathrm{C}_n$ such that $H = e^{i\phi}XGX^\dagger$.

For example, for a single hexon, the sequence $\mathcal{S} = \Pi_+^{(34)}\Pi_+^{(23)}\Pi_+^{(13)}\Pi_+^{(34)}$, which compiles to the phase gate $S$, yields a total of 16 distinct compilations by starting with either $\mathcal{S}$ or $\mathcal{S}^{M*}$ and conjugating by elements of $\mathrm{Stab}(\mathcal{S})$.

It may be useful to impose a locality constraint that restricts which elements of $\mathrm{Stab}(\mathcal{G})$ and $\mathrm{Conj}(\mathcal{G})$ we utilize, in order to prevent the physical measurements from increasing in complexity, i.e. so that the resulting measurements do not involve a larger number of MZMs nor additional hexons. This can be accomplished by restricting $\mathcal{A}$ and $\mathcal{K}$ in these definitions to the subset of Clifford gates generated by $\{S_{a_j}, S_{q_j}, H_{q_j}, \mathrm{C}(Z)_{a_jq_j}\}$, where $a_j$ and $q_j$ labels the $j$th hexon's ancillary and computational qubits, respectively. If we denote the subset of Clifford gates generated by these gate as the *hexon-local* Clifford gates $\widehat{\mathrm{C}_{2N}} \subset \mathrm{C}_{2N}$, then we define the *hexon-local sequence stabilizer* to be

$$\widehat{\mathrm{Stab}}(\mathcal{G}) = \left\{\mathcal{A} \in \widehat{\mathrm{C}_{2N}} \,\middle|\, \mathcal{A}\mathcal{G}\mathcal{A}^{-1} \propto \mathbf{\Pi}_+^{(\mathrm{anc})} \otimes G\right\}, \tag{117}$$

and the *hexon-local sequence conjugacy class* to be

$$\widehat{\mathrm{Conj}}(\mathcal{G}) = \left\{\mathcal{G}' \,\middle|\, \begin{array}{l} \exists \mathcal{K} \in \widehat{\mathrm{C}_{2N}} \text{ s.t. } \mathcal{G}' = e^{i\phi}\mathcal{K}\mathcal{G}\mathcal{K}^{-1}, \\ \mathcal{G}' \propto \mathbf{\Pi}_+^{(\mathrm{anc})} \otimes G', G' \in \mathrm{Conj}_{\mathrm{C}_N}(G) \end{array}\right\}. \tag{118}$$

Conjugating a fermionic parity operator $\Gamma_{\mathcal{M}}$ by an element $\mathcal{A} \in \widehat{\mathrm{C}_{2N}}$ of the hexon-local Clifford gates yields a fermionic parity operator $\Gamma_{\widetilde{\mathcal{M}}}$ that involves the same number of MZMs from each hexon, though possibly with different locations within each hexon. Hence, the corresponding projectors and measurements for these operators involve the same number of MZMs from each hexon. In other words, the locality with respect to hexons of the corresponding measurement is preserved.

### B.2.3  Sequence reduction

The goal of this subsection is to find an efficient compilation for a target gate $G$ that can be generated from a generating gate set $\{G_1, \ldots, G_N\}$, for which we have the corresponding compilations $\{\mathcal{G}_1, \ldots \mathcal{G}_N\}$ with respective sequence lengths $\{L_1, L_2, \ldots, L_N\}$. (We do not count the initial ancillary projector $\mathbf{\Pi}_+^{(\mathrm{anc})}$ in the sequence lengths $L_j$, since each prior step ends with such a projector.) Our strategy will be to start from a projector sequence obtained by naïvely taking the product of generating gates' projector sequences and then iteratively reducing the combined sequence length via the reduction formulas outlined in Sec. B.1.

The protocol for doing this is as follows:

1) For each expression $G = G_{j_m} \dots G_{j_1}$ of the target gate in terms of the generating gates, concatenate the corresponding projector sequences to obtain a projector sequence

$$\mathcal{G} = \mathcal{G}_{j_m} \dots \mathcal{G}_{j_1}, \tag{119}$$

that compiles to $G$. The resulting projector sequence has length $L = \sum_{q=1}^{m} L_{j_q}$.

2) Find all alternate compilations for each generator's projector sequence $\mathcal{G}_{j_q}$, using the methods of Sec. B.2. We denote the distinct projector sequences for generator $G_{j_q}$ as $\mathcal{G}_{j_q}^{(\alpha_q)}$ for $\alpha_q = 1, \dots, K_{j_q}$, where $K_{j_q}$ is the number of distinct sequences. Construct all possible projector sequences (up to scalar factors) by independently replacing each $\mathcal{G}_{j_q}$ in the sequence with the alternates $\mathcal{G}_{j_q}^{(\alpha_q)}$ to get

$$\mathcal{G}^{(\vec{\alpha})} = \mathcal{G}_{j_m}^{(\alpha_m)} \dots \mathcal{G}_{j_1}^{(\alpha_1)}, \tag{120}$$

where $\vec{\alpha}$ is used to label the different compilations. In this way, we have produced a (naïve) total of $\prod_{q=1}^{m} K_{j_q}$ possible compilations for $G$.

3) For each $\mathcal{G}^{(\vec{\alpha})}$, search for and apply all possible reductions of each sequence $\mathcal{G}^{(\vec{\alpha})}$ via the reduction formulas introduced in Sec. B.1. Repeat until no further reduction is possible. Each reduction will lower the length of the overall sequence by 1-2 projectors. We denote the fully reduced projector sequence obtained from a projector sequence $\mathcal{G}$ as $\breve{\mathcal{G}}$.

   If each generator's projector sequence is already fully reduced, i.e. $\mathcal{G}_{j_q}^{(\alpha_q)} = \breve{\mathcal{G}}_{j_q}^{(\alpha_q)}$, the remaining reductions will be found at the locations in the projector sequence where the generator subsequences are concatenated (at least at the initial reduction iterations).

Gates compiled from a large number of generator gates can obtain a significant reduction in the length of their projector sequence using this procedure. Note that single-qubit gates acting on different qubits only benefit from the reduction procedure applied individually within qubits, but it is possible to obtain collective reductions for combinations of single-qubit and two-qubit gates.

As an example, let us apply the reduction procedure to the compilation $\mathcal{G} \propto \Pi_+^{(\text{anc})} \otimes C(Z)$, using the gate compilation $C(Z) = S_2 S_1 W$ and the generating gate projector sequences from Ref. [9]:

$$S_1 = \Pi_+^{(34)} \Pi_+^{(13)} \Pi_+^{(23)}, \tag{121}$$

$$S_2 = \Pi_+^{(3'4')} \Pi_+^{(1'3')} \Pi_+^{(2'3')}, \tag{122}$$

$$\mathcal{W} = \Pi_+^{(34)} \Pi_+^{(35)} \Pi_+^{(56;1'2')} \Pi_+^{(45)}. \tag{123}$$

$$\tag{124}$$

The naïve compilation $\mathcal{G} = S_2 S_1 \mathcal{W}$ has length 10. Applying the reduction procedure, we obtain a compilation of length 8, as follows:

$$\begin{aligned}
\mathcal{G} &= \left[ \Pi_+^{(3'4')} \Pi_+^{(1'3')} \Pi_+^{(2'3')} \right] \left[ \Pi_+^{(34)} \Pi_+^{(13)} \Pi_+^{(23)} \right] \left[ \Pi_+^{(34)} \Pi_+^{(35)} \Pi_+^{(56;1'2')} \Pi_+^{(45)} \right] \\
&\rightarrow \left[ \Pi_+^{(3'4')} \Pi_+^{(1'3')} \Pi_+^{(2'3')} \right] \left[ \Pi_+^{(34)} \Pi_-^{(36)} \Pi_+^{(35)} \right] \left[ \Pi_+^{(34)} \Pi_+^{(35)} \Pi_+^{(56;1'2')} \Pi_+^{(45)} \right] \\
&\rightarrow \Pi_+^{(3'4')} \Pi_+^{(1'3')} \Pi_+^{(2'3')} \Pi_+^{(34)} \Pi_-^{(36)} \Pi_+^{(35)} \Pi_+^{(56;1'2')} \Pi_+^{(45)}. \tag{125}
\end{aligned}$$

In the first step, we replaced $\mathcal{S}_1$ with the alternate compilation $\widetilde{\mathcal{S}}_1 = \Pi_+^{(34)}\Pi_-^{(36)}\Pi_+^{(35)}$. In the second step, we applied the reduction formula $\Pi_+^{(35)}\Pi_+^{(34)}\Pi_+^{(35)} \rightarrow \Pi_+^{(35)}$. For this example, a more thorough search would have yielded a better result, as one can find more efficient compilations of both $W$ and $\mathsf{C}(Z)$, with lengths 3 and 4, respectively, given by

$$W = \Pi_+^{(34)}\Pi_+^{(35)}\Pi_+^{(36;1'2')} \tag{126}$$

$$\mathsf{C}(Z) = \Pi_+^{(34)}\Pi_+^{(46)}\Pi_+^{(56)}\Pi_+^{(46;1'2')}. \tag{127}$$

However, while brute-force search may be employed in this example, it is not practical to do so in general, as the space of projector sequences grows exponentially in the sequence length. Furthermore, one can in principle consider other cost functions to optimize against, for example taking into account that some projectors can be applied simultaneously and that some measurements may be more difficult than others. We will discuss these points in more detail in Sec. 2.4.

## C  Demonstration Details

In this appendix, we present the details of the demonstration of our methods outlined in Sec. 5.3 for the very roughly estimated weight factor values: $w_c = 1.25$, $w_t = 1.65$, $w_a = 1.01$, and $f(N) = (\prod_{n=1}^{N} n!)^{(N-1)!}$.

The two options for forced-measurement operations are described in Sec. 3.

### C.1  Two-Sided Hexon with Configuration $\langle 3, 4, 1, 2, 6, 5 \rangle$

The MZM labeling configuration $\langle 3, 4, 1, 2, 6, 5 \rangle$ is optimal for the two-sided hexon architecture, when using either the forced-measurement methods or the Majorana-Pauli tracking methods, for the gates or Pauli cosets of gates: the Hadamard gate $H$, the geometric average of single-qubit Clifford gates $\mathsf{C}_1$, the controlled-$X$ gate $\mathsf{C}(X)$, and the geometric average of the controlled-Pauli gates $\mathsf{C}(P)$. This configuration within an array looks like:

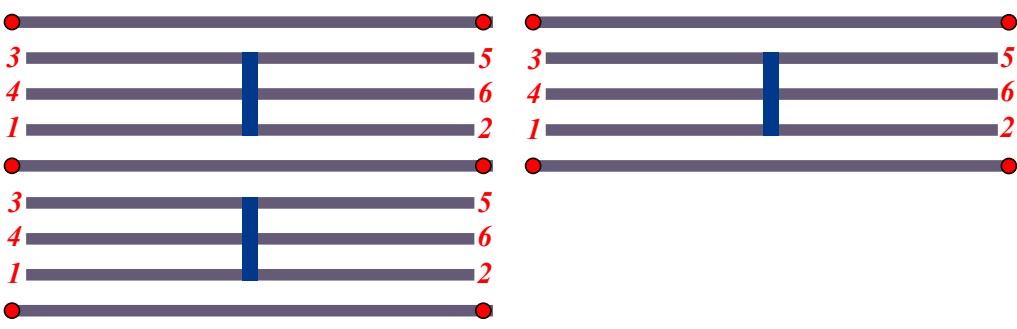

Table 5: Minimal difficulty weight measurement sequences for each single-qubit Clifford gate when using forced-measurement methods. For comparison, we also present the corresponding realization of the gates formed by using the generating gate sets $\langle S,H\rangle$ and $\langle S,B\rangle$.

| Gate | Forced Measurement Sequence | Weight | $\langle S,H\rangle$ | Weight | $\langle S,B\rangle$ | Weight |
|---|---|---|---|---|---|---|
| $X$ | $\Pi_+^{(34)}\Pi_{s_3}^{(36)}\Pi_-^{(34)}\Pi_{s_1}^{(14)}$ | $3.67\times10^5$ | $HSSH$ | $1.38\times10^{26}$ | $BB$ | $1.82\times10^{10}$ |
| $Y$ | $\Pi_+^{(34)}\Pi_{s_3}^{(35)}\Pi_-^{(34)}\Pi_{s_1}^{(14)}$ | $2.30\times10^5$ | $HSSHSS$ | $9.85\times10^{35}$ | $BBSS$ | $1.30\times10^{20}$ |
| $Z$ | $\Pi_+^{(34)}\Pi_{-s_1}^{(14)}\Pi_{s_2}^{(12)}\Pi_{s_1}^{(14)}$ | $2.30\times10^5$ | $SS$ | $7.14\times10^9$ | $SS$ | $7.14\times10^9$ |
| $S$ | $\Pi_+^{(34)}\Pi_{s_1}^{(14)}\Pi_{s_1}^{(24)}$ | $8.45\times10^4$ | $S$ | $8.45\times10^4$ | $S$ | $8.45\times10^4$ |
| $XS$ | $\Pi_+^{(34)}\Pi_{s_4}^{(35)}\Pi_-^{(34)}\Pi_{-s_1}^{(14)}\Pi_{s_1}^{(24)}$ | $1.39\times10^8$ | $HSSHS$ | $1.17\times10^{31}$ | $BBS$ | $1.54\times10^{15}$ |
| $YS$ | $\Pi_+^{(34)}\Pi_{s_4}^{(35)}\Pi_-^{(34)}\Pi_{s_1}^{(14)}\Pi_{s_1}^{(24)}$ | $1.39\times10^8$ | $HSSHS^\dagger$ | $1.17\times10^{31}$ | $SBB$ | $1.54\times10^{15}$ |
| $ZS$ | $\Pi_+^{(34)}\Pi_{-s_1}^{(14)}\Pi_{s_1}^{(24)}$ | $8.45\times10^4$ | $S^\dagger$ | $8.45\times10^4$ | $S^\dagger$ | $8.45\times10^4$ |
| $H$ | $\Pi_+^{(34)}\Pi_{-s_1}^{(35)}\Pi_{s_2}^{(56)}\Pi_{s_2}^{(25)}\Pi_{s_1}^{(35)}$ | $1.39\times10^8$ | $H$ | $1.39\times10^8$ | $SBS$ | $9.64\times10^{14}$ |
| $XH$ | $\Pi_+^{(34)}\Pi_{-s_1}^{(13)}\Pi_{s_1}^{(35)}$ | $1.07\times10^5$ | $HSS$ | $9.92\times10^{17}$ | $S^\dagger B^\dagger S$ | $9.64\times10^{14}$ |
| $YH$ | $\Pi_+^{(34)}\Pi_{-s_1}^{(35)}\Pi_{-s_2}^{(56)}\Pi_{s_2}^{(25)}\Pi_{s_1}^{(35)}$ | $1.39\times10^8$ | $SSHSS$ | $7.09\times10^{27}$ | $SB^\dagger S$ | $9.64\times10^{14}$ |
| $ZH$ | $\Pi_+^{(34)}\Pi_{s_1}^{(13)}\Pi_{s_1}^{(35)}$ | $1.07\times10^5$ | $SSH$ | $9.92\times10^{17}$ | $S^\dagger BS$ | $9.64\times10^{14}$ |
| $SH$ | $\Pi_+^{(34)}\Pi_{s_1}^{(14)}\Pi_{-s_1}^{(24)}\Pi_{s_1}^{(46)}$ | $8.16\times10^7$ | $SH$ | $1.17\times10^{13}$ | $B^\dagger S^\dagger$ | $1.14\times10^{10}$ |
| $XSH$ | $\Pi_+^{(34)}\Pi_{-s_1}^{(14)}\Pi_{s_1}^{(24)}\Pi_{s_1}^{(46)}$ | $8.16\times10^7$ | $S^\dagger HSS$ | $8.39\times10^{22}$ | $BS^\dagger$ | $1.14\times10^{10}$ |
| $YSH$ | $\Pi_+^{(34)}\Pi_{s_1}^{(14)}\Pi_{s_1}^{(24)}\Pi_{s_1}^{(46)}$ | $8.16\times10^7$ | $SHSS$ | $8.39\times10^{22}$ | $B^\dagger S$ | $1.14\times10^{10}$ |
| $ZSH$ | $\Pi_+^{(34)}\Pi_{-s_1}^{(14)}\Pi_{s_1}^{(24)}\Pi_{s_1}^{(46)}$ | $8.16\times10^7$ | $S^\dagger H$ | $1.17\times10^{13}$ | $BS$ | $1.14\times10^{10}$ |
| $HS$ | $\Pi_+^{(34)}\Pi_{s_1}^{(13)}\Pi_{-s_1}^{(35)}\Pi_{s_1}^{(36)}$ | $6.46\times10^7$ | $HS$ | $1.17\times10^{13}$ | $S^\dagger B^\dagger$ | $1.14\times10^{10}$ |
| $XHS$ | $\Pi_+^{(34)}\Pi_{-s_1}^{(13)}\Pi_{s_1}^{(35)}\Pi_{s_1}^{(36)}$ | $6.46\times10^7$ | $HS^\dagger$ | $1.17\times10^{13}$ | $SB$ | $1.14\times10^{10}$ |
| $YHS$ | $\Pi_+^{(34)}\Pi_{-s_1}^{(13)}\Pi_{s_1}^{(35)}\Pi_{s_1}^{(36)}$ | $6.46\times10^7$ | $SSHS^\dagger$ | $8.39\times10^{22}$ | $S^\dagger B$ | $1.14\times10^{10}$ |
| $ZHS$ | $\Pi_+^{(34)}\Pi_{s_1}^{(13)}\Pi_{s_1}^{(35)}\Pi_{s_1}^{(36)}$ | $6.46\times10^7$ | $SSHS$ | $8.39\times10^{22}$ | $SB^\dagger$ | $1.14\times10^{10}$ |
| $SHS$ | $\Pi_+^{(34)}\Pi_{s_1}^{(14)}\Pi_{s_1}^{(46)}$ | $1.35\times10^5$ | $SHS$ | $9.92\times10^{17}$ | $B^\dagger$ | $1.35\times10^5$ |
| $XSHS$ | $\Pi_+^{(34)}\Pi_{-s_1}^{(14)}\Pi_{s_1}^{(46)}$ | $1.35\times10^5$ | $S^\dagger HS^\dagger$ | $9.92\times10^{17}$ | $B$ | $1.35\times10^5$ |
| $YSHS$ | $\Pi_+^{(34)}\Pi_{-s_2s_1}^{(14)}\Pi_{s_2}^{(12)}\Pi_{s_2}^{(26)}\Pi_{s_1}^{(23)}$ | $1.76\times10^8$ | $SHS^\dagger$ | $9.92\times10^{17}$ | $B^\dagger SS$ | $9.64\times10^{14}$ |
| $ZSHS$ | $\Pi_+^{(34)}\Pi_{s_2s_1}^{(14)}\Pi_{s_2}^{(12)}\Pi_{s_2}^{(26)}\Pi_{s_1}^{(23)}$ | $1.76\times10^8$ | $S^\dagger HS$ | $9.92\times10^{17}$ | $BSS$ | $9.64\times10^{14}$ |
| Average | | $7.72\times10^6$ | | $6.04\times10^{18}$ | | $2.25\times10^{11}$ |

Table 6: Minimal difficulty weight measurement sequences for each Pauli coset of single-qubit Clifford gates when using Majorana-Pauli tracking methods. For comparison, we also present the corresponding realization of the gates formed by using the generating gate sets $\langle S,H\rangle$ and $\langle S,B\rangle$.

| Pauli Class | Tracked Measurement Sequence | Weight | $\langle S,H\rangle$ Decomp. | Weight | $\langle S,B\rangle$ Decomp. | Weight |
|---|---|---|---|---|---|---|
| $[S]$ | $\Pi_{s_3}^{(34)}\Pi_{s_2}^{(24)}\Pi_{s_1}^{(14)}$ | $1.76\times10^2$ | $S$ | $1.76\times10^2$ | $S$ | $1.76\times10^2$ |
| $[H]$ | $\Pi_{s_3}^{(34)}\Pi_{s_2}^{(35)}\Pi_{s_1}^{(13)}$ | $1.76\times10^2$ | $H$ | $1.76\times10^2$ | $SBS$ | $6.88\times10^6$ |
| $[SH]$ | $\Pi_{s_4}^{(34)}\Pi_{s_3}^{(36)}\Pi_{s_2}^{(35)}\Pi_{s_1}^{(13)}$ | $2.63\times10^3$ | $SH$ | $3.10\times10^4$ | $BS$ | $3.91\times10^4$ |
| $[HS]$ | $\Pi_{s_4}^{(34)}\Pi_{s_3}^{(36)}\Pi_{s_2}^{(13)}\Pi_{s_1}^{(35)}$ | $2.63\times10^3$ | $HS$ | $3.10\times10^4$ | $SB$ | $3.91\times10^4$ |
| $[SHS]$ | $\Pi_{s_3}^{(34)}\Pi_{s_2}^{(46)}\Pi_{s_1}^{(14)}$ | $2.22\times10^2$ | $SHS$ | $5.45\times10^6$ | $B$ | $2.22\times10^2$ |
| Average | | $5.44\times10^2$ | | $1.10\times10^4$ | | $1.33\times10^4$ |

Table 7: Pauli gate corrections tracked for the corresponding single-qubit gates implemented using Majorana-Pauli tracking methods. The implicit action on the ancillary qubit is $X^{\frac{1-s_n s_0}{2}}\Pi_{s_0}^{(34)}$ for sequences of length $n$.

| Gate | Tracked Measurement Sequence |
|---|---|
| $Y^{\frac{1-s_3 s_0}{2}} Z^{\frac{1-s_2 s_1}{2}} S$ | $\Pi_{s_3}^{(34)}\Pi_{s_2}^{(14)}\Pi_{s_1}^{(24)}\Pi_{s_0}^{(34)}$ |
| $Y^{\frac{1-s_3 s_0}{2}} Y^{\frac{1-s_2 s_1 s_0}{2}} ZH$ | $\Pi_{s_3}^{(34)}\Pi_{s_2}^{(13)}\Pi_{s_1}^{(35)}\Pi_{s_0}^{(34)}$ |
| $Z^{\frac{1-s_3 s_0}{2}} Z^{\frac{1-s_3 s_2 s_0}{2}} X^{\frac{1-s_2 s_1 s_0}{2}} XSH$ | $\Pi_{s_4}^{(34)}\Pi_{s_3}^{(36)}\Pi_{s_2}^{(35)}\Pi_{s_1}^{(13)}\Pi_{s_0}^{(34)}$ |
| $Y^{\frac{1-s_3 s_0}{2}} Y^{\frac{1-s_3 s_2 s_0}{2}} X^{\frac{1-s_2 s_1 s_0}{2}} ZHS$ | $\Pi_{s_4}^{(34)}\Pi_{s_3}^{(13)}\Pi_{s_2}^{(35)}\Pi_{s_1}^{(36)}\Pi_{s_0}^{(34)}$ |
| $Y^{\frac{1-s_3 s_0}{2}} X^{\frac{1-s_2 s_1}{2}} SHS$ | $\Pi_{s_3}^{(34)}\Pi_{s_2}^{(14)}\Pi_{s_1}^{(46)}\Pi_{s_0}^{(34)}$ |

Table 8: Minimal difficulty weight measurement sequences for controlled-Pauli two-qubit gates when using forced-measurement methods. The labels $(u)$, $(d)$, $(r)$, and $(l)$ indicates that for a hexon acting as the control qubit of the $\mathsf{C}(P)$ gate, the corresponding target qubit is the nearest neighbor hexon in the up, down, right, and left direction, respectively. Notice that the choice of control and target qubit is arbitrary for $\mathsf{C}(Z)$, so $(u)$ and $(d)$ are related by symmetry, as are $(r)$, and $(l)$. The average difficulty weight of the four directions is labeled by $(a)$. For comparison, we also present the corresponding realization of the gates formed by using the generating gate sets $\langle S, H, \mathsf{C}(Z)\rangle$ and $\langle S, B, W\rangle$. $^A\mathsf{C}(Z) = A_2\mathsf{C}(Z)A_2^\dagger$ denotes conjugation of $\mathsf{C}(Z)$ by $A$ on the target qubit.

| Gate | Forced Measurement Sequence | Weight | $\langle S, H, \mathsf{C}(Z)\rangle$ | Weight | $\langle S, B, W\rangle$ | Weight |
|---|---|---|---|---|---|---|
| $\mathsf{C}(X)$ $(u)$ | $\overset{ef}{\Pi}_+^{(34)}\overset{ef}{\Pi}_s^{(35)}\overset{ef}{\Pi}_+^{(56)}\Pi_s^{(35;1'6')}$ | $1.79\times10^9$ | $^H\mathsf{C}(Z)$ | $2.16\times10^{25}$ | $S_2 B_2 S_1 W^\dagger S_2^\dagger B_2 S_2$ | $2.56\times10^{36}$ |
| $(d)$ | $\overset{ef}{\Pi}_+^{(3'4')}\overset{ef}{\Pi}_s^{(3'5')}\overset{ef}{\Pi}_+^{(2'5')}\Pi_s^{(12;3'5')}$ | $2.26\times10^9$ | | $2.16\times10^{25}$ | | $2.56\times10^{36}$ |
| $(r)$ | $\overset{ef}{\Pi}_+^{(3'4')}\overset{ef}{\Pi}_{-s}^{(1'4')}\overset{ef}{\Pi}_+^{(56;3'4')}\Pi_s^{(3'6')}$ | $3.69\times10^8$ | | $8.89\times10^{24}$ | | $4.88\times10^{36}$ |
| $(l)$ | $\overset{ef}{\Pi}_+^{(34)}\overset{ef}{\Pi}_s^{(14)}\overset{ef}{\Pi}_+^{(12)}\Pi_s^{(14;2'5')}$ | $2.88\times10^8$ | | $8.89\times10^{24}$ | | $4.88\times10^{36}$ |
| $(a)$ | | $8.10\times10^8$ | | $1.39\times10^{25}$ | | $3.54\times10^{36}$ |
| $\mathsf{C}(Y)$ $(u)$ | $\overset{ef}{\Pi}_+^{(34)}\overset{ef}{\Pi}_s^{(35)}\overset{ef}{\Pi}_+^{(56)}\Pi_s^{(35;1'5')}$ | $2.85\times10^9$ | $^{SH}\mathsf{C}(Z)$ | $1.55\times10^{35}$ | $B_2^\dagger W^\dagger S_2 B_2 S_1$ | $3.59\times10^{26}$ |
| $(d)$ | $\overset{ef}{\Pi}_+^{(3'4')}\overset{\frown}{\Pi}_s^{(1'4')}\overset{ef}{\Pi}_+^{(1'5')}\Pi_s^{(12;3'5')}$ | $3.60\times10^9$ | | $1.55\times10^{35}$ | | $3.59\times10^{26}$ |
| $(r)$ | $\overset{ef}{\Pi}_+^{(3'4')}\overset{ef}{\Pi}_{-s}^{(1'4')}\overset{ef}{\Pi}_+^{(56;3'4')}\Pi_s^{(3'5')}$ | $2.32\times10^8$ | | $6.34\times10^{34}$ | | $6.83\times10^{26}$ |
| $(l)$ | $\overset{ef}{\Pi}_+^{(34)}\overset{ef}{\Pi}_{-s}^{(14)}\overset{ef}{\Pi}_+^{(12)}\Pi_s^{(14;2'6')}$ | $1.81\times10^8$ | | $6.34\times10^{34}$ | | $6.83\times10^{26}$ |
| $(a)$ | | $8.10\times10^8$ | | $9.91\times10^{34}$ | | $4.95\times10^{26}$ |
| $\mathsf{C}(Z)$ $(u)$ | $\overset{ef}{\Pi}_+^{(34)}\overset{ef}{\Pi}_s^{(35)}\overset{ef}{\Pi}_+^{(56)}\Pi_s^{(35;1'2')}$ | $1.12\times10^9$ | $\mathsf{C}(Z)$ | $1.12\times10^9$ | $S_1 S_2 W^\dagger$ | $1.97\times10^{16}$ |
| $(d)$ | $\overset{ef}{\Pi}_+^{(3'4')}\overset{ef}{\Pi}_s^{(3'5')}\overset{ef}{\Pi}_+^{(5'6')}\Pi_s^{(12;3'5')}$ | $1.12\times10^9$ | | $1.12\times10^9$ | | $1.97\times10^{16}$ |
| $(r)$ | $\overset{\frown}{\Pi}_+^{(3'4')}\overset{ef}{\Pi}_s^{(1'3')}\overset{ef}{\Pi}_+^{(1'2')}\Pi_s^{(56;1'3')}$ | $4.60\times10^8$ | | $4.60\times10^8$ | | $3.75\times10^{16}$ |
| $(l)$ | $\overset{\frown}{\Pi}_+^{(34)}\overset{ef}{\Pi}_s^{(13)}\overset{ef}{\Pi}_+^{(12)}\Pi_s^{(13;5'6')}$ | $4.60\times10^8$ | | $4.60\times10^8$ | | $3.75\times10^{16}$ |
| $(a)$ | | $7.18\times10^8$ | | $7.18\times10^8$ | | $2.72\times10^{16}$ |
| Average $(u)$ | | $1.79\times10^9$ | | $1.55\times10^{23}$ | | $2.63\times10^{26}$ |
| $(d)$ | | $2.09\times10^9$ | | $1.55\times10^{23}$ | | $2.63\times10^{26}$ |
| $(r)$ | | $3.40\times10^8$ | | $6.38\times10^{22}$ | | $5.00\times10^{26}$ |
| $(l)$ | | $2.88\times10^8$ | | $6.38\times10^{22}$ | | $5.00\times10^{26}$ |
| $(a)$ | | $7.78\times10^8$ | | $9.96\times10^{22}$ | | $3.62\times10^{26}$ |

Table 9: Minimal difficulty weight measurement sequences for Pauli cosets of controlled-Pauli two-qubit gates when using Majorana-Pauli tracking methods. The labels $(u)$, $(d)$, $(r)$, and $(l)$ indicates that for a hexon acting as the control qubit of the $\mathsf{C}(P)$ gate, the corresponding target qubit is the nearest neighbor hexon in the up, down, right, and left direction, respectively. Notice that the choice of control and target qubit is arbitrary for $\mathsf{C}(Z)$, so $(u)$ and $(d)$ are related by symmetry, as are $(r)$, and $(l)$. The average difficulty weight of the four directions is labeled by $(a)$. For comparison, we also present the corresponding realization of the gates formed by using the generating gate sets $\langle S, H, \mathsf{C}(Z)\rangle$ and $\langle S, B, W\rangle$. ${}^{A}\mathsf{C}(Z) = A_2 \mathsf{C}(Z) A_2^{\dagger}$ denotes conjugation of $\mathsf{C}(Z)$ by $A$ on the target qubit.

| Pauli Class | Tracked Measurement Sequence | Weight | $\langle S, H, \mathsf{C}(Z)\rangle$ | Weight | $\langle S, B, W\rangle$ | Weight |
|---|---|---|---|---|---|---|
| $[\mathsf{C}(X)]$ $(u)$ | $\Pi_{s_4}^{(34)}\Pi_{s_3}^{(35)}\Pi_{s_2}^{(56)}\Pi_{s_1}^{(35;1'6')}$ | $6.64 \times 10^3$ | ${}^{H}\mathsf{C}(Z)$ | $1.63 \times 10^8$ | $S_2 B_2 S_1 W^{\dagger} S_2^{\dagger} B_2 S_2$ | $4.23 \times 10^{16}$ |
| $(d)$ | $\Pi_{s_4'}^{(3'4')}\Pi_{s_3}^{(3'5')}\Pi_{s_2}^{(2'5')}\Pi_{s_1}^{(12;3'5')}$ | $6.64 \times 10^3$ | | $1.63 \times 10^8$ | | $4.23 \times 10^{16}$ |
| $(r)$ | $\Pi_{s_4'}^{(3'4')}\Pi_{s_3}^{(1'4')}\Pi_{s_2}^{(56;3'4')}\Pi_{s_1}^{(3'6')}$ | $2.66 \times 10^3$ | | $1.04 \times 10^8$ | | $5.86 \times 10^{16}$ |
| $(l)$ | $\Pi_{s_4}^{(34)}\Pi_{s_3}^{(14)}\Pi_{s_2}^{(12)}\Pi_{s_1}^{(14;2'5')}$ | $2.66 \times 10^3$ | | $1.04 \times 10^8$ | | $5.86 \times 10^{16}$ |
| $(a)$ | | $4.20 \times 10^3$ | | $1.30 \times 10^8$ | | $4.98 \times 10^{16}$ |
| $[\mathsf{C}(Y)]$ $(u)$ | $\Pi_{s_4}^{(34)}\Pi_{s_3}^{(35)}\Pi_{s_2}^{(56)}\Pi_{s_1}^{(35;1'5')}$ | $8.38 \times 10^3$ | ${}^{SH}\mathsf{C}(Z)$ | $5.04 \times 10^{12}$ | $B_2^{\dagger} W^{\dagger} S_2 B_2 S_1$ | $1.37 \times 10^{12}$ |
| $(d)$ | $\Pi_{s_4'}^{(3'4')}\Pi_{s_3}^{(3'5')}\Pi_{s_2}^{(1,2;1'5')}\Pi_{s_1}^{(;1'4')}$ | $8.38 \times 10^3$ | | $5.04 \times 10^{12}$ | | $1.37 \times 10^{12}$ |
| $(r)$ | $\Pi_{s_4'}^{(3'4')}\Pi_{s_3}^{(1'4')}\Pi_{s_2}^{(56;3'4')}\Pi_{s_1}^{(3'5')}$ | $2.11 \times 10^3$ | | $3.23 \times 10^{12}$ | | $1.89 \times 10^{12}$ |
| $(l)$ | $\Pi_{s_4}^{(34)}\Pi_{s_3}^{(14)}\Pi_{s_2}^{(12)}\Pi_{s_1}^{(14;2'6')}$ | $2.11 \times 10^3$ | | $3.23 \times 10^{12}$ | | $1.89 \times 10^{12}$ |
| $(a)$ | | $4.20 \times 10^3$ | | $4.04 \times 10^{12}$ | | $1.61 \times 10^{12}$ |
| $[\mathsf{C}(Z)]$ $(u)$ | $\Pi_{s_4}^{(34)}\Pi_{s_3}^{(35)}\Pi_{s_2}^{(56)}\Pi_{s_1}^{(35;1'2')}$ | $5.26 \times 10^3$ | $\mathsf{C}(Z)$ | $5.26 \times 10^3$ | $S_1 S_2 W^{\dagger}$ | $2.77 \times 10^7$ |
| $(d)$ | $\Pi_{s_4'}^{(3'4')}\Pi_{s_3}^{(3'5')}\Pi_{s_2}^{(5'6')}\Pi_{s_1}^{(12;3'5')}$ | $5.26 \times 10^3$ | | $5.26 \times 10^3$ | | $2.77 \times 10^7$ |
| $(r)$ | $\Pi_{s_4'}^{(3'4')}\Pi_{s_3}^{(2'3')}\Pi_{s_2}^{(56;3'4')}\Pi_{s_1}^{(1'4')}$ | $3.36 \times 10^3$ | | $3.36 \times 10^3$ | | $3.84 \times 10^7$ |
| $(r)$ | $\Pi_{s_4}^{(34)}\Pi_{s_3}^{(23)}\Pi_{s_2}^{(34;5'6')}\Pi_{s_1}^{(14)}$ | $3.36 \times 10^3$ | | $3.36 \times 10^3$ | | $3.84 \times 10^7$ |
| $(a)$ | | $4.20 \times 10^3$ | | $4.20 \times 10^3$ | | $3.26 \times 10^7$ |
| Average $(u)$ | | $6.64 \times 10^3$ | | $1.63 \times 10^8$ | | $1.17 \times 10^{12}$ |
| $(d)$ | | $6.64 \times 10^3$ | | $1.63 \times 10^8$ | | $1.17 \times 10^{12}$ |
| $(r)$ | | $2.66 \times 10^3$ | | $1.04 \times 10^8$ | | $1.62 \times 10^{12}$ |
| $(l)$ | | $2.66 \times 10^3$ | | $1.04 \times 10^8$ | | $1.62 \times 10^{12}$ |
| $(a)$ | | $4.20 \times 10^3$ | | $1.30 \times 10^8$ | | $1.38 \times 10^{12}$ |

Table 10: Pauli gate corrections tracked for the corresponding controlled-Pauli gates implemented using Majorana-Pauli tracking methods. The implicit action on the two ancillary qubits is $X^{\frac{1-s_n s_0}{2}}\Pi_{s_0}^{(34)}\otimes X^{\frac{1-s_n' s_0'}{2}}\Pi_{s_0'}^{(3'4')}$ for sequences of length $n$. (When there is not a final projector for one of the ancillary pairs, it is equavalent to there being a projector for that ancillary pair onto its initial projection channel, e.g. $s_n = s_0$ or $s_n' = s_0'$.)

| Gate | Tracked Measurement Sequence |
|---|---|
| $\left(Z^{\frac{1-s_3 s_1}{2}} \otimes X^{\frac{1-s_2 s_0}{2}}\right)C(X)_u$ | $\Pi_{s_4}^{(34)}\Pi_{s_3}^{(35)}\Pi_{s_2}^{(56)}\Pi_{s_1}^{(35;1'6')}\boldsymbol{\Pi}_{s_0}^{(anc)}$ |
| $\left(Z^{\frac{1-s_2 s_0'}{2}} \otimes X^{\frac{1-s_3 s_1}{2}}\right)C(X)_d$ | $\Pi_{s_4'}^{(3'4')}\Pi_{s_3}^{(3'5')}\Pi_{s_2}^{(2'5')}\Pi_{s_1}^{(12;3'5')}\boldsymbol{\Pi}_{s_0}^{(anc)}$ |
| $\left(Z^{\frac{1+s_3 s_1 s_0'}{2}} \otimes Y^{\frac{1-s_4' s_0'}{2}} X^{\frac{1-s_2 s_0 s_0'}{2}}\right)C(X)_r$ | $\Pi_{s_4'}^{(3'4')}\Pi_{s_3}^{(1'4')}\Pi_{s_2}^{(56;3'4')}\Pi_{s_1}^{(3'6')}\boldsymbol{\Pi}_{s_0}^{(anc)}$ |
| $\left(Y^{\frac{1-s_4 s_0}{2}} Z^{\frac{1-s_3 s_1 s_0'}{2}} \otimes X^{\frac{1-s_2}{2}}\right)C(X)_l$ | $\Pi_{s_4}^{(34)}\Pi_{s_3}^{(14)}\Pi_{s_2}^{(12)}\Pi_{s_1}^{(14;2'5')}\boldsymbol{\Pi}_{s_0}^{(anc)}$ |
| $\left(Z^{\frac{1-s_3 s_1 s_0'}{2}} \otimes Y^{\frac{1-s_2 s_0}{2}}\right)C(Y)_u$ | $\Pi_{s_4}^{(34)}\Pi_{s_3}^{(35)}\Pi_{s_2}^{(56)}\Pi_{s_1}^{(35;1'5')}\boldsymbol{\Pi}_{s_0}^{(anc)}$ |
| $\left(Z^{\frac{1+s_3 s_1}{2}} \otimes Y^{\frac{1+s_2 s_0'}{2}}\right)C(Y)_d$ | $\Pi_{s_4'}^{(3'4')}\Pi_{s_3}^{(3'5')}\Pi_{s_2}^{(1,2;1'5')}\Pi_{s_1}^{(;1'4')}\boldsymbol{\Pi}_{s_0}^{(anc)}$ |
| $\left(Z^{\frac{1+s_3 s_1}{2}} \otimes Y^{\frac{1-s_4' s_2 s_0}{2}}\right)C(Y)_r$ | $\Pi_{s_4'}^{(3'4')}\Pi_{s_3}^{(1'4')}\Pi_{s_2}^{(56;3'4')}\Pi_{s_1}^{(3'5')}\boldsymbol{\Pi}_{s_0}^{(anc)}$ |
| $\left(Y^{\frac{1-s_4 s_0}{2}} Z^{\frac{1+s_3 s_1}{2}} \otimes Y^{\frac{1-s_2}{2}}\right)C(Y)_l$ | $\Pi_{s_4}^{(34)}\Pi_{s_3}^{(14)}\Pi_{s_2}^{(12)}\Pi_{s_1}^{(14;2'6')}\boldsymbol{\Pi}_{s_0}^{(anc)}$ |
| $\left(Z^{\frac{1-s_3 s_1}{2}} \otimes Z^{\frac{1-s_2 s_0}{2}}\right)C(Z)_u$ | $\Pi_{s_4}^{(34)}\Pi_{s_3}^{(35)}\Pi_{s_2}^{(56)}\Pi_{s_1}^{(35;1'2')}\boldsymbol{\Pi}_{s_0}^{(anc)}$ |
| $\left(Z^{\frac{1-s_2 s_0'}{2}} \otimes Z^{\frac{1-s_3 s_1}{2}}\right)C(Z)_d$ | $\Pi_{s_4'}^{(3'4')}\Pi_{s_3}^{(3'5')}\Pi_{s_2}^{(5'6')}\Pi_{s_1}^{(12;3'5')}\boldsymbol{\Pi}_{s_0}^{(anc)}$ |
| $\left(Z^{\frac{1-s_3 s_1 s_0'}{2}} \otimes X^{\frac{1-s_4' s_0'}{2}} Z^{\frac{1-s_2 s_0 s_0'}{2}}\right)C(Z)_r$ | $\Pi_{s_4'}^{(3'4')}\Pi_{s_3}^{(2'3')}\Pi_{s_2}^{(56;3'4')}\Pi_{s_1}^{(1'4')}\boldsymbol{\Pi}_{s_0}^{(anc)}$ |
| $\left(X^{\frac{1-s_4 s_0}{2}} Z^{\frac{1-s_2 s_0 s_0'}{2}} \otimes Z^{\frac{1-s_3 s_1 s_0}{2}}\right)C(Z)_l$ | $\Pi_{s_4}^{(34)}\Pi_{s_3}^{(23)}\Pi_{s_2}^{(34;5'6')}\Pi_{s_1}^{(14)}\boldsymbol{\Pi}_{s_0}^{(anc)}$ |

Table 11: Minimal difficulty weight measurement sequences for the two-qubit SWAP and $W$ gates when using forced-measurement methods and the Pauli cosets of SWAP and $W$ when using Majorana-Pauli tracking methods. For comparison, we also present the corresponding realization of the gates formed by using $\mathrm{SWAP} = \mathsf{C}(X)_{12}\mathsf{C}(X)_{21}\mathsf{C}(X)_{12}$ with $\mathsf{C}(X)_{12}$ as given in Tables 8 and 9.

| Gate | Forced Measurement Sequence | Weight | $\langle S, H, \mathsf{C}(Z)\rangle$ | Weight | $\langle S, B, W\rangle$ | Weight |
|---|---|---|---|---|---|---|
| SWAP$(u)$ | $\Pi_+^{(34)}\cap_{-s_1}^{(14)}\Pi_+^{(16;1'6')}\Pi_{s_1}^{(35;1'5')}$ | $1.20\times10^{12}$ | $\mathsf{C}(X)^3$ | $1.01\times10^{76}$ | $\mathsf{C}(X)^3$ | $1.68\times10^{109}$ |
| $(d)$ | $\Pi_+^{(3'4')}\Pi_{-s_1}^{(1'4')}\Pi_+^{(16;1'6')}\Pi_{s_1}^{(15;3'5')}$ | $1.20\times10^{12}$ | | $1.01\times10^{76}$ | | $1.68\times10^{109}$ |
| $(r)$ | $\Pi_+^{(3'4')}\Pi_{s_1}^{(1'4')}\Pi_+^{(26;1'5')}\Pi_{s_1}^{(56;2'3')}$ | $1.87\times10^{12}$ | | $7.03\times10^{74}$ | | $1.16\times10^{110}$ |
| $(l)$ | $\Pi_+^{(34)}\Pi_{s_1}^{(14)}\Pi_+^{(15;2'6')}\Pi_{s_1}^{(23;5'6')}$ | $1.87\times10^{12}$ | | $7.03\times10^{74}$ | | $1.16\times10^{110}$ |
| $(a)$ | | $1.50\times10^{12}$ | | $2.66\times10^{75}$ | | $4.42\times10^{109}$ |
| $W(u)$ | $\cap_+^{(34)}\Pi_{s_1}^{(13)}\Pi_{s_1}^{(23;1'2')}$ | $2.76\times10^{6}$ | $S_1S_2\mathsf{C}(Z)$ | $8.00\times10^{18}$ | $W$ | $2.76\times10^{6}$ |
| $(d)$ | $\cap_+^{(3'4')}\Pi_{s_1}^{(1'3')}\Pi_{s_1}^{(12;2'3')}$ | $2.76\times10^{6}$ | | $8.00\times10^{18}$ | | $2.76\times10^{6}$ |
| $(r)$ | $\cap_+^{(3'4')}\Pi_{s_1}^{(1'3')}\Pi_{s_1}^{(56;2'3')}$ | $5.25\times10^{6}$ | | $3.28\times10^{18}$ | | $5.25\times10^{6}$ |
| $(l)$ | $\Pi_+^{(34)}\Pi_{s_1}^{(13)}\Pi_{s_1}^{(23;5'6')}$ | $5.25\times10^{6}$ | | $3.28\times10^{18}$ | | $5.25\times10^{6}$ |
| $(a)$ | | $3.81\times10^{6}$ | | $5.13\times10^{18}$ | | $3.81\times10^{6}$ |
| **Pauli Class** | **Tracked Measurement Sequence** | **Weight** | $\langle S, H, \mathsf{C}(Z)\rangle$ | **Weight** | $\langle S, B, W\rangle$ | **Weight** |
| $[\mathrm{SWAP}](u)$ | $\Pi_{s_4}^{(34)}\Pi_{s_3}^{(14)}\Pi_{s_2}^{(15;1'5')}\Pi_{s_1}^{(23;1'2')}$ | $6.79\times10^{4}$ | $\mathsf{C}(X)^3$ | $4.33\times10^{24}$ | $\mathsf{C}(X)^3$ | $7.57\times10^{49}$ |
| $(d)$ | $\Pi_{s_4'}^{(3'4')}\Pi_{s_3}^{(1'4')}\Pi_{s_2}^{(15;1'5')}\Pi_{s_1}^{(12;2'3')}$ | $6.79\times10^{4}$ | | $4.33\times10^{24}$ | | $7.57\times10^{49}$ |
| $(r)$ | $\Pi_{s_4'}^{(3'4')}\Pi_{s_3}^{(1'4')}\Pi_{s_2}^{(26;1'5')}\Pi_{s_1}^{(56;2'3')}$ | $8.11\times10^{4}$ | | $1.12\times10^{24}$ | | $2.01\times10^{50}$ |
| $(l)$ | $\Pi_{s_4}^{(34)}\Pi_{s_3}^{(14)}\Pi_{s_2}^{(15;2'6')}\Pi_{s_1}^{(23;5'6')}$ | $8.11\times10^{4}$ | | $1.12\times10^{24}$ | | $2.01\times10^{50}$ |
| $(a)$ | | $7.42\times10^{4}$ | | $2.21\times10^{24}$ | | $1.23\times10^{50}$ |
| $[W](u)$ | $\Pi_{s_3}^{(34)}\Pi_{s_2}^{(13)}\Pi_{s_1}^{(23;1'2')}$ | $8.95\times10^{2}$ | $S_1S_2\mathsf{C}(Z)$ | $1.63\times10^{8}$ | $W$ | $8.95\times10^{2}$ |
| $(d)$ | $\Pi_{s_3}^{(3'4')}\Pi_{s_2}^{(1'3')}\Pi_{s_1}^{(12;2'3')}$ | $8.95\times10^{2}$ | | $1.63\times10^{8}$ | | $8.95\times10^{2}$ |
| $(r)$ | $\Pi_{s_3}^{(3'4')}\Pi_{s_2}^{(2'3')}\Pi_{s_1}^{(56;1'3')}$ | $1.24\times10^{3}$ | | $1.04\times10^{8}$ | | $1.24\times10^{3}$ |
| $(l)$ | $\Pi_{s_3}^{(34)}\Pi_{s_2}^{(23)}\Pi_{s_1}^{(13;5'6')}$ | $1.24\times10^{3}$ | | $1.04\times10^{8}$ | | $1.24\times10^{3}$ |
| $(a)$ | | $1.05\times10^{3}$ | | $1.30\times10^{8}$ | | $1.05\times10^{3}$ |

Table 12: Pauli gate corrections tracked for the corresponding SWAP gates implemented using Majorana-Pauli tracking methods. The implicit action on the two ancillary qubits is $X^{\frac{1-s_ns_0}{2}}\Pi_{s_0}^{(34)}\otimes X^{\frac{1-s_n's_0'}{2}}\Pi_{s_0'}^{(3'4')}$ for sequences of length $n$. (When there is not a final projector for one of the ancillary pairs, it is equavalent to there being a projector for that ancillary pair onto its initial projection channel, e.g. $s_n = s_0$ or $s_n' = s_0'$.)

| Gate | Tracked Measurement Sequence |
|---|---|
| $(Y\otimes\mathbb{1})^{\frac{1-s_4s_0}{2}}(Z\otimes Z)^{\frac{1-s_2s_0s_0'}{2}}(Y\otimes Y)^{\frac{1-s_3s_1s_0}{2}}\mathrm{SWAP}_u$ | $\Pi_{s_4}^{(34)}\Pi_{s_3}^{(14)}\Pi_{s_2}^{(15;1'5')}\Pi_{s_1}^{(23;1'2')}\boldsymbol{\Pi}_{\boldsymbol{s_0}}^{(\mathrm{anc})}$ |
| $(\mathbb{1}\otimes Y)^{\frac{1-s_4's_0'}{2}}(Z\otimes Z)^{\frac{1-s_2s_0s_0'}{2}}(Y\otimes Y)^{\frac{1-s_3s_1s_0'}{2}}\mathrm{SWAP}_d$ | $\Pi_{s_4'}^{(3'4')}\Pi_{s_3}^{(1'4')}\Pi_{s_2}^{(15;1'5')}\Pi_{s_1}^{(12;2'3')}\boldsymbol{\Pi}_{\boldsymbol{s_0}}^{(\mathrm{anc})}$ |
| $(\mathbb{1}\otimes Y)^{\frac{1-s_4's_0'}{2}}(Z\otimes Z)^{\frac{1+s_2s_0'}{2}}(Y\otimes Y)^{\frac{1-s_3s_1s_0s_0'}{2}}\mathrm{SWAP}_r$ | $\Pi_{s_4'}^{(3'4')}\Pi_{s_3}^{(1'4')}\Pi_{s_2}^{(26;1'5')}\Pi_{s_1}^{(56;2'3')}\boldsymbol{\Pi}_{\boldsymbol{s_0}}^{(\mathrm{anc})}$ |
| $(Y\otimes\mathbb{1})^{\frac{1-s_4s_0}{2}}(Z\otimes Z)^{\frac{1+s_2s_0}{2}}(Y\otimes Y)^{\frac{1-s_3s_1s_0s_0'}{2}}\mathrm{SWAP}_l$ | $\Pi_{s_4}^{(34)}\Pi_{s_3}^{(14)}\Pi_{s_2}^{(15;2'6')}\Pi_{s_1}^{(23;5'6')}\boldsymbol{\Pi}_{\boldsymbol{s_0}}^{(\mathrm{anc})}$ |
| $(Y\otimes\mathbb{1})^{\frac{1-s_3s_0}{2}}(Z\otimes Z)^{\frac{1-s_2s_1}{2}}W_u$ | $\Pi_{s_3}^{(34)}\Pi_{s_2}^{(13)}\Pi_{s_1}^{(23;1'2')}\boldsymbol{\Pi}_{\boldsymbol{s_0}}^{(\mathrm{anc})}$ |
| $(\mathbb{1}\otimes Y)^{\frac{1-s_3's_0'}{2}}(Z\otimes Z)^{\frac{1-s_2s_1}{2}}W_d$ | $\Pi_{s_3'}^{(3'4')}\Pi_{s_2}^{(1'3')}\Pi_{s_1}^{(12;2'3')}\boldsymbol{\Pi}_{\boldsymbol{s_0}}^{(\mathrm{anc})}$ |
| $(\mathbb{1}\otimes X)^{\frac{1-s_3's_0'}{2}}(Z\otimes Z)^{\frac{1+s_2s_1s_0}{2}}W_r$ | $\Pi_{s_3'}^{(3'4')}\Pi_{s_2}^{(2'3')}\Pi_{s_1}^{(56;1'3')}\boldsymbol{\Pi}_{\boldsymbol{s_0}}^{(\mathrm{anc})}$ |
| $(X\otimes\mathbb{1})^{\frac{1-s_3s_0}{2}}(Z\otimes Z)^{\frac{1+s_2s_1s_0'}{2}}W_l$ | $\Pi_{s_3}^{(34)}\Pi_{s_2}^{(23)}\Pi_{s_1}^{(13;5'6')}\boldsymbol{\Pi}_{\boldsymbol{s_0}}^{(\mathrm{anc})}$ |

Table 13: Minimal difficulty weight measurement sequences for each single-qubit Clifford gate when using forced-measurement methods. For comparison, we also present the corresponding realization of the gates formed by using the generating gate sets $\langle S,H\rangle$ and $\langle S,B\rangle$.

| Gate | Forced Measurement Sequence | Weight | $\langle S,H\rangle$ | Weight | $\langle S,B\rangle$ | Weight |
|---|---|---|---|---|---|---|
| $X$ | $\widehat{\Pi}_+^{(34)}\Pi_{s_2}^{(45)}\widehat{\Pi}_-^{(34)}\Pi_{s_1}^{(23)}$ | $3.10\times10^4$ | $HSSH$ | $8.14\times10^{19}$ | $BB$ | $1.30\times10^8$ |
| $Y$ | $\widehat{\Pi}_+^{(34)}\Pi_{-s_1}^{(36)}\Pi_{s_2}^{(26)}\Pi_{s_1}^{(36)}$ | $1.95\times10^4$ | $HSSHSS$ | $6.64\times10^{27}$ | $BBSS$ | $1.06\times10^6$ |
| $Z$ | $\widehat{\Pi}_+^{(34)}\Pi_{s_2}^{(36)}\widehat{\Pi}_+^{(34)}\Pi_{s_1}^{(45)}$ | $1.95\times10^4$ | $SS$ | $8.15\times10^7$ | $SS$ | $8.15\times10^7$ |
| $S$ | $\overline{\Pi}_+^{(34)}\Pi_{-s_1}^{(46)}\Pi_{s_1}^{(45)}$ | $9.03\times10^3$ | $S$ | $9.03\times10^3$ | $S$ | $9.03\times10^3$ |
| $XS$ | $\widehat{\Pi}_+^{(34)}\overline{\Pi}_{-s_1}^{(36)}\widehat{\Pi}_{s_2}^{(26)}\Pi_{s_2}^{(16)}\Pi_{s_1}^{(36)}$ | $9.99\times10^5$ | $HSSHS$ | $7.35\times10^{23}$ | $BBS$ | $1.17\times10^{12}$ |
| $YS$ | $\widehat{\Pi}_+^{(34)}\overline{\Pi}_{-s_1}^{(36)}\widehat{\Pi}_{-s_2}^{(26)}\Pi_{s_2}^{(16)}\Pi_{s_1}^{(36)}$ | $9.99\times10^5$ | $SHSSH$ | $7.35\times10^{23}$ | $SBB$ | $1.17\times10^{12}$ |
| $ZS$ | $\overline{\Pi}_+^{(34)}\Pi_{s_1}^{(46)}\Pi_{s_1}^{(45)}$ | $9.03\times10^3$ | $S^\dagger$ | $9.03\times10^3$ | $S^\dagger$ | $9.03\times10^3$ |
| $H$ | $\widehat{\Pi}_+^{(34)}\Pi_{s_2}^{(45)}\overline{\Pi}_-^{(34)}\widehat{\Pi}_1^{(36)}\Pi_{s_1}^{(23)}$ | $9.99\times10^5$ | $H$ | $9.99\times10^5$ | $SBS$ | $9.30\times10^{11}$ |
| $XH$ | $\widehat{\Pi}_+^{(34)}\Pi_{-s_1}^{(36)}\Pi_{s_1}^{(23)}$ | $7.16\times10^3$ | $HSS$ | $8.15\times10^{13}$ | $S^\dagger B^\dagger S$ | $9.30\times10^{11}$ |
| $YH$ | $\widehat{\Pi}_+^{(34)}\Pi_{s_2}^{(45)}\widehat{\Pi}_-^{(34)}\Pi_{-s_1}^{(36)}\Pi_{s_1}^{(23)}$ | $9.99\times10^5$ | $SSHSS$ | $6.64\times10^{21}$ | $SB^\dagger S$ | $9.30\times10^{11}$ |
| $ZH$ | $\widehat{\Pi}_+^{(34)}\widehat{\Pi}_{s_1}^{(36)}\Pi_{s_1}^{(23)}$ | $7.16\times10^3$ | $SSH$ | $8.15\times10^{13}$ | $S^\dagger BS$ | $9.30\times10^{11}$ |
| $SH$ | $\overline{\Pi}_+^{(34)}\Pi_{-s_1}^{(35)}\overline{\Pi}_1^{(36)}\Pi_{s_1}^{(23)}$ | $4.63\times10^5$ | $SH$ | $9.02\times10^9$ | $B^\dagger S^\dagger$ | $1.03\times10^8$ |
| $XSH$ | $\overline{\Pi}_+^{(34)}\Pi_1^{(35)}\Pi_{-s_1}^{(36)}\Pi_{s_1}^{(23)}$ | $4.63\times10^5$ | $S^\dagger HSS$ | $7.36\times10^{17}$ | $BS^\dagger$ | $1.03\times10^8$ |
| $YSH$ | $\overline{\Pi}_+^{(34)}\Pi_{-s_1}^{(35)}\Pi_{-s_1}^{(36)}\Pi_{s_1}^{(23)}$ | $4.63\times10^5$ | $SHSS$ | $7.36\times10^{17}$ | $B^\dagger S$ | $1.03\times10^8$ |
| $ZSH$ | $\overline{\Pi}_+^{(34)}\Pi_1^{(35)}\overline{\Pi}_1^{(36)}\Pi_{s_1}^{(23)}$ | $4.63\times10^5$ | $S^\dagger H$ | $9.02\times10^9$ | $BS$ | $1.03\times10^8$ |
| $HS$ | $\widehat{\Pi}_+^{(34)}\overline{\Pi}_{-s_1}^{(36)}\widehat{\Pi}_1^{(23)}\Pi_{s_1}^{(13)}$ | $5.85\times10^5$ | $HS$ | $9.02\times10^9$ | $S^\dagger B^\dagger$ | $1.03\times10^8$ |
| $XHS$ | $\widehat{\Pi}_+^{(34)}\overline{\Pi}_{s_1}^{(36)}\widehat{\Pi}_{-s_1}^{(23)}\Pi_{s_1}^{(13)}$ | $5.85\times10^5$ | $HS^\dagger$ | $9.02\times10^9$ | $SB$ | $1.03\times10^8$ |
| $YHS$ | $\widehat{\Pi}_+^{(34)}\overline{\Pi}_{s_1}^{(36)}\overline{\Pi}_{s_1}^{(23)}\Pi_{s_1}^{(13)}$ | $5.85\times10^5$ | $SSHS^\dagger$ | $7.36\times10^{17}$ | $S^\dagger B$ | $1.03\times10^8$ |
| $ZHS$ | $\widehat{\Pi}_+^{(34)}\overline{\Pi}_{-s_1}^{(36)}\overline{\Pi}_{s_1}^{(23)}\Pi_{s_1}^{(13)}$ | $5.85\times10^5$ | $SSHS$ | $7.36\times10^{17}$ | $SB^\dagger$ | $1.03\times10^8$ |
| $SHS$ | $\widehat{\Pi}_+^{(34)}\overline{\Pi}_{-s_1}^{(36)}\Pi_{s_1}^{(13)}$ | $1.14\times10^4$ | $SHS$ | $8.15\times10^{13}$ | $B^\dagger$ | $1.14\times10^4$ |
| $XSHS$ | $\widehat{\Pi}_+^{(34)}\overline{\Pi}_{s_1}^{(36)}\Pi_{s_1}^{(13)}$ | $1.14\times10^4$ | $S^\dagger HS^\dagger$ | $8.15\times10^{13}$ | $B$ | $1.14\times10^4$ |
| $YSHS$ | $\widehat{\Pi}_+^{(34)}\widehat{\Pi}_{s_1}^{(36)}\widehat{\Pi}_{-s_2}^{(26)}\Pi_{s_2}^{(12)}\Pi_{s_1}^{(13)}$ | $1.26\times10^6$ | $SHS^\dagger$ | $8.15\times10^{13}$ | $B^\dagger SS$ | $1.14\times10^4$ |
| $ZSHS$ | $\widehat{\Pi}_+^{(34)}\widehat{\Pi}_{s_1}^{(36)}\Pi_{s_2}^{(26)}\Pi_{s_2}^{(12)}\Pi_{s_1}^{(13)}$ | $1.26\times10^6$ | $S^\dagger HS$ | $8.15\times10^{13}$ | $BSS$ | $1.14\times10^4$ |
| Average | | $1.45\times10^5$ | | $3.28\times10^{14}$ | | $1.12\times10^9$ |

## C.2 One-Sided Hexon with Configuration $\langle 1,2,6,3,4,5\rangle$

The MZM labeling configuration $\langle 1,2,6,3,4,5\rangle$ is optimal for the one-sided hexon architecture, when using the forced-measurement methods for the gates: the Hadamard gate $H$, the geometric average of single-qubit Clifford gates $C_1$; or when using the Majorana-Pauli tracking methods, for the Pauli cosets of gates: the Hadamard gate $H$, the geometric average of single-qubit Clifford gates $C_1$, the controlled-$X$ gate $C(X)$, and the geometric average of the controlled-Pauli gates $C(P)$. This configuration within an array looks like:

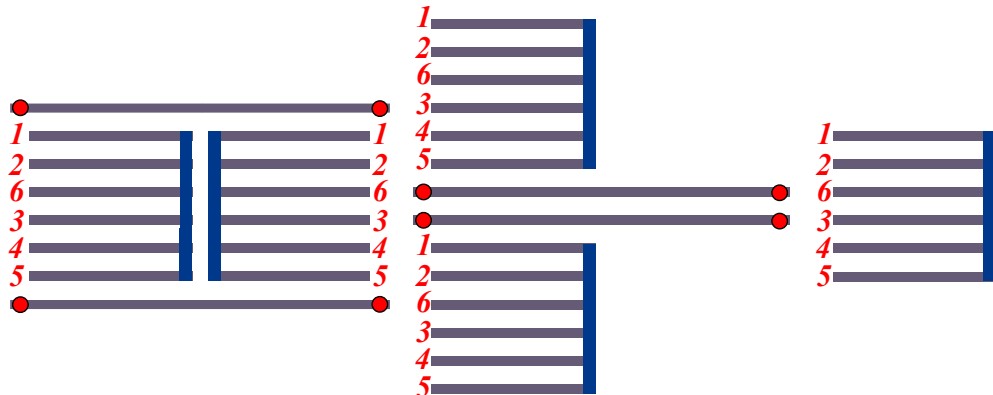

Table 14: Minimal difficulty weight measurement sequences for each Pauli coset of single-qubit Clifford gates when using Majorana-Pauli tracking methods. For comparison, we also present the corresponding realization of the gates formed by using the generating gate sets $\langle S, H \rangle$ and $\langle S, B \rangle$.

| Pauli Class | Unforced Measurement Sequence | Weight | $\langle S,H\rangle$ Decomp. | Weight | $\langle S,B\rangle$ Decomp. | Weight |
|---|---|---|---|---|---|---|
| $[S]$ | $\Pi_{s_3}^{(34)}\Pi_{s_2}^{(36)}\Pi_{s_1}^{(35)}$ | $5.13\times10^1$ | $S$ | $5.13\times10^1$ | $S$ | $5.13\times10^1$ |
| $[H]$ | $\Pi_{s_3}^{(34)}\Pi_{s_2}^{(36)}\Pi_{s_1}^{(23)}$ | $5.13\times10^1$ | $H$ | $5.13\times10^1$ | $SBS$ | $1.70\times10^5$ |
| $[SH]$ | $\Pi_{s_4}^{(34)}\Pi_{s_3}^{(35)}\Pi_{s_2}^{(36)}\Pi_{s_1}^{(23)}$ | $2.22\times10^2$ | $SH$ | $2.63\times10^3$ | $BS$ | $3.32\times10^3$ |
| $[HS]$ | $\Pi_{s_4}^{(34)}\Pi_{s_3}^{(36)}\Pi_{s_2}^{(35)}\Pi_{s_1}^{(23)}$ | $2.22\times10^2$ | $HS$ | $2.63\times10^3$ | $SB$ | $3.32\times10^3$ |
| $[SHS]$ | $\Pi_{s_3}^{(34)}\Pi_{s_2}^{(35)}\Pi_{s_1}^{(23)}$ | $6.47\times10^1$ | $SHS$ | $1.35\times10^5$ | $B$ | $6.47\times10^1$ |
| Average | | $9.66\times10^1$ | | $1.20\times10^3$ | | $1.44\times10^3$ |

Table 15: Pauli gate corrections tracked for the corresponding single-qubit gates implemented using Majorana-Pauli tracking methods. The implicit action on the ancillary qubit is $X^{\frac{1-s_n s_0}{2}}\Pi_{s_0}^{(34)}$ for sequences of length $n$.

| Gate | Tracked Measurement Sequence |
|---|---|
| $Z^{\frac{1-s_3 s_0}{2}}Z^{\frac{1+s_2 s_1 s_0}{2}}S$ | $\Pi_{s_3}^{(34)}\Pi_{s_2}^{(36)}\Pi_{s_1}^{(35)}\Pi_{s_0}^{(34)}$ |
| $Z^{\frac{1-s_3 s_0}{2}}Y^{\frac{1-s_2 s_1}{2}}ZH$ | $\Pi_{s_3}^{(34)}\Pi_{s_2}^{(36)}\Pi_{s_1}^{(23)}\Pi_{s_0}^{(34)}$ |
| $Z^{\frac{1-s_3 s_2 s_0}{2}}X^{\frac{1-s_2 s_1}{2}}ZSH$ | $\Pi_{s_4}^{(34)}\Pi_{s_3}^{(35)}\Pi_{s_2}^{(36)}\Pi_{s_1}^{(23)}\Pi_{s_0}^{(34)}$ |
| $Z^{\frac{1-s_4 s_0}{2}}Z^{\frac{1-s_3 s_2 s_0}{2}}Y^{\frac{1-s_2 s_1 s_0}{2}}YHS$ | $\Pi_{s_4}^{(34)}\Pi_{s_3}^{(36)}\Pi_{s_2}^{(35)}\Pi_{s_1}^{(23)}\Pi_{s_0}^{(34)}$ |
| $X^{\frac{1+s_2 s_1 s_0}{2}}SHS$ | $\Pi_{s_3}^{(34)}\Pi_{s_2}^{(35)}\Pi_{s_1}^{(23)}\Pi_{s_0}^{(34)}$ |

Table 16: Minimal difficulty weight measurement sequences for controlled-Pauli two-qubit gates when using forced-measurement methods. The labels $(u)$, $(d)$, $(r)$, and $(l)$ indicates that for a hexon acting as the control qubit of the $\mathsf{C}(P)$ gate, the corresponding target qubit is the nearest neighbor hexon in the up, down, right, and left direction, respectively. Notice that the choice of control and target qubit is arbitrary for $\mathsf{C}(Z)$, so $(u)$ and $(d)$ are related by symmetry, as are $(r)$, and $(l)$. The average difficulty weight of the four directions is labeled by $(a)$. For comparison, we also present the corresponding realization of the gates formed from the generating gate sets $\langle S, H, \mathsf{C}(Z)\rangle$ and $\langle S, B, W\rangle$. $^{A}\mathsf{C}(Z) = A_2\mathsf{C}(Z)A_2^{\dagger}$ denotes conjugation of $\mathsf{C}(Z)$ by $A$ on the target qubit.

| Gate | Forced Measurement Sequence | Weight | $\langle S, H, \mathsf{C}(Z)\rangle$ | Weight | $\langle S, B, W\rangle$ | Weight |
|---|---|---|---|---|---|---|
| $\mathsf{C}(X)(u)$ | $\overset{\text{ef}}{\Pi}{}_+^{(3'4')}\overset{\frown}{\Pi}{}_{s_1}^{(4'5')}\overset{\text{ef}}{\Pi}{}_+^{(2'5')}\Pi_{s_1}^{(12;4'5')}$ | $2.88 \times 10^7$ | $^{H}\mathsf{C}(Z)$ | $1.43 \times 10^{19}$ | $S_2B_2S_1W^{\dagger}S_2^{\dagger}B_2S_2$ | $1.51 \times 10^{29}$ |
| $(d)$ | $\overset{\text{ef}}{\Pi}{}_+^{(34)}\overset{\frown}{\Pi}{}_{s_1}^{(45)}\overset{\text{ef}}{\Pi}{}_+^{(56)}\Pi_{s_1}^{(45;1'6')}$ | $2.28 \times 10^7$ | | $1.43 \times 10^{19}$ | | $1.51 \times 10^{29}$ |
| $(r)$ | $\overset{\text{ef}}{\Pi}{}_+^{(3'4')}\overset{\text{ef}}{\Pi}{}_{s_1}^{(3'6')}\overset{\text{ef}}{\Pi}{}_+^{(1'6')}\Pi_{s_1}^{(56;3'6')}$ | $1.30 \times 10^9$ | | $2.08 \times 10^{21}$ | | $2.19 \times 10^{31}$ |
| $(l)$ | $\overset{\text{ef}}{\Pi}{}_+^{(34)}\overset{\text{ef}}{\Pi}{}_{-s_1}^{(23)}\overset{\text{ef}}{\Pi}{}_+^{(12)}\Pi_{s_1}^{(14;2'5')}$ | $7.76 \times 10^9$ | | $1.24 \times 10^{22}$ | | $2.64 \times 10^{32}$ |
| $(a)$ | | $2.85 \times 10^8$ | | $2.69 \times 10^{20}$ | | $3.39 \times 10^{30}$ |
| $\mathsf{C}(Y)(u)$ | $\overset{\text{ef}}{\Pi}{}_+^{(3'4')}\overset{\frown}{\Pi}{}_{-s_1}^{(4'5')}\overset{\text{ef}}{\Pi}{}_+^{(1'5')}\Pi_{s_1}^{(12;4'5')}$ | $5.76 \times 10^7$ | $^{SH}\mathsf{C}(Z)$ | $1.16 \times 10^{27}$ | $B_2^{\dagger}W^{\dagger}S_2B_2S_1$ | $1.85 \times 10^{21}$ |
| $(d)$ | $\overset{\text{ef}}{\Pi}{}_+^{(34)}\overset{\frown}{\Pi}{}_{-s_1}^{(45)}\overset{\text{ef}}{\Pi}{}_+^{(56)}\Pi_{s_1}^{(45;1'5')}$ | $9.22 \times 10^7$ | | $1.16 \times 10^{27}$ | | $1.85 \times 10^{21}$ |
| $(r)$ | $\overset{\text{ef}}{\Pi}{}_+^{(3'4')}\overset{\text{ef}}{\Pi}{}_{-}^{(3'6')}\overset{\text{ef}}{\Pi}{}_{-}^{(2'6')}\Pi_{s_1}^{(56;3'6')}$ | $6.47 \times 10^8$ | | $1.69 \times 10^{29}$ | | $2.69 \times 10^{23}$ |
| $(l)$ | $\overset{\text{ef}}{\Pi}{}_+^{(34)}\overset{\text{ef}}{\Pi}{}_{-s_1}^{(23)}\overset{\text{ef}}{\Pi}{}_+^{(12)}\Pi_{s_1}^{(14;1'5')}$ | $4.87 \times 10^9$ | | $1.01 \times 10^{30}$ | | $3.23 \times 10^{24}$ |
| $(a)$ | | $3.60 \times 10^8$ | | $4.16 \times 10^{22}$ | | $4.15 \times 10^{22}$ |
| $\mathsf{C}(Z)(u)$ | $\overset{\text{ef}}{\Pi}{}_+^{(3'4')}\overset{\frown}{\Pi}{}_{-s_1}^{(4'5')}\overset{\text{ef}}{\Pi}{}_+^{(5'6')}\Pi_{s_1}^{(12;4'5')}$ | $1.43 \times 10^7$ | $\mathsf{C}(Z)$ | $1.43 \times 10^7$ | $S_1S_2W^{\dagger}$ | $1.43 \times 10^{13}$ |
| $(d)$ | $\overset{\text{ef}}{\Pi}{}_+^{(34)}\overset{\frown}{\Pi}{}_{-s_1}^{(45)}\overset{\text{ef}}{\Pi}{}_+^{(56)}\Pi_{s_1}^{(45;1'2')}$ | $1.43 \times 10^7$ | | $1.43 \times 10^7$ | | $1.43 \times 10^{13}$ |
| $(r)$ | $\overset{\text{ef}}{\Pi}{}_+^{(34)}\overset{\text{ef}}{\Pi}{}_{s_1}^{(23)}\overset{\text{ef}}{\Pi}{}_+^{(12)}\Pi_{s_1}^{(23;5'6')}$ | $2.08 \times 10^9$ | | $2.08 \times 10^9$ | | $2.07 \times 10^{15}$ |
| $(l)$ | $\overset{\text{ef}}{\Pi}{}_+^{(34)}\overset{\text{ef}}{\Pi}{}_{-s_1}^{(23)}\overset{\text{ef}}{\Pi}{}_+^{(12)}\Pi_{s_1}^{(14;5'6')}$ | $1.24 \times 10^{10}$ | | $1.24 \times 10^{10}$ | | $2.49 \times 10^{16}$ |
| $(a)$ | | $2.69 \times 10^8$ | | $2.69 \times 10^8$ | | $3.20 \times 10^{14}$ |
| Average$(u)$ | | $2.87 \times 10^7$ | | $6.19 \times 10^{17}$ | | $1.59 \times 10^{21}$ |
| $(d)$ | | $3.11 \times 10^7$ | | $6.19 \times 10^{17}$ | | $1.59 \times 10^{21}$ |
| $(r)$ | | $1.20 \times 10^9$ | | $9.01 \times 10^{19}$ | | $2.30 \times 10^{23}$ |
| $(l)$ | | $7.77 \times 10^9$ | | $5.38 \times 10^{20}$ | | $2.77 \times 10^{24}$ |
| $(a)$ | | $3.02 \times 10^8$ | | $1.17 \times 10^{19}$ | | $3.56 \times 10^{22}$ |

Table 17: Minimal difficulty weight measurement sequences for Pauli cosets of controlled-Pauli two-qubit gates when using Majorana-Pauli tracking methods. The labels $(u)$, $(d)$, $(r)$, and $(l)$ indicates that for a hexon acting as the control qubit of the $C(P)$ gate, the corresponding target qubit is the nearest neighbor hexon in the up, down, right, and left direction, respectively. Notice that the choice of control and target qubit is arbitrary for $C(Z)$, so $(u)$ and $(d)$ are related by symmetry, as are $(r)$, and $(l)$. The average difficulty weight of the four directions is labeled by $(a)$. For comparison, we also present the corresponding realization of the gates formed by using the generating gate sets $\langle S, H, C(Z)\rangle$ and $\langle S, B, W\rangle$. $^A C(Z) = A_2 C(Z) A_2^\dagger$ denotes conjugation of $C(Z)$ by $A$ on the target qubit.

| Pauli Class | Tracked Measurement Sequence | Weight | $\langle S,H,C(Z)\rangle$ | Weight | $\langle S,B,W\rangle$ | Weight |
|---|---|---|---|---|---|---|
| $[C(X)](u)$ | $\Pi_{s_4}^{(3'4')}\Pi_{s_3}^{(4'5')}\Pi_{s_2}^{(2'5')}\Pi_{s_1}^{(12;4'5')}$ | $1.24\times10^3$ | $^H C(Z)$ | $2.58\times10^6$ | $S_2 B_2 S_1 W^\dagger S_2^\dagger B_2 S_2$ | $6.55\times10^{12}$ |
| $(d)$ | $\Pi_{s_4}^{(34)}\Pi_{s_3}^{(45)}\Pi_{s_2}^{(56;1'6')}\Pi_{s_1}^{(36)}$ | $1.24\times10^3$ | | $2.58\times10^6$ | | $6.55\times10^{12}$ |
| $(r)$ | $\Pi_{s_4}^{(3'4')}\Pi_{s_3}^{(3'6')}\Pi_{s_2}^{(1'6')}\Pi_{s_1}^{(56;3'6')}$ | $9.34\times10^3$ | | $3.11\times10^7$ | | $7.89\times10^{13}$ |
| $(l)$ | $\Pi_{s_4}^{(34)}\Pi_{s_3}^{(45)}\Pi_{s_2}^{(56;2'5')}\Pi_{s_1}^{(36)}$ | $2.28\times10^4$ | | $7.58\times10^7$ | | $3.07\times10^{14}$ |
| $(a)$ | | $4.25\times10^3$ | | $1.12\times10^7$ | | $3.19\times10^{13}$ |
| $[C(Y)](u)$ | $\Pi_{s_4}^{(3'4')}\Pi_{s_3}^{(4'5')}\Pi_{s_2}^{(1'5')}\Pi_{s_1}^{(12;4'5')}$ | $1.56\times10^3$ | $^{SH} C(Z)$ | $6.78\times10^9$ | $B_2^\dagger W^\dagger S_2 B_2 S_1$ | $2.49\times10^9$ |
| $(d)$ | $\Pi_{s_4}^{(34)}\Pi_{s_3}^{(45)}\Pi_{s_2}^{(56;1'5')}\Pi_{s_1}^{(36)}$ | $2.49\times10^3$ | | $6.78\times10^9$ | | $2.49\times10^9$ |
| $(r)$ | $\Pi_{s_4}^{(3'4')}\Pi_{s_3}^{(3'6')}\Pi_{s_2}^{(2'6')}\Pi_{s_1}^{(56;3'6')}$ | $7.40\times10^3$ | | $8.17\times10^{10}$ | | $3.00\times10^{10}$ |
| $(l)$ | $\Pi_{s_4}^{(34)}\Pi_{s_3}^{(45)}\Pi_{s_2}^{(56;1'5')}\Pi_{s_1}^{(36)}$ | $1.81\times10^4$ | | $1.99\times10^{11}$ | | $1.17\times10^{11}$ |
| $(a)$ | | $4.77\times10^3$ | | $2.94\times10^{10}$ | | $1.21\times10^{10}$ |
| $[C(Z)](u)$ | $\Pi_{s_4}^{(3'4')}\Pi_{s_3}^{(4'5')}\Pi_{s_2}^{(12;5'6')}\Pi_{s_1}^{(3'6')}$ | $9.79\times10^2$ | $C(Z)$ | $9.79\times10^2$ | $S_1 S_2 W^\dagger$ | $5.95\times10^5$ |
| $(d)$ | $\Pi_{s_4}^{(34)}\Pi_{s_3}^{(45)}\Pi_{s_2}^{(56;1'2')}\Pi_{s_1}^{(36)}$ | $9.79\times10^2$ | | $9.79\times10^2$ | | $5.95\times10^5$ |
| $(r)$ | $\Pi_{s_4}^{(34)}\Pi_{s_3}^{(23)}\Pi_{s_2}^{(12)}\Pi_{s_1}^{(23;5'6')}$ | $1.18\times10^4$ | | $1.18\times10^4$ | | $7.16\times10^6$ |
| $(l)$ | $\Pi_{s_4}^{(34)}\Pi_{s_3}^{(23)}\Pi_{s_2}^{(12)}\Pi_{s_1}^{(14;5'6')}$ | $2.88\times10^4$ | | $2.88\times10^4$ | | $2.79\times10^7$ |
| $(a)$ | | $4.25\times10^3$ | | $4.25\times10^3$ | | $2.90\times10^6$ |
| Average$(u)$ | | $1.24\times10^3$ | | $2.58\times10^6$ | | $2.13\times10^9$ |
| $(d)$ | | $1.45\times10^3$ | | $2.58\times10^6$ | | $2.13\times10^9$ |
| $(r)$ | | $9.34\times10^3$ | | $3.11\times10^7$ | | $2.57\times10^{10}$ |
| $(l)$ | | $2.28\times10^4$ | | $7.57\times10^7$ | | $1.00\times10^{11}$ |
| $(a)$ | | $4.42\times10^3$ | | $1.12\times10^7$ | | $1.04\times10^{10}$ |

Table 18: Pauli gate corrections tracked for the corresponding controlled-Pauli gates implemented using Majorana-Pauli tracking methods. The implicit action on the two ancillary qubits is $X^{\frac{1-s_n s_0}{2}}\Pi^{(34)}_{s_0} \otimes X^{\frac{1-s_n' s_0'}{2}}\Pi^{(3'4')}_{s_0'}$ for sequences of length $n$. (When there is not a final projector for one of the ancillary pairs, it is equivalent to there being a projector for that ancillary pair onto its initial projection channel, e.g. $s_n = s_0$ or $s_n' = s_0'$.)

| Gate | Tracked Measurement Sequence |
|---|---|
| $\left(Z^{\frac{1-s_2 s_0'}{2}} \otimes X^{\frac{1-s_3 s_1}{2}}\right)C(X)_u$ | $\Pi^{(3'4')}_{s_4'}\Pi^{(4'5')}_{s_3}\Pi^{(2'5')}_{s_2}\Pi^{(12;4'5')}_{s_1}\mathbf{\Pi}^{(\mathrm{anc})}_{s_0}$ |
| $\left(Z^{\frac{1+s_2 s_0}{2}} \otimes X^{\frac{1-s_3 s_1 s_0}{2}}\right)C(X)_d$ | $\Pi^{(34)}_{s_4}\Pi^{(45)}_{s_3}\Pi^{(56;1'6')}_{s_2}\Pi^{(36)}_{s_1}\mathbf{\Pi}^{(\mathrm{anc})}_{s_0}$ |
| $\left(Z^{\frac{1-s_2}{2}} \otimes Z^{\frac{1-s_4' s_0'}{2}}X^{\frac{1-s_3 s_1 s_0}{2}}\right)C(X)_r$ | $\Pi^{(3'4')}_{s_4'}\Pi^{(3'6')}_{s_3}\Pi^{(1'6')}_{s_2}\Pi^{(56;3'6')}_{s_1}\mathbf{\Pi}^{(\mathrm{anc})}_{s_0}$ |
| $\left(Z^{\frac{1+s_2 s_0 s_0'}{2}} \otimes X^{\frac{1-s_3 s_1 s_0}{2}}\right)C(X)_l$ | $\Pi^{(34)}_{s_4}\Pi^{(45)}_{s_3}\Pi^{(56;2'5')}_{s_2}\Pi^{(36)}_{s_1}\mathbf{\Pi}^{(\mathrm{anc})}_{s_0}$ |
| $\left(Z^{\frac{1-s_2 s_0'}{2}} \otimes Y^{\frac{1-s_3 s_1}{2}}\right)C(Y)_u$ | $\Pi^{(3'4')}_{s_4'}\Pi^{(4'5')}_{s_3}\Pi^{(1'5')}_{s_2}\Pi^{(12;4'5')}_{s_1}\mathbf{\Pi}^{(\mathrm{anc})}_{s_0}$ |
| $\left(Z^{\frac{1+s_2 s_0 s_0'}{2}} \otimes Y^{\frac{1-s_3 s_1}{2}}\right)C(Y)_d$ | $\Pi^{(34)}_{s_4}\Pi^{(45)}_{s_3}\Pi^{(56;1'5')}_{s_2}\Pi^{(36)}_{s_1}\mathbf{\Pi}^{(\mathrm{anc})}_{s_0}$ |
| $\left(Z^{\frac{1+s_2}{2}}Y^{\frac{1-s_3 s_1 s_0}{2}} \otimes Z^{\frac{1-s_4' s_0'}{2}}\right)C(Y)_r$ | $\Pi^{(3'4')}_{s_4'}\Pi^{(3'6')}_{s_3}\Pi^{(2'6')}_{s_2}\Pi^{(56;3'6')}_{s_1}\mathbf{\Pi}^{(\mathrm{anc})}_{s_0}$ |
| $\left(Z^{\frac{1+s_2 s_0 s_0'}{2}} \otimes Y^{\frac{1-s_3 s_1 s_0}{2}}\right)C(Y)_l$ | $\Pi^{(34)}_{s_4}\Pi^{(45)}_{s_3}\Pi^{(56;1'5')}_{s_2}\Pi^{(36)}_{s_1}\mathbf{\Pi}^{(\mathrm{anc})}_{s_0}$ |
| $\left(Z^{\frac{1-s_3 s_1}{2}} \otimes Z^{\frac{1+s_2 s_0'}{2}}\right)C(Z)_u$ | $\Pi^{(3'4')}_{s_4'}\Pi^{(4'5')}_{s_3}\Pi^{(12;5'6')}_{s_2}\Pi^{(3'6')}_{s_1}\mathbf{\Pi}^{(\mathrm{anc})}_{s_0}$ |
| $\left(Z^{\frac{1+s_2 s_0}{2}} \otimes Z^{\frac{1-s_3 s_1}{2}}\right)C(Z)_d$ | $\Pi^{(34)}_{s_4}\Pi^{(45)}_{s_3}\Pi^{(56;1'2')}_{s_2}\Pi^{(36)}_{s_1}\mathbf{\Pi}^{(\mathrm{anc})}_{s_0}$ |
| $\left(X^{\frac{1-s_4 s_0}{2}}Z^{\frac{1-s_3 s_1 s_0'}{2}} \otimes Z^{\frac{1-s_2}{2}}\right)C(Z)_r$ | $\Pi^{(34)}_{s_4}\Pi^{(23)}_{s_3}\Pi^{(12)}_{s_2}\Pi^{(23;5'6')}_{s_1}\mathbf{\Pi}^{(\mathrm{anc})}_{s_0}$ |
| $\left(X^{\frac{1-s_4 s_0}{2}}Z^{\frac{1+s_3 s_1 s_0 s_0'}{2}} \otimes Z^{\frac{1-s_2}{2}}\right)C(Z)_l$ | $\Pi^{(34)}_{s_4}\Pi^{(23)}_{s_3}\Pi^{(12)}_{s_2}\Pi^{(14;5'6')}_{s_1}\mathbf{\Pi}^{(\mathrm{anc})}_{s_0}$ |

Table 19: Minimal difficulty weight measurement sequences for the two-qubit SWAP and $W$ gates when using forced-measurement methods and the Pauli cosets of SWAP and $W$ when using Majorana-Pauli tracking methods. For comparison, we also present the corresponding realization of the gates formed by using $\mathrm{SWAP} = \mathsf{C}(X)_{12}\mathsf{C}(X)_{21}\mathsf{C}(X)_{12}$ with $\mathsf{C}(X)_{12}$ as given in Tables 16 and 17.

| Gate | Forced Measurement Sequence | Weight | $\langle S, H, \mathsf{C}(Z)\rangle$ | Weight | $\langle S, B, W\rangle$ | Weight |
|---|---|---|---|---|---|---|
| SWAP $(u)$ | $\Pi_+^{(3'4')}\Pi_{s_2s_1}^{(12;4'5')}\Pi_{-s_2}^{(26)}$ $\Pi_+^{(12;5'6')}\Pi_{s_2}^{(1'5')}\Pi_{s_1}^{(16;3'5')}$ | $4.27 \times 10^{13}$ | $\mathsf{C}(X)^3$ | $2.92 \times 10^{57}$ | $\mathsf{C}(X)^3$ | $3.44 \times 10^{87}$ |
| $(d)$ | $\Pi_+^{(34)}\Pi_{s_2s_1}^{(45;1'2')}\Pi_{-s_2}^{(2'6')}$ $\Pi_+^{(56;1'2')}\Pi_{s_2}^{(15)}\Pi_{s_1}^{(35;1'6')}$ | $4.27 \times 10^{13}$ | | $2.92 \times 10^{57}$ | | $3.44 \times 10^{87}$ |
| $(r)$ | $\Pi_+^{(3'4')}\widehat{\Pi}_{s_1}^{(4'5')}\Pi_+^{(56;5'6')}\Pi_{s_1}^{(25;2'3')}$ | $7.86 \times 10^{14}$ | | $9.00 \times 10^{63}$ | | $1.05 \times 10^{94}$ |
| $(l)$ | $\widehat{\Pi}_+^{(3'4')}\Pi_{-s_1}^{(3'5')}\Pi_+^{(15;1'5')}\Pi_{s_1}^{(25;2'4')}$ | $5.25 \times 10^{15}$ | | $1.91 \times 10^{66}$ | | $1.84 \times 10^{97}$ |
| $(a)$ | | $2.95 \times 10^{14}$ | | $1.96 \times 10^{61}$ | | $3.89 \times 10^{91}$ |
| $W(u)$ | $\Pi_+^{(3'4')}\Pi_{-s_1}^{(3'6')}\Pi_{s_1}^{(12;3'5')}$ | $1.75 \times 10^5$ | $S_1S_2\mathsf{C}(Z)$ | $1.17 \times 10^{15}$ | $W$ | $1.75 \times 10^5$ |
| $(d)$ | $\Pi_+^{(34)}\Pi_{-s_1}^{(36)}\Pi_{s_1}^{(35;1'2')}$ | $1.75 \times 10^5$ | | $1.17 \times 10^{15}$ | | $1.75 \times 10^5$ |
| $(r)$ | $\Pi_+^{(34)}\Pi_{s_1}^{(45)}\Pi_{s_1}^{(46;5'6')}$ | $2.54 \times 10^7$ | | $1.70 \times 10^{17}$ | | $2.54 \times 10^7$ |
| $(l)$ | $\Pi_+^{(34)}\widehat{\Pi}_{s_1}^{(45)}\Pi_{s_1}^{(46;5'6')}$ | $3.05 \times 10^8$ | | $1.01 \times 10^{18}$ | | $3.05 \times 10^8$ |
| $(a)$ | | $3.92 \times 10^6$ | | $2.20 \times 10^{16}$ | | $3.92 \times 10^6$ |
| Pauli Class | Tracked Measurement Sequence | Weight | $\langle S, H, \mathsf{C}(Z)\rangle$ | Weight | $\langle S, B, W\rangle$ | Weight |
| [SWAP] $(u)$ | $\Pi_{s_6}^{(3'4')}\Pi_{s_5}^{(12;4'5')}\Pi_{s_4}^{(26)}$ $\Pi_{s_3}^{(12;5'6')}\Pi_{s_2}^{(1'5')}\Pi_{s_1}^{(16;3'5')}$ | $9.06 \times 10^5$ | $\mathsf{C}(X)^3$ | $1.72 \times 10^{19}$ | $\mathsf{C}(X)^3$ | $2.81 \times 10^{38}$ |
| $(d)$ | $\Pi_{s_6}^{(34)}\Pi_{s_5}^{(45;1'2')}\Pi_{s_4}^{(2'6')}$ $\Pi_{s_3}^{(56;1'2')}\Pi_{s_2}^{(15)}\Pi_{s_1}^{(35;1'6')}$ | $9.06 \times 10^5$ | | $1.72 \times 10^{19}$ | | $2.81 \times 10^{38}$ |
| $(r)$ | $\Pi_{s_4'}^{(3'4')}\Pi_{s_3}^{(4'5')}\Pi_{s_2}^{(25;2'5')}\Pi_{s_1}^{(56;3'6')}$ | $9.97 \times 10^5$ | | $3.01 \times 10^{22}$ | | $4.91 \times 10^{41}$ |
| $(l)$ | $\Pi_{s_4'}^{(3'4')}\Pi_{s_3}^{(4'5')}\Pi_{s_2}^{(15;1'5')}\Pi_{s_1}^{(25;2'3')}$ | $2.34 \times 10^6$ | | $4.36 \times 10^{21}$ | | $2.89 \times 10^{43}$ |
| $(a)$ | | $1.18 \times 10^6$ | | $1.40 \times 10^{21}$ | | $3.25 \times 10^{40}$ |
| [$W$] $(u)$ | $\Pi_{s_3'}^{(3'4')}\Pi_{s_2}^{(3'6')}\Pi_{s_1}^{(12;3'5')}$ | $2.26 \times 10^2$ | $S_1S_2\mathsf{C}(Z)$ | $2.58 \times 10^6$ | $W$ | $2.26 \times 10^2$ |
| $(d)$ | $\Pi_{s_3}^{(34)}\Pi_{s_2}^{(36)}\Pi_{s_1}^{(35;1'2')}$ | $2.26 \times 10^2$ | | $2.58 \times 10^6$ | | $2.26 \times 10^2$ |
| $(r)$ | $\Pi_{s_3}^{(34)}\Pi_{s_2}^{(45)}\Pi_{s_1}^{(46;5'6')}$ | $2.72 \times 10^3$ | | $3.11 \times 10^7$ | | $2.72 \times 10^3$ |
| $(l)$ | $\Pi_{s_3}^{(34)}\Pi_{s_2}^{(45)}\Pi_{s_1}^{(46;5'6')}$ | $1.06 \times 10^4$ | | $7.58 \times 10^7$ | | $1.06 \times 10^4$ |
| $(a)$ | | $1.10 \times 10^3$ | | $1.12 \times 10^7$ | | $1.10 \times 10^3$ |

Table 20: Pauli gate corrections tracked for the corresponding SWAP and $W$ gates implemented using Majorana-Pauli tracking methods. The implicit action on the two ancillary qubits is $X^{\frac{1-s_n s_0}{2}} \Pi^{(34)}_{s_0} \otimes X^{\frac{1-s'_n s'_0}{2}} \Pi^{(3'4')}_{s'_0}$ for sequences of length $n$. (When there is not a final projector for one of the ancillary pairs, it is equivalent to there being a projector for that ancillary pair onto its initial projection channel, e.g. $s_n = s_0$ or $s'_n = s'_0$.)

| Gate | Tracked Measurement Sequence |
|---|---|
| $\left( X^{\frac{1+s_4}{2}} Y^{\frac{1-s_5 s_3 s_1}{2}} Z^{\frac{1-s'_6 s_2}{2}} \otimes X^{\frac{1-s'_0}{2}} Y^{\frac{1-s_3}{2}} Z^{\frac{1+s_4 s_2}{2}} \right) \mathrm{SWAP}_u$ | $\begin{bmatrix} \Pi^{(3'4')}_{s'_6} \Pi^{(12;4'5')}_{s_5} \Pi^{(26)}_{s_4} \Pi^{(12;5'6')}_{s_3} \\ \Pi^{(1'5')}_{s_2} \Pi^{(16;3'5')}_{s_1} \mathbf{\Pi}^{(\mathrm{anc})}_{s_0} \end{bmatrix}$ |
| $\left( X^{\frac{1-s'_0}{2}} Y^{\frac{1-s_3}{2}} Z^{\frac{1+s_4 s_2}{2}} \otimes X^{\frac{1+s_4}{2}} Y^{\frac{1-s_5 s_3 s_1}{2}} Z^{\frac{1-s_6 s_2}{2}} \right) \mathrm{SWAP}_d$ | $\begin{bmatrix} \Pi^{(34)}_{s_6} \Pi^{(45;1'2')}_{s_5} \Pi^{(2'6')}_{s_4} \Pi^{(56;1'2')}_{s_3} \\ \Pi^{(15)}_{s_2} \Pi^{(35;1'6')}_{s_1} \mathbf{\Pi}^{(\mathrm{anc})}_{s_0} \end{bmatrix}$ |
| $(Z \otimes Z)^{\frac{1-s_2 s_0 s'_0}{2}} (X \otimes X)^{\frac{1-s_3 s_1 s_0}{2}} \mathrm{SWAP}_r$ | $\Pi^{(3'4')}_{s'_4} \Pi^{(4'5')}_{s_1} \Pi^{(25;2'5')}_{s_2} \Pi^{(56;3'6')}_{s_3} \mathbf{\Pi}^{(\mathrm{anc})}_{s_0}$ |
| $(X \otimes X)^{\frac{1-s_2 s_0 s'_0}{2}} (Y \otimes Y)^{\frac{1-s_3 s_1 s_0}{2}} \mathrm{SWAP}_l$ | $\Pi^{(3'4')}_{s'_4} \Pi^{(4'5')}_{s_3} \Pi^{(15;1'5')}_{s_2} \Pi^{(25;2'3')}_{s_1} \mathbf{\Pi}^{(\mathrm{anc})}_{s_0}$ |
| $(\mathbb{1} \otimes Z)^{\frac{1-s'_3 s'_0}{2}} (Z \otimes Z)^{\frac{1+s_2 s_1 s'_0}{2}} W_u$ | $\Pi^{(3'4')}_{s'_3} \Pi^{(3'6')}_{s_2} \Pi^{(12;3'5')}_{s_1} \mathbf{\Pi}^{(\mathrm{anc})}_{s_0}$ |
| $(Z \otimes \mathbb{1})^{\frac{1-s_3 s_0}{2}} (Z \otimes Z)^{\frac{1+s_2 s_1 s_0}{2}} W_d$ | $\Pi^{(34)}_{s_3} \Pi^{(36)}_{s_2} \Pi^{(35;1'2')}_{s_1} \mathbf{\Pi}^{(\mathrm{anc})}_{s_0}$ |
| $(Z \otimes Z)^{\frac{1-s_2 s_1 s_0 s'_0}{2}} W_r$ | $\Pi^{(34)}_{s_3} \Pi^{(45)}_{s_2} \Pi^{(46;5'6')}_{s_1} \mathbf{\Pi}^{(\mathrm{anc})}_{s_0}$ |
| $(Z \otimes Z)^{\frac{1-s_2 s_1 s_0 s'_0}{2}} W_l$ | $\Pi^{(34)}_{s_3} \Pi^{(45)}_{s_2} \Pi^{(46;5'6')}_{s_1} \mathbf{\Pi}^{(\mathrm{anc})}_{s_0}$ |

### C.3 One-Sided Hexon with Configuration ⟨3, 4, 1, 2, 6, 5⟩

The MZM labeling configuration ⟨3, 4, 1, 2, 6, 5⟩ is optimal for the one-sided hexon architecture, when using the forced-measurement methods for the gates: the controlled-$X$ gate C($X$), and the geometric average of the controlled-Pauli gates C($P$). This configuration within an array looks like:

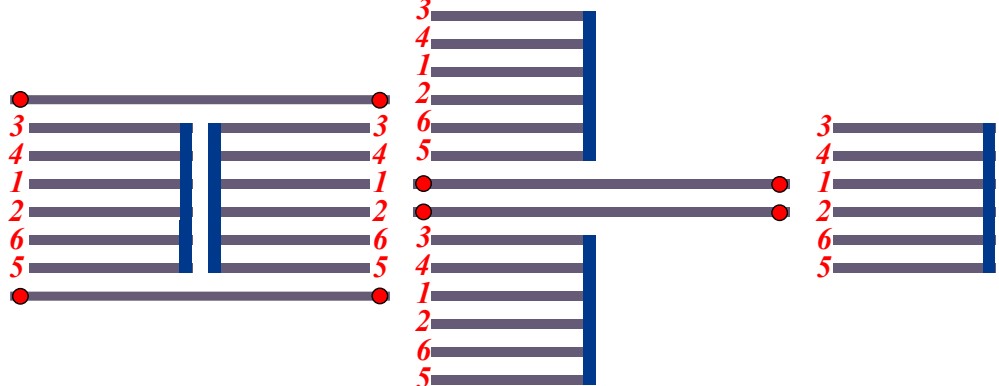

Table 21: Minimal difficulty weight measurement sequences for each single-qubit Clifford gate when using forced-measurement methods. For comparison, we also present the corresponding realization of the gates formed by using the generating gate sets $\langle S,H\rangle$ and $\langle S,B\rangle$.

| Gate | Forced Measurement Sequence | Weight | $\langle S,H\rangle$ Decomp. | Weight | $\langle S,B\rangle$ Decomp. | Weight |
|---|---|---|---|---|---|---|
| $X$ | $\overset{\epsilon p}{\Pi}{}_+^{(34)}\overset{\epsilon p}{\Pi}{}_{-s_1}^{(14)}\Pi_{s_2}^{(16)}\Pi_{s_1}^{(14)}$ | $3.10\times10^4$ | $HSSH$ | $5.12\times10^{19}$ | $BB$ | $1.30\times10^8$ |
| $Y$ | $\overset{\epsilon p}{\Pi}{}_+^{(34)}\overset{\epsilon p}{\Pi}{}_{-s_1}^{(24)}\Pi_{s_2}^{(26)}\Pi_{s_1}^{(24)}$ | $4.95\times10^4$ | $HSSHSS$ | $6.62\times10^{27}$ | $BBSS$ | $6.66\times10^{15}$ |
| $Z$ | $\overset{\epsilon p}{\Pi}{}_+^{(34)}\overset{\epsilon p}{\Pi}{}_{-s_1}^{(14)}\Pi_{s_2}^{(12)}\Pi_{s_1}^{(14)}$ | $1.95\times10^4$ | $SS$ | $5.13\times10^7$ | $SS$ | $5.13\times10^7$ |
| $S$ | $\overset{\epsilon p}{\Pi}{}_+^{(34)}\Pi_{s_1}^{(14)}\Pi_{s_1}^{(24)}$ | $7.16\times10^3$ | $S$ | $7.16\times10^3$ | $S$ | $7.16\times10^3$ |
| $XS$ | $\overset{\epsilon p}{\Pi}{}_+^{(34)}\Pi_{-s_1}^{(14)}\Pi_{s_2}^{(16)}\Pi_{s_1}^{(14)}\Pi_{s_1}^{(24)}$ | $1.59\times10^6$ | $HSSHS$ | $3.66\times10^{23}$ | $BBS$ | $9.31\times10^{11}$ |
| $YS$ | $\overset{\epsilon p}{\Pi}{}_+^{(34)}\Pi_{s_1}^{(14)}\Pi_{s_2}^{(16)}\Pi_{-s_1}^{(14)}\Pi_{s_1}^{(24)}$ | $1.59\times10^6$ | $SHSSH$ | $3.66\times10^{23}$ | $SBB$ | $9.31\times10^{11}$ |
| $ZS$ | $\overset{\epsilon p}{\Pi}{}_+^{(34)}\Pi_{s_1}^{(14)}\Pi_{s_1}^{(24)}$ | $7.16\times10^3$ | $S^\dagger$ | $7.16\times10^3$ | $S^\dagger$ | $7.16\times10^3$ |
| $H$ | $\overset{\epsilon p}{\Pi}{}_+^{(34)}\overset{\frown}{\Pi}{}_{-s_1}^{(14)}\overset{\epsilon p}{\Pi}{}_{s_2}^{(12)}\Pi_{s_2}^{(16)}\Pi_{s_1}^{(14)}$ | $9.99\times10^5$ | $H$ | $9.99\times10^5$ | $SBS$ | $5.84\times10^{11}$ |
| $XH$ | $\overset{\epsilon p}{\Pi}{}_+^{(34)}\overset{\frown}{\Pi}{}_{s_1}^{(14)}\Pi_{s_1}^{(45)}$ | $1.82\times10^4$ | $HSS$ | $5.12\times10^{13}$ | $S^\dagger B^\dagger S$ | $5.84\times10^{11}$ |
| $YH$ | $\overset{\epsilon p}{\Pi}{}_+^{(34)}\overset{\frown}{\Pi}{}_{-s_1}^{(14)}\overset{\epsilon p}{\Pi}{}_{s_2}^{(12)}\Pi_{s_2}^{(16)}\Pi_{s_1}^{(14)}$ | $9.99\times10^5$ | $SSHSS$ | $2.63\times10^{21}$ | $SB^\dagger S$ | $5.84\times10^{11}$ |
| $ZH$ | $\overset{\epsilon p}{\Pi}{}_+^{(34)}\Pi_{-s_1}^{(14)}\Pi_{s_1}^{(45)}$ | $1.82\times10^4$ | $SSH$ | $5.12\times10^{13}$ | $S^\dagger BS$ | $5.84\times10^{11}$ |
| $SH$ | $\overset{\epsilon p}{\Pi}{}_+^{(34)}\overset{\frown}{\Pi}{}_{s_1}^{(14)}\overset{\frown}{\Pi}{}_{s_1}^{(24)}\Pi_{s_1}^{(46)}$ | $5.85\times10^5$ | $SH$ | $7.15\times10^9$ | $B^\dagger S^\dagger$ | $8.16\times10^7$ |
| $XSH$ | $\overset{\epsilon p}{\Pi}{}_+^{(34)}\overset{\frown}{\Pi}{}_{-s_1}^{(14)}\overset{\frown}{\Pi}{}_{s_1}^{(24)}\Pi_{s_1}^{(46)}$ | $5.85\times10^5$ | $S^\dagger HSS$ | $3.67\times10^{17}$ | $BS^\dagger$ | $8.16\times10^7$ |
| $YSH$ | $\overset{\epsilon p}{\Pi}{}_+^{(34)}\overset{\frown}{\Pi}{}_{s_1}^{(14)}\overset{\frown}{\Pi}{}_{-s_1}^{(24)}\Pi_{s_1}^{(46)}$ | $5.85\times10^5$ | $SHSS$ | $3.67\times10^{17}$ | $B^\dagger S$ | $8.16\times10^7$ |
| $ZSH$ | $\overset{\epsilon p}{\Pi}{}_+^{(34)}\overset{\frown}{\Pi}{}_{-s_1}^{(14)}\overset{\frown}{\Pi}{}_{-s_1}^{(24)}\Pi_{s_1}^{(46)}$ | $5.85\times10^5$ | $S^\dagger H$ | $7.15\times10^9$ | $BS$ | $8.16\times10^7$ |
| $HS$ | $\overset{\epsilon p}{\Pi}{}_+^{(34)}\overset{\frown}{\Pi}{}_{-s_1}^{(14)}\overset{\frown}{\Pi}{}_{s_1}^{(24)}\Pi_{s_1}^{(45)}$ | $9.32\times10^5$ | $HS$ | $7.15\times10^9$ | $S^\dagger B^\dagger$ | $8.16\times10^7$ |
| $XHS$ | $\overset{\epsilon p}{\Pi}{}_+^{(34)}\overset{\frown}{\Pi}{}_{-s_1}^{(14)}\overset{\frown}{\Pi}{}_{-s_1}^{(24)}\Pi_{s_1}^{(45)}$ | $9.32\times10^5$ | $HS^\dagger$ | $7.15\times10^9$ | $SB$ | $8.16\times10^7$ |
| $YHS$ | $\overset{\epsilon p}{\Pi}{}_+^{(34)}\overset{\frown}{\Pi}{}_{s_1}^{(14)}\overset{\frown}{\Pi}{}_{-s_1}^{(24)}\Pi_{s_1}^{(45)}$ | $9.32\times10^5$ | $SSHS^\dagger$ | $3.67\times10^{17}$ | $S^\dagger B$ | $8.16\times10^7$ |
| $ZHS$ | $\overset{\epsilon p}{\Pi}{}_+^{(34)}\overset{\frown}{\Pi}{}_{s_1}^{(14)}\overset{\frown}{\Pi}{}_{s_1}^{(24)}\Pi_{s_1}^{(45)}$ | $9.32\times10^5$ | $SSHS$ | $3.67\times10^{17}$ | $SB^\dagger$ | $8.16\times10^7$ |
| $SHS$ | $\overset{\epsilon p}{\Pi}{}_+^{(34)}\overset{\frown}{\Pi}{}_{s_1}^{(14)}\Pi_{s_1}^{(46)}$ | $1.14\times10^4$ | $SHS$ | $5.12\times10^{13}$ | $B^\dagger$ | $1.14\times10^4$ |
| $XSHS$ | $\overset{\epsilon p}{\Pi}{}_+^{(34)}\overset{\frown}{\Pi}{}_{s_1}^{(14)}\Pi_{s_1}^{(46)}$ | $1.14\times10^4$ | $S^\dagger HS^\dagger$ | $5.12\times10^{13}$ | $B$ | $1.14\times10^4$ |
| $YSHS$ | $\overset{\epsilon p}{\Pi}{}_+^{(34)}\overset{\epsilon p}{\Pi}{}_{-s_2 s_1}^{(14)}\overset{\epsilon p}{\Pi}{}_{-s_2}^{(12)}\Pi_{s_2}^{(26)}\Pi_{s_1}^{(23)}$ | $1.26\times10^6$ | $SHS^\dagger$ | $5.12\times10^{13}$ | $B^\dagger SS$ | $5.84\times10^{11}$ |
| $ZSHS$ | $\overset{\epsilon p}{\Pi}{}_+^{(34)}\overset{\epsilon p}{\Pi}{}_{s_2 s_1}^{(14)}\overset{\epsilon p}{\Pi}{}_{s_2}^{(12)}\Pi_{s_2}^{(26)}\Pi_{s_1}^{(23)}$ | $1.26\times10^6$ | $S^\dagger HS$ | $5.12\times10^{13}$ | $BSS$ | $5.84\times10^{11}$ |
| Average | | $1.89\times10^5$ | | $2.02\times10^{14}$ | | $8.44\times10^8$ |

Table 22: Minimal difficulty weight measurement sequences for each Pauli coset of single-qubit Clifford gates when using Majorana-Pauli tracking methods. For comparison, we also present the corresponding realization of the gates formed by using the generating gate sets $\langle S,H\rangle$ and $\langle S,B\rangle$.

| Pauli Class | Unforced Measurement Sequence | Weight | $\langle S,H\rangle$ Decomp. | Weight | $\langle S,B\rangle$ Decomp. | Weight |
|---|---|---|---|---|---|---|
| $[S]$ | $\Pi_{s_3}^{(34)}\Pi_{s_2}^{(24)}\Pi_{s_1}^{(14)}$ | $5.13\times10^1$ | $S$ | $5.13\times10^1$ | $S$ | $5.13\times10^1$ |
| $[H]$ | $\Pi_{s_3}^{(34)}\Pi_{s_2}^{(45)}\Pi_{s_1}^{(14)}$ | $8.17\times10^1$ | $H$ | $8.17\times10^1$ | $SBS$ | $1.70\times10^5$ |
| $[SH]$ | $\Pi_{s_4}^{(34)}\Pi_{s_3}^{(46)}\Pi_{s_2}^{(14)}\Pi_{s_1}^{(24)}$ | $2.81\times10^2$ | $SH$ | $4.19\times10^3$ | $BS$ | $3.32\times10^3$ |
| $[HS]$ | $\Pi_{s_4}^{(34)}\Pi_{s_3}^{(46)}\Pi_{s_2}^{(24)}\Pi_{s_1}^{(14)}$ | $2.81\times10^2$ | $HS$ | $4.19\times10^3$ | $SB$ | $3.32\times10^3$ |
| $[SHS]$ | $\Pi_{s_3}^{(34)}\Pi_{s_2}^{(46)}\Pi_{s_1}^{(14)}$ | $6.47\times10^1$ | $SHS$ | $2.15\times10^5$ | $B$ | $6.47\times10^1$ |
| Average | | $1.16\times10^2$ | | $1.74\times10^3$ | | $1.44\times10^3$ |

Table 23: Pauli gate corrections tracked for the corresponding single-qubit gates implemented using Majorana-Pauli tracking methods. The implicit action on the ancillary qubit is $X^{\frac{1-s_n s_0}{2}}\Pi_{s_0}^{(34)}$ for sequences of length $n$.

| Gate | Tracked Measurement Sequence |
|---|---|
| $Y^{\frac{1-s_3 s_0}{2}} Z^{\frac{1-s_2 s_1}{2}} S$ | $\Pi_{s_3}^{(34)}\Pi_{s_2}^{(14)}\Pi_{s_1}^{(24)}\Pi_{s_0}^{(34)}$ |
| $Y^{\frac{1-s_3 s_0}{2}} Y^{\frac{1-s_2 s_1 s_0}{2}} ZH$ | $\Pi_{s_3}^{(34)}\Pi_{s_2}^{(14)}\Pi_{s_1}^{(45)}\Pi_{s_0}^{(34)}$ |
| $Y^{\frac{1-s_4 s_0}{2}} Z^{\frac{1-s_3 s_2}{2}} X^{\frac{1-s_2 s_1}{2}} YSH$ | $\Pi_{s_4}^{(34)}\Pi_{s_3}^{(14)}\Pi_{s_2}^{(24)}\Pi_{s_1}^{(46)}\Pi_{s_0}^{(34)}$ |
| $Z^{\frac{1-s_4 s_0}{2}} Y^{\frac{1-s_3 s_2}{2}} X^{\frac{1-s_2 s_1}{2}} YHS$ | $\Pi_{s_4}^{(34)}\Pi_{s_3}^{(46)}\Pi_{s_2}^{(24)}\Pi_{s_1}^{(14)}\Pi_{s_0}^{(34)}$ |
| $Y^{\frac{1-s_3 s_0}{2}} X^{\frac{1-s_2 s_1}{2}} SHS$ | $\Pi_{s_3}^{(34)}\Pi_{s_2}^{(14)}\Pi_{s_1}^{(46)}\Pi_{s_0}^{(34)}$ |

Table 24: Minimal difficulty weight measurement sequences for controlled-Pauli two-qubit gates when using forced-measurement methods. The labels $(u)$, $(d)$, $(r)$, and $(l)$ indicates that for a hexon acting as the control qubit of the $\mathsf{C}(P)$ gate, the corresponding target qubit is the nearest neighbor hexon in the up, down, right, and left direction, respectively. Notice that the choice of control and target qubit is arbitrary for $\mathsf{C}(Z)$, so $(u)$ and $(d)$ are related by symmetry, as are $(r)$, and $(l)$. The average difficulty weight of the four directions is labeled by $(a)$. For comparison, we also present the corresponding realization of the gates formed by using the generating gate sets $\langle S, H, \mathsf{C}(Z)\rangle$ and $\langle S, B, W\rangle$. $^A\mathsf{C}(Z) = A_2\mathsf{C}(Z)A_2^\dagger$ denotes conjugation of $\mathsf{C}(Z)$ by $A$ on the target qubit.

| Gate | Forced Measurement Sequence | Weight | $\langle S, H, \mathsf{C}(Z)\rangle$ | Weight | $\langle S, B, W\rangle$ | Weight |
|---|---|---|---|---|---|---|
| $\mathsf{C}(X)(u)$ | $\Pi_+^{(34)}\Pi_{-s_1}^{(14)}\Pi_+^{(12)}\Pi_{s_1}^{(23;2'5')}$ | $1.43\times10^7$ | $^H\mathsf{C}(Z)$ | $8.94\times10^{18}$ | $S_2 B_2 S_1 W^\dagger S_2^\dagger B_2 S_2$ | $1.51\times10^{29}$ |
| $(d)$ | $\Pi_+^{(3'4')}\Pi_{s_1}^{(1'3')}\Pi_+^{(1'6')}\Pi_{s_1}^{(56;1'3')}$ | $2.28\times10^7$ | | $8.94\times10^{18}$ | | $1.51\times10^{29}$ |
| $(r)$ | $\Pi_+^{(34)}\Pi_{-s_1}^{(14)}\Pi_+^{(12)}\Pi_{s_1}^{(23;1'6')}$ | $1.03\times10^9$ | | $6.46\times10^{20}$ | | $2.19\times10^{31}$ |
| $(l)$ | $\Pi_+^{(34)}\Pi_{-s_1}^{(14)}\Pi_+^{(12)}\Pi_{s_1}^{(23;1'6')}$ | $1.56\times10^{10}$ | | $2.49\times10^{22}$ | | $2.64\times10^{32}$ |
| $(a)$ | | $2.69\times10^8$ | | $1.89\times10^{20}$ | | $3.39\times10^{30}$ |
| $\mathsf{C}(Y)(u)$ | $\Pi_+^{(34)}\Pi_{-s_1}^{(14)}\Pi_+^{(12)}\Pi_{s_1}^{(23;1'5')}$ | $2.28\times10^7$ | $^{SH}\mathsf{C}(Z)$ | $4.59\times10^{26}$ | $B_2^\dagger W^\dagger S_2 B_2 S_1$ | $1.85\times10^{21}$ |
| $(d)$ | $\Pi_+^{(3'4')}\Pi_{s_1}^{(2'4')}\Pi_+^{(2'6')}\Pi_{s_1}^{(56;3'6')}$ | $2.88\times10^7$ | | $4.59\times10^{26}$ | | $1.85\times10^{21}$ |
| $(r)$ | $\Pi_+^{(3'4')}\Pi_{s_1}^{(2'4')}\Pi_+^{(2'6')}\Pi_{s_1}^{(12;2'4')}$ | $8.17\times10^8$ | | $3.31\times10^{28}$ | | $2.69\times10^{23}$ |
| $(l)$ | $\Pi_+^{(34)}\Pi_{-s_1}^{(14)}\Pi_+^{(12)}\Pi_{s_1}^{(23;1'5')}$ | $9.80\times10^9$ | | $1.27\times10^{30}$ | | $3.23\times10^{24}$ |
| $(a)$ | | $2.69\times10^8$ | | $9.70\times10^{27}$ | | $4.16\times10^{22}$ |
| $\mathsf{C}(Z)(u)$ | $\Pi_+^{(34)}\Pi_{-s_1}^{(14)}\Pi_+^{(12)}\Pi_{s_1}^{(23;5'6')}$ | $8.96\times10^6$ | $\mathsf{C}(Z)$ | $8.96\times10^6$ | $S_1 S_2 W^\dagger$ | $1.43\times10^{13}$ |
| $(d)$ | $\Pi_+^{(3'4')}\Pi_{-s_1}^{(1'4')}\Pi_+^{(1'2')}\Pi_{s_1}^{(56;2'3')}$ | $8.96\times10^6$ | | $8.96\times10^6$ | | $1.43\times10^{13}$ |
| $(r)$ | $\Pi_{s_1}^{(34)}\Pi_{-s_1}^{(13)}\Pi_+^{(12)}\Pi_{s_1}^{(24;1'2')}$ | $6.47\times10^8$ | | $6.47\times10^8$ | | $2.07\times10^{15}$ |
| $(l)$ | $\Pi_+^{(34)}\Pi_{-s_1}^{(14)}\Pi_+^{(12)}\Pi_{s_1}^{(23;1'2')}$ | $2.49\times10^{10}$ | | $2.49\times10^{10}$ | | $2.49\times10^{16}$ |
| $(a)$ | | $1.90\times10^8$ | | $1.90\times10^8$ | | $3.20\times10^{14}$ |
| Average$(u)$ | | $1.43\times10^7$ | | $3.33\times10^{17}$ | | $1.56\times10^{21}$ |
| $(d)$ | | $1.80\times10^7$ | | $3.33\times10^{17}$ | | $1.56\times10^{21}$ |
| $(r)$ | | $8.17\times10^8$ | | $2.40\times10^{19}$ | | $2.30\times10^{23}$ |
| $(l)$ | | $1.56\times10^{10}$ | | $9.23\times10^{20}$ | | $2.77\times10^{24}$ |
| $(a)$ | | $2.39\times10^8$ | | $7.04\times10^{18}$ | | $3.56\times10^{22}$ |

Table 25: Minimal difficulty weight measurement sequences for Pauli cosets of controlled-Pauli two-qubit gates when using Majorana-Pauli tracking methods. The labels $(u)$, $(d)$, $(r)$, and $(l)$ indicates that for a hexon acting as the control qubit of the $C(P)$ gate, the corresponding target qubit is the nearest neighbor hexon in the up, down, right, and left direction, respectively. Notice that the choice of control and target qubit is arbitrary for $C(Z)$, so $(u)$ and $(d)$ are related by symmetry, as are $(r)$, and $(l)$. The average difficulty weight of the four directions is labeled by $(a)$. For comparison, we also present the corresponding realization of the gates formed by using the generating gate sets $\langle S, H, C(Z) \rangle$ and $\langle S, B, W \rangle$. $^A C(Z) = A_2 C(Z) A_2^\dagger$ denotes conjugation of $C(Z)$ by $A$ on the target qubit.

| Pauli Class | Tracked Measurement Sequence | Weight | $\langle S,H,C(Z)\rangle$ | Weight | $\langle S,B,C(Z)\rangle$ | Weight |
|---|---|---|---|---|---|---|
| $[C(X)](u)$ | $\Pi_{s_4}^{(34)}\Pi_{s_3}^{(24)}\Pi_{s_2}^{(12)}\Pi_{s_1}^{(13;2'5')}$ | $1.24\times10^3$ | $^H C(Z)$ | $6.53\times10^6$ | $S_2 B_2 S_1 W^\dagger S_2^\dagger B_2 S_2$ | $1.04\times10^{13}$ |
| $(d)$ | $\Pi_{s_4'}^{(3'4')}\Pi_{s_3}^{(1'3')}\Pi_{s_2}^{(1'6')}\Pi_{s_1}^{(56;1'3')}$ | $1.24\times10^3$ | | $6.53\times10^6$ | | $1.04\times10^{13}$ |
| $(r)$ | $\Pi_{s_4}^{(34)}\Pi_{s_3}^{(14)}\Pi_{s_2}^{(12)}\Pi_{s_1}^{(23;1'6')}$ | $9.34\times10^3$ | | $4.94\times10^7$ | | $4.96\times10^{13}$ |
| $(l)$ | $\Pi_{s_4}^{(34)}\Pi_{s_3}^{(14)}\Pi_{s_2}^{(12)}\Pi_{s_1}^{(23;1'6')}$ | $3.64\times10^4$ | | $3.06\times10^8$ | | $4.90\times10^{14}$ |
| $(a)$ | | $4.78\times10^3$ | | $2.83\times10^7$ | | $4.03\times10^{13}$ |
| $[C(Y)](u)$ | $\Pi_{s_4}^{(34)}\Pi_{s_3}^{(24)}\Pi_{s_2}^{(34;1'5')}\Pi_{s_1}^{(13)}$ | $1.56\times10^3$ | $^{SH} C(Z)$ | $1.72\times10^{10}$ | $B_2^\dagger W^\dagger S_2 B_2 S_1$ | $3.97\times10^9$ |
| $(d)$ | $\Pi_{s_4'}^{(3'4')}\Pi_{s_3}^{(3'5')}\Pi_{s_2}^{(56;3'4')}\Pi_{s_1}^{(1'4')}$ | $1.56\times10^3$ | | $1.72\times10^{10}$ | | $3.97\times10^9$ |
| $(r)$ | $\Pi_{s_4'}^{(3'4')}\Pi_{s_3}^{(2'4')}\Pi_{s_2}^{(2'6')}\Pi_{s_1}^{(12;2'4')}$ | $7.40\times10^3$ | | $1.30\times10^{11}$ | | $1.88\times10^{10}$ |
| $(l)$ | $\Pi_{s_4}^{(34)}\Pi_{s_3}^{(14)}\Pi_{s_2}^{(12)}\Pi_{s_1}^{(23;1'5')}$ | $2.88\times10^4$ | | $8.06\times10^{11}$ | | $1.86\times10^{11}$ |
| $(a)$ | | $4.77\times10^3$ | | $7.46\times10^{10}$ | | $1.53\times10^{10}$ |
| $[C(Z)](u)$ | $\Pi_{s_4}^{(34)}\Pi_{s_3}^{(14)}\Pi_{s_2}^{(12)}\Pi_{s_1}^{(23;5'6')}$ | $9.79\times10^2$ | $C(Z)$ | $9.79\times10^2$ | $S_1 S_2 W^\dagger$ | $9.47\times10^5$ |
| $(d)$ | $\Pi_{s_4'}^{(3'4')}\Pi_{s_3}^{(1'4')}\Pi_{s_2}^{(1'2')}\Pi_{s_1}^{(56;2'3')}$ | $9.79\times10^2$ | | $9.79\times10^2$ | | $9.47\times10^5$ |
| $(r)$ | $\Pi_{s_4}^{(34)}\Pi_{s_3}^{(23)}\Pi_{s_2}^{(12;1'2')}\Pi_{s_1}^{(14)}$ | $7.40\times10^3$ | | $7.40\times10^3$ | | $4.50\times10^6$ |
| $(l)$ | $\Pi_{s_4}^{(34)}\Pi_{s_3}^{(14)}\Pi_{s_2}^{(12)}\Pi_{s_1}^{(23;1'2')}$ | $4.59\times10^4$ | | $4.59\times10^4$ | | $4.45\times10^7$ |
| $(a)$ | | $4.25\times10^3$ | | $4.25\times10^3$ | | $3.66\times10^6$ |
| Average$(u)$ | | $1.24\times10^3$ | | $4.79\times10^6$ | | $3.40\times10^9$ |
| $(d)$ | | $1.24\times10^3$ | | $4.79\times10^6$ | | $3.40\times10^9$ |
| $(r)$ | | $8.00\times10^3$ | | $3.62\times10^7$ | | $1.61\times10^{10}$ |
| $(l)$ | | $3.64\times10^4$ | | $2.25\times10^8$ | | $1.59\times10^{11}$ |
| $(a)$ | | $4.59\times10^3$ | | $2.07\times10^7$ | | $1.31\times10^{10}$ |

Table 26: Pauli gate corrections tracked for the corresponding controlled-Pauli gates implemented using Majorana-Pauli tracking methods. The implicit action on the two ancillary qubits is $X^{\frac{1-s_n s_0}{2}}\Pi^{(34)}_{s_0}\otimes X^{\frac{1-s'_n s'_0}{2}}\Pi^{(3'4')}_{s'_0}$ for sequences of length $n$. (When there is not a final projector for one of the ancillary pairs, it is equivalent to there being a projector for that ancillary pair onto its initial projection channel, e.g. $s_n = s_0$ or $s'_n = s'_0$.)

| Gate | Tracked Measurement Sequence |
|---|---|
| $\left( X^{\frac{1-s_4 s_0}{2}} Z^{\frac{1-s_3 s_2 s_1 s_0 s'_0}{2}} \otimes X^{\frac{1-s_2}{2}} \right)\mathsf{C}(X)_u$ | $\Pi^{(34)}_{s_4}\Pi^{(24)}_{s_3}\Pi^{(12)}_{s_2}\Pi^{(13;2'5')}_{s_1}\mathbf{\Pi}^{(\mathrm{anc})}_{s_0}$ |
| $\left( Z^{\frac{1-s_2}{2}} \otimes Y^{\frac{1-s'_4 s'_0}{2}} X^{\frac{1-s_3 s_1 s_0}{2}} \right)\mathsf{C}(X)_d$ | $\Pi^{(3'4')}_{s'_4}\Pi^{(1'3')}_{s_3}\Pi^{(1'6')}_{s_2}\Pi^{(56;1'3')}_{s_1}\mathbf{\Pi}^{(\mathrm{anc})}_{s_0}$ |
| $\left( Y^{\frac{1-s_4 s_0}{2}} Z^{\frac{1+s_3 s_1 s_2 s_0}{2}} \otimes X^{\frac{1-s_2}{2}} \right)\mathsf{C}(X)_r$ | $\Pi^{(34)}_{s_4}\Pi^{(14)}_{s_3}\Pi^{(12)}_{s_2}\Pi^{(23;1'6')}_{s_1}\mathbf{\Pi}^{(\mathrm{anc})}_{s_0}$ |
| $\left( Y^{\frac{1-s_4 s_0}{2}} Z^{\frac{1+s_3 s_2 s_1 s_0}{2}} \otimes X^{\frac{1-s_2}{2}} \right)\mathsf{C}(X)_l$ | $\Pi^{(34)}_{s_4}\Pi^{(14)}_{s_3}\Pi^{(12)}_{s_2}\Pi^{(23;1'6')}_{s_1}\mathbf{\Pi}^{(\mathrm{anc})}_{s_0}$ |
| $\left( X^{\frac{1-s_4 s_0}{2}} Z^{\frac{1-s_2 s_0 s'_0}{2}} \otimes Y^{\frac{1+s_3 s_1 s_0}{2}} \right)\mathsf{C}(Y)_u$ | $\Pi^{(34)}_{s_4}\Pi^{(24)}_{s_3}\Pi^{(34;1'5')}_{s_2}\Pi^{(13)}_{s_1}\mathbf{\Pi}^{(\mathrm{anc})}_{s_0}$ |
| $\left( Z^{\frac{1+s_3 s_1}{2}} \otimes Y^{\frac{1-s_2 s_0 s'_0}{2}} \right)\mathsf{C}(Y)_d$ | $\Pi^{(3'4')}_{s'_4}\Pi^{(3'5')}_{s_3}\Pi^{(56;3'4')}_{s_2}\Pi^{(1'4')}_{s_1}\mathbf{\Pi}^{(\mathrm{anc})}_{s_0}$ |
| $\left( Z^{\frac{1+s_2}{2}} \otimes X^{\frac{1-s'_4 s'_0}{2}} Y^{\frac{1-s_3 s_1}{2}} \right)\mathsf{C}(Y)_r$ | $\Pi^{(3'4')}_{s'_4}\Pi^{(2'4')}_{s_3}\Pi^{(2'6')}_{s_2}\Pi^{(12;2'4')}_{s_1}\mathbf{\Pi}^{(\mathrm{anc})}_{s_0}$ |
| $\left( Y^{\frac{1-s_4 s_0}{2}} Z^{\frac{1+s_3 s_2 s_1 s_0 s'_0}{2}} \otimes Y^{\frac{1-s_2}{2}} \right)\mathsf{C}(Y)_l$ | $\Pi^{(34)}_{s_4}\Pi^{(14)}_{s_3}\Pi^{(12)}_{s_2}\Pi^{(23;1'5')}_{s_1}\mathbf{\Pi}^{(\mathrm{anc})}_{s_0}$ |
| $\left( Y^{\frac{1-s_4 s_0}{2}} Z^{\frac{1+s_3 s_2 s_1 s_0 s'_0}{2}} \otimes Z \right)^{\frac{1-s_2}{2}}\mathsf{C}(Z)_u$ | $\Pi^{(34)}_{s_4}\Pi^{(14)}_{s_3}\Pi^{(12)}_{s_2}\Pi^{(23;5'6')}_{s_1}\mathbf{\Pi}^{(\mathrm{anc})}_{s_0}$ |
| $\left( Z^{\frac{1-s_2}{2}} \otimes Y^{\frac{1-s'_4 s'_0}{2}} Z^{\frac{1+s_3 s_2 s_1 s_0 s'_0}{2}} \right)\mathsf{C}(Z)_d$ | $\Pi^{(3'4')}_{s'_4}\Pi^{(1'4')}_{s_3}\Pi^{(1'2')}_{s_2}\Pi^{(56;2'3')}_{s_1}\mathbf{\Pi}^{(\mathrm{anc})}_{s_0}$ |
| $\left( X^{\frac{1-s_4 s_0}{2}} Z^{\frac{1+s_3 s_2 s_1 s_0}{2}} \otimes Z^{\frac{1-s_3 s_1 s_0}{2}} \right)\mathsf{C}(Z)_r$ | $\Pi^{(34)}_{s_4}\Pi^{(23)}_{s_3}\Pi^{(12;1'2')}_{s_2}\Pi^{(14)}_{s_1}\mathbf{\Pi}^{(\mathrm{anc})}_{s_0}$ |
| $\left( Y^{\frac{1-s_4 s_0}{2}} Z^{\frac{1+s_3 s_2 s_1 s_0}{2}} \otimes Z^{\frac{1-s_2}{2}} \right)\mathsf{C}(Z)_l$ | $\Pi^{(34)}_{s_4}\Pi^{(14)}_{s_3}\Pi^{(12)}_{s_2}\Pi^{(23;1'2')}_{s_1}\mathbf{\Pi}^{(\mathrm{anc})}_{s_0}$ |

Table 27: Minimal difficulty weight measurement sequences for the two-qubit SWAP and $W$ gates when using forced-measurement methods and the Pauli cosets of SWAP and $W$ when using Majorana-Pauli tracking methods. For comparison, we also present the corresponding realization of the gates formed by using SWAP $= \mathsf{C}(X)_{12}\mathsf{C}(X)_{21}\mathsf{C}(X)_{12}$ with $\mathsf{C}(X)_{12}$ as given in Tables 24 and 25.

| Gate | Forced Measurement Sequence | Weight | $\langle S,H,\mathsf{C}(Z)\rangle$ | Weight | $\langle S,B,W\rangle$ | Weight |
|---|---|---|---|---|---|---|
| SWAP$(u)$ | $\begin{bmatrix}\overset{\text{ef}}{\Pi}{}^{(34)}_{+}\overset{\text{ef}}{\Pi}{}^{(23;2'5')}_{s_1}\overset{\frown}{\Pi}{}^{(26)}_{-s_2s_1}\\ \overset{\text{ef}}{\Pi}{}^{(35;5'6')}_{-s_1}\Pi^{(13)}_{s_2}\Pi^{(23;2'5')}_{s_1}\end{bmatrix}$ | $1.73\times10^{14}$ | $\mathsf{C}(Z)$ | $7.15\times10^{56}$ | $\mathsf{C}(Z)$ | $3.44\times10^{87}$ |
| $(d)$ | $\begin{bmatrix}\overset{\text{ef}}{\Pi}{}^{(3'4')}_{+}\overset{\text{ef}}{\Pi}{}^{(25;2'3')}_{s_1}\overset{\frown}{\Pi}{}^{(2'6')}_{-s_2s_1}\\ \overset{\text{ef}}{\Pi}{}^{(56;3'6')}_{-s_1}\Pi^{(1'3')}_{s_2}\Pi^{(25;2'3')}_{s_1}\end{bmatrix}$ | $1.73\times10^{14}$ | | $7.15\times10^{56}$ | | $3.44\times10^{87}$ |
| $(r)$ | $\overset{\frown}{\Pi}{}^{(34)}_{+}\overset{\text{ef}}{\Pi}{}^{(13)}_{s_1}\overset{\text{ef}}{\Pi}{}^{(12;1'2')}_{+}\Pi^{(46;1'6')}_{s_1}$ | $6.05\times10^{13}$ | | $2.70\times10^{62}$ | | $1.05\times10^{94}$ |
| $(l)$ | $\overset{\text{ef}}{\Pi}{}^{(34)}_{+}\overset{\frown}{\Pi}{}^{(14)}_{-s_1}\overset{\text{ef}}{\Pi}{}^{(15;1'5')}_{+}\Pi^{(36;1'6')}_{s_1}$ | $6.82\times10^{16}$ | | $1.54\times10^{67}$ | | $1.84\times10^{97}$ |
| $(a)$ | | $5.93\times10^{14}$ | | $6.79\times10^{60}$ | | $3.89\times10^{91}$ |
| $W(u)$ | $\overset{\frown}{\Pi}{}^{(34)}_{+}\overset{\text{ef}}{\Pi}{}^{(13)}_{s_1}\Pi^{(23;5'6')}_{s_1}$ | $4.44\times10^{5}$ | $S_1S_2W$ | $4.59\times10^{14}$ | $W$ | $4.44\times10^{5}$ |
| $(d)$ | $\overset{\frown}{\Pi}{}^{(3'4')}_{+}\overset{\text{ef}}{\Pi}{}^{(1'3')}_{s_1}\Pi^{(56;2'3')}_{s_1}$ | $4.44\times10^{5}$ | | $4.59\times10^{14}$ | | $4.44\times10^{5}$ |
| $(r)$ | $\overset{\text{ef}}{\Pi}{}^{(34)}_{+}\overset{\frown}{\Pi}{}^{(14)}_{s_1}\Pi^{(24;1'2')}_{s_1}$ | $7.92\times10^{6}$ | | $3.32\times10^{16}$ | | $7.92\times10^{6}$ |
| $(l)$ | $\overset{\frown}{\Pi}{}^{(34)}_{+}\overset{\text{ef}}{\Pi}{}^{(13)}_{s_1}\Pi^{(23;1'2')}_{s_1}$ | $9.77\times10^{8}$ | | $1.28\times10^{18}$ | | $9.77\times10^{8}$ |
| $(a)$ | | $6.25\times10^{6}$ | | $9.72\times10^{15}$ | | $6.25\times10^{6}$ |

| Pauli Class | Tracked Measurement Sequence | Weight | $\langle S,H,\mathsf{C}(Z)\rangle$ | Weight | $\langle S,B,W\rangle$ | Weight |
|---|---|---|---|---|---|---|
| [SWAP]$(u)$ | $\begin{bmatrix}\Pi^{(34)}_{s_6}\Pi^{(23;2'5')}_{s_5}\Pi^{(26)}_{s_4}\\ \Pi^{(35;5'6')}_{s_3}\Pi^{(13)}_{s_2}\Pi^{(23;2'5')}_{s_1}\end{bmatrix}$ | $1.44\times10^{6}$ | $\mathsf{C}(X)^3$ | $2.78\times10^{20}$ | $\mathsf{C}(X)^3$ | $1.12\times10^{39}$ |
| $(d)$ | $\begin{bmatrix}\Pi^{(3'4')}_{s_6}\Pi^{(25;2'3')}_{s_5}\Pi^{(2'6')}_{s_4}\\ \Pi^{(56;3'6')}_{s_3}\Pi^{(1'3')}_{s_2}\Pi^{(25;2'3')}_{s_1}\end{bmatrix}$ | $1.44\times10^{6}$ | | $2.78\times10^{20}$ | | $1.12\times10^{39}$ |
| $(r)$ | $\Pi^{(34)}_{s_4}\Pi^{(13)}_{s_3}\Pi^{(12;1'2')}_{s_2}\Pi^{(46;1'6')}_{s_1}$ | $3.92\times10^{5}$ | | $1.21\times10^{23}$ | | $1.22\times10^{41}$ |
| $(l)$ | $\Pi^{(3'4')}_{s'_4}\Pi^{(1'4')}_{s_3}\Pi^{(16;1'6')}_{s_2}\Pi^{(15;3'5')}_{s_1}$ | $5.94\times10^{6}$ | | $2.87\times10^{25}$ | | $1.18\times10^{44}$ |
| $(a)$ | | $1.48\times10^{6}$ | | $2.27\times10^{22}$ | | $6.53\times10^{40}$ |
| [$W$]$(u)$ | $\Pi^{(34)}_{+}\Pi^{(13)}_{s_1}\Pi^{(23;5'6')}_{s_1}$ | $3.60\times10^{2}$ | $S_1S_2\mathsf{C}(Z)$ | $2.58\times10^{6}$ | $W$ | $3.60\times10^{2}$ |
| $(d)$ | $\Pi^{(3'4')}_{+}\Pi^{(1'3')}_{s_1}\Pi^{(56;2'3')}_{s_1}$ | $3.60\times10^{2}$ | | $2.58\times10^{6}$ | | $3.60\times10^{2}$ |
| $(r)$ | $\Pi^{(34)}_{+}\Pi^{(14)}_{s_1}\Pi^{(24;1'2')}_{s_1}$ | $1.71\times10^{3}$ | | $1.95\times10^{7}$ | | $1.71\times10^{3}$ |
| $(l)$ | $\Pi^{(34)}_{+}\Pi^{(13)}_{s_1}\Pi^{(23;1'2')}_{s_1}$ | $1.69\times10^{4}$ | | $1.21\times10^{8}$ | | $1.69\times10^{4}$ |
| $(a)$ | | $1.39\times10^{3}$ | | $1.12\times10^{7}$ | | $1.39\times10^{3}$ |

Table 28: Pauli gate corrections tracked for the corresponding SWAP and $W$ gates implemented using Majorana-Pauli tracking methods. The implicit action on the two ancillary qubits is $X^{\frac{1-s_n s_0}{2}}\Pi^{(34)}_{s_0}\otimes X^{\frac{1-s'_n s'_0}{2}}\Pi^{(3'4')}_{s'_0}$ for sequences of length $n$. (When there is not a final projector for one of the ancillary pairs, it is equivalent to there being a projector for that ancillary pair onto its initial projection channel, e.g. $s_n = s_0$ or $s'_n = s'_0$.)

| Gate | Tracked Measurement Sequence |
|---:|:---:|
| $\left(X^{\frac{1-s_6 s_3 s_1 s_0}{2}}Y^{\frac{1-s_5 s_4 s_2 s'_0}{2}}Z\otimes X^{\frac{1-s_6 s_3 s_0}{2}}Y^{\frac{1-s_4 s_2 s'_0}{2}}Z^{\frac{1+s_1}{2}}\right)\mathsf{SWAP}_u$ | $\begin{bmatrix}\Pi^{(34)}_{s_6}\Pi^{(23;2'5')}_{s_5}\Pi^{(26)}_{s_4}\Pi^{(35;5'6')}_{s_3}\\ \Pi^{(13)}_{s_2}\Pi^{(23;2'5')}_{s_1}\boldsymbol{\Pi}^{(\mathrm{anc})}_{\boldsymbol{s}_0}\end{bmatrix}$ |
| $\left(X^{\frac{1-s'_6 s_3 s'_0}{2}}Y^{\frac{1-s_4 s_2 s_0}{2}}Z^{\frac{1+s_1}{2}}\otimes X^{\frac{1-s'_6 s_3 s_1 s'_0}{2}}Y^{\frac{1-s_5 s_4 s_2 s_0}{2}}Z\right)\mathsf{SWAP}_d$ | $\begin{bmatrix}\Pi^{(3'4')}_{s'_6}\Pi^{(25;2'3')}_{s_5}\Pi^{(2'6')}_{s_4}\Pi^{(56;3'6')}_{s_3}\\ \Pi^{(1'3')}_{s_2}\Pi^{(25;2'3')}_{s_1}\boldsymbol{\Pi}^{(\mathrm{anc})}_{\boldsymbol{s}_0}\end{bmatrix}$ |
| $(Y\otimes\mathbb{1})^{\frac{1-s_4 s_0}{2}}(X\otimes X)^{\frac{1-s_2}{2}}(Z\otimes Z)^{\frac{1-s_3 s_1 s_0}{2}}\mathsf{SWAP}_r$ | $\Pi^{(34)}_{s_4}\Pi^{(13)}_{s_3}\Pi^{(12;1'2')}_{s_2}\Pi^{(46;1'6')}_{s_1}\boldsymbol{\Pi}^{(\mathrm{anc})}_{\boldsymbol{s}_0}$ |
| $(\mathbb{1}\otimes Y)^{\frac{1-s'_4 s'_0}{2}}(Y\otimes Y)^{\frac{1-s_2}{2}}(X\otimes X)^{\frac{1+s_3 s_1 s_0}{2}}\mathsf{SWAP}_l$ | $\Pi^{(3'4')}_{s'_4}\Pi^{(1'4')}_{s_3}\Pi^{(16;1'6')}_{s_2}\Pi^{(15;3'5')}_{s_1}\boldsymbol{\Pi}^{(\mathrm{anc})}_{\boldsymbol{s}_0}$ |
| $(Y\otimes\mathbb{1})^{\frac{1-s_3 s_0}{2}}(Z\otimes Z)^{\frac{1-s_2 s_1 s'_0}{2}}W_u$ | $\Pi^{(34)}_{s_3}\Pi^{(13)}_{s_2}\Pi^{(23;5'6')}_{s_1}\boldsymbol{\Pi}^{(\mathrm{anc})}_{\boldsymbol{s}_0}$ |
| $(\mathbb{1}\otimes Y)^{\frac{1-s'_3 s'_0}{2}}(Z\otimes Z)^{\frac{1-s_2 s_1 s_0}{2}}W_d$ | $\Pi^{(3'4')}_{s'_3}\Pi^{(1'3')}_{s_2}\Pi^{(56;2'3')}_{s_1}\boldsymbol{\Pi}^{(\mathrm{anc})}_{\boldsymbol{s}_0}$ |
| $(Y\otimes\mathbb{1})^{\frac{1-s_3 s_0}{2}}(Z\otimes Z)^{\frac{1-s_2 s_1}{2}}W_r$ | $\Pi^{(34)}_{s_3}\Pi^{(14)}_{s_2}\Pi^{(24;1'2')}_{s_1}\boldsymbol{\Pi}^{(\mathrm{anc})}_{\boldsymbol{s}_0}$ |
| $(Y\otimes\mathbb{1})^{\frac{1-s_3 s_0}{2}}(Z\otimes Z)^{\frac{1-s_2 s_1}{2}}W_l$ | $\Pi^{(34)}_{s_3}\Pi^{(13)}_{s_2}\Pi^{(23;1'2')}_{s_1}\boldsymbol{\Pi}^{(\mathrm{anc})}_{\boldsymbol{s}_0}$ |

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
