# Peer review of "Optimizing Clifford gate generation for measurement-only topological quantum computation with Majorana zero modes"

_SciPost Physics, doi:SciPost Phys. 8, 091 (2020)_

## Round 3 · Referee Report · Anonymous · 2019-11-25

Strengths

1. The topic can become important if scalable topological qubits become a reality. The measurement-based braiding scheme seems like a good candidate for such systems.
2. The paper is well written and gives a very comprehensive treatment of the topic.
3. Advanced theoretical methodology is developed which may find uses beyond the work presented here.

Weaknesses

1. The usefulness for near-future experiment seems rather limited, as the results will only become important when (if) it becomes possible to scale topological qubits (if the authors disagree, I do not think this is clear from the present version of the paper). But, on the other hand, short-term impact on experiment should not be a requirement for good theoretical work.
2. Related to point 1., the paper does not discuss new device concepts (or improvements of existing concepts) which might have better performance, nor does it compare (even on a rather rough level) the measurement-based scheme to topological quantum computation based on more "traditional" braiding operations.
3. I mentioned the comprehensiveness of the paper as a strength, but I think it can also be a weakness. The paper is very long and lots of space is taken up for things which I do not find crucial for (what I perceive to be) the main message. For example, much space is spent on explaining and discussing the details of the forced measurement scheme, even though it is inferior to the tracked scheme (the authors even state in the end that this inferiority is obvious). In fact, one has to read rather far in the paper to discover that there is a better option than the forced measurement scheme.
4. As a result of the points above, I think this paper (as it is written now) will be of interest to a rather narrow group of researchers (although it's a high-quality work and likely to be of very high interest for that particular group).

Report

This paper focuses on one particular path towards topological quantum computation, namely based on using measurements to perform the topologically protected operations (Clifford gates in the case of Majorana zero modes (MZMs)). Both the general idea of measurement-based topological quantum computation and the specific proposals for experimental devices (based on MZMs) included in the paper have been presented before. The new work in the paper is instead the optimization of sequences of measurement-operations needed to perform certain computational operations. The paper is very comprehensive, with a substantial introduction to the (previously proposed) device layouts and measurement-based methods. Then the optimization methods and the results of applying those are described, with all the details given in Appendices.

I save my impression of the paper for the other sections (Strengths, Weaknesses and Changes), but have a few physics questions and comments here:

1. I think the advantage of the Hexon (and Hexon-like) architectures in terms of protection from quasiparticle poisoning is somewhat exaggerated. The charging energy only offers protection from quasiparticles tunneling in from outside the Hexon, but not from quasiparticles which are excited by splitting Cooper pairs within the Hexon itself. Because of the large number of Cooper pairs, a population of such quasiparticles are expected to persist even for temperatures far below the superconducting gap.

2. What are the requirements for measuring the collective parities of 4 (or more) MZMs (needed to go beyond single-qubit gates) without learning anything about the individual parities? Perhaps related to this, I do not understand the reason behind the statement (on page 11) that two-qubit operations require the ancillary pairs (of MZMs) to begin and end in the same state.

3. Flux noise is mentioned as a problem during measurement (which gets worse for increasing areas enclosed by the loop). I wonder if this is really a problem? I thought flux could be kept rather constant (important in most conventional superconducting qubits). And doesn't the energy of the QD shift in different ways (up or down) depending on the parity (which should then be easy to see even if there is some noise)?

Requested changes

I think the weaknesses described above warrant some changes. As a minimum, I think the authors should say at an early point that the forced measurement scheme is not needed and, in fact, an inferior method. However, I think the paper (and in particular it's usefulness for a somewhat broader readership) would benefit greatly from writing it in a way that makes the crucial parts accessible without reading all the details which are less important.

---

## Round 3 · Referee Report · Anonymous · 2020-1-8

Report

The authors study optimal, measurement-based implementations of Clifford gates in a quantum computer based on Majorana zero modes. Specifically, the authors investigate alternatives to the well-known “forced-measurement” scheme for realizing Clifford gates in a Majorana-based quantum computer based on the "hexon" qubit architecture. These alternatives generally require fewer measurements in order to implement the same gates. As an example, the authors discuss a “Majorana-Pauli tracking” scheme as an alternative to the forced-measurement protocol. Here, measurement outcomes are tracked (not forced), so that the implemented logical gate will differ from the desired Clifford gate by a Pauli operator, which may be easily determined from the tracked measurement outcomes. These Pauli operators may be taken into account when implementing Clifford gates at later steps in a given computation.

The newest part of this work is presented in Sec. V, where the authors optimize the measurements needed to implement a logical Clifford gate, with respect to a weighting function which measures the difficulty of implementing a series of measurements. For a sample set of trial weights, the authors are able to completely optimize the measurement scheme for single-qubit logical gates by brute force, and perform a partial optimization for two-qubit gates. The latter presents difficulties since the cost of a brute-force optimization scales exponentially in the number of total measurements that are permitted in a gate implementation.

The results of this work are sound, and I recommend the paper for publication after minor revisions.

Requested changes

(1) I believe there is a typo in the middle of the fourth paragraph of the introduction. The second half of the sentence

“We will describe different strategies and protocols for optimizing the generation of computational gates via measurement sequences, using measurements as the physical measurements as the generating set of operations”

reads incorrectly and should be fixed.

(2) A plot that shows the optimized difficulty weights for each logical gate would provide a simpler summary of the information presented in Tables I & II. This would more clearly show the relative difficulty of implementing various gates in the forced-measurement and Majorana-Pauli tracking schemes, and in the different hexon architectures.

---

## Round 4 · Referee Report · Anonymous · 2020-4-4

Report

I think the authors have provided satisfying answers to the reports of myself and the other referee. In my opinion the paper can now be accepted.

---

## Round 4 · Referee Report · Anonymous · 2020-5-20

Report

The authors have addressed my concerns in their response and in their resubmitted manuscript, which I now recommend for publication.

---

## Round 4 · Author Response

Dear Editors,

We thank the referees for their useful feedback. We have made revisions to our paper accordingly, along with numerous minor improvements. More significantly, we have added a section introducing a proposed Majorana hexon and tetron layout for implementation of a surface code and applied the methods of our paper to find optimized stabilizer measurement operations.

We address the referees' specific comments below.

Best regards,

Alan Tran, Alex Bocharov, Bela Bauer, Parsa Bonderson

REPORT 1:

Strengths 1. The topic can become important if scalable topological qubits become a reality. The measurement-based braiding scheme seems like a good candidate for such systems. 2. The paper is well written and gives a very comprehensive treatment of the topic. 3. Advanced theoretical methodology is developed which may find uses beyond the work presented here.

Weaknesses 1. The usefulness for near-future experiment seems rather limited, as the results will only become important when (if) it becomes possible to scale topological qubits (if the authors disagree, I do not think this is clear from the present version of the paper). But, on the other hand, short-term impact on experiment should not be a requirement for good theoretical work. 2. Related to point 1., the paper does not discuss new device concepts (or improvements of existing concepts) which might have better performance, nor does it compare (even on a rather rough level) the measurement-based scheme to topological quantum computation based on more "traditional" braiding operations. 3. I mentioned the comprehensiveness of the paper as a strength, but I think it can also be a weakness. The paper is very long and lots of space is taken up for things which I do not find crucial for (what I perceive to be) the main message. For example, much space is spent on explaining and discussing the details of the forced measurement scheme, even though it is inferior to the tracked scheme (the authors even state in the end that this inferiority is obvious). In fact, one has to read rather far in the paper to discover that there is a better option than the forced measurement scheme. 4. As a result of the points above, I think this paper (as it is written now) will be of interest to a rather narrow group of researchers (although it's a high-quality work and likely to be of very high interest for that particular group).

Report This paper focuses on one particular path towards topological quantum computation, namely based on using measurements to perform the topologically protected operations (Clifford gates in the case of Majorana zero modes (MZMs)). Both the general idea of measurement-based topological quantum computation and the specific proposals for experimental devices (based on MZMs) included in the paper have been presented before. The new work in the paper is instead the optimization of sequences of measurement-operations needed to perform certain computational operations. The paper is very comprehensive, with a substantial introduction to the (previously proposed) device layouts and measurement-based methods. Then the optimization methods and the results of applying those are described, with all the details given in Appendices.

I save my impression of the paper for the other sections (Strengths, Weaknesses and Changes), but have a few physics questions and comments here:

  1. I think the advantage of the Hexon (and Hexon-like) architectures in terms of protection from quasiparticle poisoning is somewhat exaggerated. The charging energy only offers protection from quasiparticles tunneling in from outside the Hexon, but not from quasiparticles which are excited by splitting Cooper pairs within the Hexon itself. Because of the large number of Cooper pairs, a population of such quasiparticles are expected to persist even for temperatures far below the superconducting gap.

AUTHORS' RESPONSE: We agree that the charging energy only offers protection from fermions tunneling between the island and external sources. These are the only processes that we refer to as quasiparticle poisoning in this context. We do not refer to quasiparticle processes that occur entirely within an island as quasiparticle poisoning. If one wished to do so, these two classes of processes should be distinguished as extrinsic and intrinsic. While the charging energy does not provide protection against intrinsic error processes, the topological gap does, as is usually the case for topological phases. We have clarified this in our paper.

  1. What are the requirements for measuring the collective parities of 4 (or more) MZMs (needed to go beyond single-qubit gates) without learning anything about the individual parities?

AUTHORS' RESPONSE: The requirement for measuring collective parities of 2N MZMs is that the measurement corresponds to a single interference loop that involves all 2N MZMs, and does not allow for any shorter loops involving fewer MZMs. In particular, this means fermions must not be able to pass directly between the various quantum dots involved, otherwise it would short-cut the interference loop.

Perhaps related to this, I do not understand the reason behind the statement (on page 11) that two-qubit operations require the ancillary pairs (of MZMs) to begin and end in the same state.

AUTHORS' RESPONSE: The ancillary pairs in a two-hexon operation should begin and end on the same state when considering forced-measurement protocols. This is to ensure that the initial and final encoding configurations are identical, i.e. one is comparing the same initial and final state space. When tracking methods are employed, this requirement is not needed.

  1. Flux noise is mentioned as a problem during measurement (which gets worse for increasing areas enclosed by the loop). I wonder if this is really a problem? I thought flux could be kept rather constant (important in most conventional superconducting qubits). And doesn't the energy of the QD shift in different ways (up or down) depending on the parity (which should then be easy to see even if there is some noise)?

AUTHORS' RESPONSE: Indeed, flux noise is expected to be a relatively minor source of errors in this scenario. We included it in the error model to be thorough. When estimating weight factors, we assigned it an estimated weight that was much smaller than the other error sources (such as those associated with tuning quantum dots and couplings).

Requested changes I think the weaknesses described above warrant some changes. As a minimum, I think the authors should say at an early point that the forced measurement scheme is not needed and, in fact, an inferior method. However, I think the paper (and in particular it's usefulness for a somewhat broader readership) would benefit greatly from writing it in a way that makes the crucial parts accessible without reading all the details which are less important.

AUTHORS' RESPONSE: We now state this specifically in the Introduction, when we outline the paper, and at the beginning of the Forced-Measurement Methods section, which the reader can skip over if he/she chooses.

REPORT 2:

Report The authors study optimal, measurement-based implementations of Clifford gates in a quantum computer based on Majorana zero modes. Specifically, the authors investigate alternatives to the well-known “forced-measurement” scheme for realizing Clifford gates in a Majorana-based quantum computer based on the "hexon" qubit architecture. These alternatives generally require fewer measurements in order to implement the same gates. As an example, the authors discuss a “Majorana-Pauli tracking” scheme as an alternative to the forced-measurement protocol. Here, measurement outcomes are tracked (not forced), so that the implemented logical gate will differ from the desired Clifford gate by a Pauli operator, which may be easily determined from the tracked measurement outcomes. These Pauli operators may be taken into account when implementing Clifford gates at later steps in a given computation.

The newest part of this work is presented in Sec. V, where the authors optimize the measurements needed to implement a logical Clifford gate, with respect to a weighting function which measures the difficulty of implementing a series of measurements. For a sample set of trial weights, the authors are able to completely optimize the measurement scheme for single-qubit logical gates by brute force, and perform a partial optimization for two-qubit gates. The latter presents difficulties since the cost of a brute-force optimization scales exponentially in the number of total measurements that are permitted in a gate implementation.

The results of this work are sound, and I recommend the paper for publication after minor revisions.

Requested changes (1) I believe there is a typo in the middle of the fourth paragraph of the introduction. The second half of the sentence

“We will describe different strategies and protocols for optimizing the generation of computational gates via measurement sequences, using measurements as the physical measurements as the generating set of operations”

reads incorrectly and should be fixed.

AUTHORS' RESPONSE: This typo has been corrected.

(2) A plot that shows the optimized difficulty weights for each logical gate would provide a simpler summary of the information presented in Tables I & II. This would more clearly show the relative difficulty of implementing various gates in the forced-measurement and Majorana-Pauli tracking schemes, and in the different hexon architectures.

AUTHORS' RESPONSE: We made a plot of the difficulty weights, but did not find it to be natural, nor significantly simpler to read. As such, we are choosing to leave the information in tables.

---

## Round 4 · List of Changes

● Emphasized that tracking methods are more efficient for MZM-based topological quantum computing earlier in the paper.

● Clarified the distinctions between extrinsic quasiparticle poisoning, which the charging energy protects against, and intrinsic quasiparticle processes, which the topological gap protects against.

● Added the stabilizer formalism for measurement sequences.

● Used stabilizer picture to improve numerical searches.

● Added a section on a proposed architecture for a measurement-only Majorana-based surface code, with stabilizer measurement operations that have been optimized using the methods of this paper.

---

## Editorial Decision

published